# A Near-Optimal Single-Loop Stochastic Algorithm for Convex Finite-Sum Coupled Compositional Optimization

**Bokun Wang**[1]  **Tianbao Yang**[1]

## Abstract

This paper studies a class of convex Finite-sum Coupled Compositional Optimization (cFCCO) problems for empirical X-risk minimization with applications including group distributionally robust optimization (GDRO) and learning with imbalanced data. To better address these problems, we introduce an efficient single-loop primal-dual block-coordinate stochastic algorithm called ALEXR. The algorithm employs block-coordinate stochastic mirror ascent with extrapolation for the dual variable and stochastic proximal gradient descent updates for the primal variable. We establish the convergence rates of ALEXR in both convex and strongly convex cases under smoothness and non-smoothness conditions of involved functions, which not only improve the best rates in previous works on smooth cFCCO problems but also expand the realm of cFCCO for solving more challenging non-smooth problems such as the dual form of GDRO. Finally, we derive lower complexity bounds, demonstrating the (near-)optimality of ALEXR within a broad class of stochastic algorithms for cFCCO. Experimental results on GDRO and partial Area Under the ROC Curve (pAUC) maximization demonstrate the promising performance of our algorithm.

## 1. Introduction

We revisit the following regularized finite-sum coupled compositional optimization problem (Wang & Yang, 2022):

$$\min_{x \in \mathcal{X}} F(x), \quad F(x) := \frac{1}{n} \sum_{i=1}^{n} f_i(g_i(x)) + r(x), \quad (1)$$

where $\mathcal{X} \subset \mathbb{R}^d$ is a convex and closed set, the inner function $g_i(x) = \mathbb{E}_{\zeta_i \sim \mathbb{P}_i}[g_i(x; \zeta_i)]$ is in expectation form, and the objective function $F(x)$ is convex.

In this paper, we study a special class of convex FCCO problems, termed cFCCO, which has a specific structure: The inner function $g_i : \mathcal{X} \to \mathbb{R}^m$ is convex and accessible via stochastic oracles (typically a loss function), while the outer function $f_i : \mathbb{R}^m \to \mathbb{R}$ is a convex and deterministic simple function (transformation) that is monotonically non-decreasing[1]. Besides, the regularizer $r$ is also convex. Although the above condition is sufficient but not necessary for the convexity of $F$, fully exploiting it allows us to design a single-loop algorithm that achieves better complexity than previous algorithms and applies to non-smooth problems with convergence guarantees. Moreover, we can prove its (near-)optimality by establishing lower complexity bounds.

Notably, the special structure of cFCCO arises in various machine learning applications, including group distributionally robust optimization (GDRO) (Sagawa et al., 2019; Soma et al., 2022; Zhang et al., 2023), sub-population fairness (Martinez et al., 2021), partial area under the ROC curve (pAUC) maximization (Zhu et al., 2022), and bipartite ranking (Rudin, 2009). We postpone detailed descriptions of some of these problems to Section 5 and Appendix B.

While several algorithms have been developed to solve the convex FCCO problem in (1) with theoretical guarantees of global convergence (Wang & Yang, 2022; Jiang et al., 2022), these methods are limited by critical drawbacks: First, these algorithms only have convergence guarantees under the strong assumption that $f_i$ and $g_i$ are both smooth and Lipschitz continuous. Second, their convergence rates have poor dependence on the target optimization error $\epsilon$, batch sizes, and, in the strongly convex case, the condition number. Third, these algorithms rely on nested inner loops where the number of iterations in each loop depends on problem-specific constants, increasing the difficulty of implementation.

To address these limitations, we leverage the structure of the problem. Using the convex conjugate of $f_i$ denoted

---

[1]Department of Computer Science and Engineering, Texas A&M University, College Station, TX, USA. Correspondence to: Tianbao Yang <tianbao-yang@tamu.edu>.

*Proceedings of the 42$^{nd}$ International Conference on Machine Learning*, Vancouver, Canada. PMLR 267, 2025. Copyright 2025 by the author(s).

---

[1]w.r.t. each coordinate of its input. Note that we only need the monotonicity of $f_i$ when $g_i$ is nonlinear.

by $f_i^*$, the cFCCO problem can be reformulated (1) into a convex-concave min-max problem:

$$\min_{x \in \mathcal{X}} \max_{y \in \mathcal{Y}} \left\{ \frac{1}{n} \sum_{i=1}^{n} \left[ g_i(x)^\top y^{(i)} - f_i^*(y^{(i)}) \right] + r(x) \right\}, \quad (2)$$

where $y^{(i)} \in \mathcal{Y}_i \subseteq \mathbb{R}_+^m$ is the $i$-th block of $y$, and $\mathcal{Y} = \mathcal{Y}_1 \times \ldots \times \mathcal{Y}_n \subseteq \mathbb{R}_+^{nm}$. This reformulation is motivated by state-of-the-art primal-dual methods for empirical risk minimization (ERM) (Alacaoglu et al., 2022) and the general convex-concave min-max optimization problem (Zhang et al., 2024). However, the problem in (2) presents unique challenges: (i) $g_i(x)$ may be neither linear nor deterministic, unlike that assumed in Alacaoglu et al. (2022); (ii) When $n$ is large, updating the entire dual variable $y$ in each iteration as in Zhang et al. (2024) becomes computationally prohibitive, motivating the block-coordinate dual update in our approach.

Our contributions can be summarized as follows:

• We propose a primal-dual block-coordinate stochastic algorithm named ALEXR to efficiently solve the cFCCO problems, which only requires $O(1)$ batch size per iteration.

• In both merely and strongly convex cases, the iteration complexities of ALEXR improve upon the iteration complexities in previous works (Wang & Yang, 2022; Jiang et al., 2022) on cFCCO problems with smooth $f_i$ and $g_i$ (See Table 1 for a detailed comparison). Besides, we also provide the convergence analysis of ALEXR for cFCCO problems with non-smooth $f_i$ and $g_i$ in convex and strongly convex cases.

• For cFCCO problems with smooth and non-smooth $f_i$, we prove lower complexity bounds for an abstract first-order update scheme, which covers our ALEXR and previous algorithms as special cases. These lower bounds demonstrate the near-optimality of our proposed algorithm.

## 2. Related Work

The cFCCO problem in (1) is related to several well-studied optimization problems. In Appendix A, we discuss the literature related to the min-max reformulation in (2).

**Convex Stochastic Compositional Optimization.** Several papers have studied the convex stochastic compositional optimization (SCO) problem $\min_{x \in \mathcal{X}} F(x) := \mathbb{E}_\xi[f_\xi(\mathbb{E}_\zeta[g_\zeta(x)])]$, where $\xi$ and $\zeta$ are mutually independent. When $F$ is merely convex and $f_\xi$ is smooth, the SCGD algorithm in Wang et al. (2017a) requires an $O(\frac{1}{\epsilon^4})$ iterations to find an $x_{\text{out}}$ s.t. $\mathbb{E}[F(x_{\text{out}}) - \min_x F(x)] \leq \epsilon$. When $F$ is $\mu$-strongly convex with unique minimizer $x_*$, SCGD requires $O(\frac{1}{\mu^2 \epsilon^{1.5}})$ iterations to make $\frac{\mu}{2} \mathbb{E} \|x_{\text{out}} - x_*\|_2^2 \leq \epsilon$. Further exploiting the smoothness of $g_\zeta$, Wang et al. (2017b) proposed ASC-PG, which improves the convergence rate to

$O(\frac{1}{\epsilon^{3.5}})$ for merely convex SCO and $O(\frac{1}{\mu \epsilon^{1.25}})$ for strongly convex SCO. When $f$ is convex and monotonically non-decreasing and $g$ is convex, Zhang & Lan (2020) reformulated the convex SCO problem as a min-max-max problem and proposed the stochastic sequential dual (SSD) method to obtain the optimal $O(\frac{1}{\epsilon^2})$ rate in the merely convex case and $O(\frac{1}{\mu \epsilon})$ rate in the $\mu$-strongly convex and smooth case. They also showed that the $O(\frac{1}{\epsilon^2})$ rate is optimal when $f_i$ is non-smooth, even if $F$ is strongly convex. However, these algorithms for SCO are inapplicable to cFCCO. In fact, FCCO introduces challenges beyond those in SCO: both the inner function $g_i$ and the distribution $\mathbb{P}_i$ in FCCO depend on the outer index $i$, whereas in SCO $\xi$ and $\zeta$ are mutually independent and the inner function $\mathbb{E}_\zeta[g_\zeta(x)]$ does not depend on $\xi$.

**Conditional stochastic optimization and FCCO.** Hu et al. (2020) studied a more general class of problems called conditional stochastic optimization: $\min_{x \in \mathcal{X}} F(x) := \mathbb{E}_\xi[f_\xi(\mathbb{E}_{\zeta|\xi}[g_\zeta(x; \xi)])]$. They proposed biased SGD (BSGD) with large batch sizes. For convex and smooth $F$, BSGD requires $O(\frac{1}{\epsilon^2})$ iterations and a large batch size of $B = O(\frac{1}{\epsilon})$ per iteration to find an $\epsilon$-accurate solution. For a $\mu$-strongly convex $F$, BSGD requires $O(\frac{1}{\mu \epsilon})$ iterations and a large batch size of $B = O(\frac{1}{\epsilon})$ per iteration to find an $\epsilon$-accurate solution. For the FCCO problem with convex $F$, Wang & Yang (2022) used the moving-average estimator and the restarting trick to find an $\epsilon$-accurate solution with only $O(1)$ batch size per iteration. In particular, restarted SOX has an iteration complexity of $O(\frac{n}{\mu^2 B S \epsilon})$ for the $\mu$-strongly convex problem and $O(\frac{n}{B S \epsilon^3})$ for the merely convex problem, where $S$ is the outer batch size and $B$ is the inner batch size. Jiang et al. (2022) proposed a variance-reduced algorithm MSVR that has improved iteration complexities $O(\frac{n}{S \sqrt{B} \mu \epsilon})$ for the $\mu$-strongly convex problem and $O(\frac{n}{S \sqrt{B} \epsilon^2})$ for the convex problem. However, the theoretical guarantees of MSVR have several limitations such as poor dependence on the batch size $B$, reliance on restrictive assumptions, and suboptimality for the $\mu$-strongly cFCCO problem with a smooth outer function $f_i$ (as we will demonstrate in Section 4).

**Applications.** FCCO serves as the algorithmic framework for optimizing a broad range of risk functions coined as empirical X-risk minimization (Yang, 2022). It has been applied to many machine learning problems, including optimizing listwise losses for learning to rank (Qiu et al., 2022), optimizing partial area under the ROC curve (pAUC) for imbalanced data classification (Zhu et al., 2022), group DRO (Hu et al., 2023b) and optimizing global contrastive losses for self-supervised learning (Yuan et al., 2022; Qiu et al., 2023). The proposed algorithm ALEXR for cFCCO is applicable to pAUC maximization and group DRO.

*Table 1.* Comparison of iteration complexities to achieve $\epsilon$-optimal solution of (1) in terms of $\mathbb{E}[F(x_{\text{out}}) - F(x_*)] \leq \epsilon$ in the merely convex case and $\frac{\mu}{2}\mathbb{E}\|x_{\text{out}} - x_*\|_2^2 \leq \epsilon$ in the $\mu$-strongly convex case, where $x_{\text{out}}$ is the output of each algorithm. $\tilde{O}$ hides poly $\log(1/\epsilon)$ factors. $S$ denotes the size of a batch $\mathcal{S} \subset [n]$ and $B$ denotes the size of batch $\mathcal{B}_i$ sampled from $\mathbb{P}_i$ for each $i \in \mathcal{S}$. In the "Monotonicity" column, $\uparrow$ means the function is monotonically non-decreasing. "N/A" means not applicable or not available. The ⟨gray parts⟩ are implications of Theorems 2 and 3.

| Method | Iteration Complexity | | Inner Batch Size $B$ | Outer Batch Size $S$ | Loops | Smoothness | Monotonicity | Convexity[†] |
|---|---|---|---|---|---|---|---|---|
| | Strongly Convex | Merely Convex | | | | | | |
| BSGD (Hu et al., 2020) | $O\left(\frac{1}{\mu\epsilon}\right)$ | $O\left(\frac{1}{\epsilon^2}\right)$ | $O\left(\frac{1}{\epsilon}\right)$ | $O(1)$ | Single | $f_i, g_i$ | None | $F$ |
| | | | $O\left(\frac{1}{\epsilon^2}\right)$ | $O(1)$ | Single | $g_i$ | None | $F$ |
| SOX-Boost (Wang & Yang, 2022) | $O\left(\frac{n}{\mu^2 BS\epsilon}\right)$ | $O\left(\frac{n}{BS\epsilon^3}\right)$ | $O(1)$ | $O(1)$ | Double | $f_i, g_i$ | None | $F$ |
| SOX (Wang & Yang, 2023) | $\tilde{O}\left(\frac{n}{\mu S\epsilon}\right)$ | N/A[★] | $O(1)$ | $O(1)$ | Single | $f_i, g_i$ | $f_i \uparrow$ | $f_i, g_i, r$ |
| MSVR (Jiang et al., 2022) | $O\left(\frac{n}{\mu\sqrt{B}\epsilon}\right)$ | $O\left(\frac{n}{\sqrt{B}\epsilon^2}\right)$ | $O(1)$ | $O(1)$ | Double | $f_i, g_i$ | None | $F$ |
| ALEXR (**This Work**) | $\tilde{O}\left(\max\left\{\frac{1}{\mu S\epsilon}, \frac{1}{\mu B\epsilon}, \frac{n}{BS\epsilon}\right\}\right)$ Theorem 1 (i) | $O\left(\max\{\frac{1}{S\epsilon^2}, \frac{n}{BS\epsilon^2}\}\right)$ | $O(1)$ | $O(1)$ | Single | $f_i, g_i$ | $f_i \uparrow$ | $f_i, g_i, r$ |
| | $\tilde{O}\left(\max\left\{\frac{1}{\mu\epsilon}, \frac{n}{BS\epsilon}\right\}\right)$ Theorem 1 (ii) | $O\left(\max\{\frac{1}{\epsilon^2}, \frac{n}{BS\epsilon^2}\}\right)$ | $O(1)$ | $O(1)$ | Single | $f_i$ | $f_i \uparrow$ | $f_i, g_i, r$ |
| | $O\left(\max\{\frac{1}{S\epsilon^2}, \frac{n}{BS\epsilon^2}\}\right)^{\sharp}$ | $O\left(\max\{\frac{1}{S\epsilon^2}, \frac{n}{BS\epsilon^2}\}\right)$ Theorem 3 | $O(1)$ | $O(1)$ | Single | $g_i$ | $f_i \uparrow$ | $f_i, g_i, r$ |
| | $O\left(\max\{\frac{1}{\epsilon^2}, \frac{n}{BS\epsilon^2}\}\right)^{\sharp}$ | $O\left(\max\{\frac{1}{\epsilon^2}, \frac{n}{BS\epsilon^2}\}\right)$ Theorem 2 | $O(1)$ | $O(1)$ | Single | None | $f_i \uparrow$ | $f_i, g_i, r$ |

[†] The sufficient condition (convexity of $f_i, g_i, r$ and monotonicity of $f_i$) for the convexity of $F$ can be met in the applications of interest described in Section 5 and Appendix B.

[★] The analysis of the merely convex case in Wang & Yang (2023) is under a weaker convergence measure that cannot be converted to the objective gap.

[♯] As shown in our lower complexity bound in Section 4, strong convexity does not yield a faster rate due to the compositional structure when the outer function $f_i$ is non-smooth. In Zhang & Lan (2020), a similar result has been established for convex stochastic compositional optimization.

## 3. Algorithm and Convergence Analysis

**Notations.** For a vector $y \in \mathbb{R}^{nm}$, we use $y^{(i)} \in \mathbb{R}^m$ to represent the $i$-th coordinate (block) of $y$, i.e., $y = (y^{(1)}, \ldots, y^{(n)})^\top$. We denote the Bregman divergence associated with $\psi_i : \mathbb{R}^m \to \mathbb{R}$ for any $u, v \in \mathbb{R}^m$ as $U_{\psi_i}(u, v) = \psi_i(u) - \psi_i(v) - \partial\psi_i(v)^\top(u - v)$ and define $U_\psi(y_1, y_2) := \sum_{i=1}^n U_{\psi_i}(y_1^{(i)}, y_2^{(i)})$ for $y_1, y_2 \in \mathbb{R}^{nm}$. For a function $g_i(x) = \mathbb{E}_{\zeta_i \sim \mathbb{P}_i}[g(x; \zeta_i)]$, we define the stochastic estimator based on the mini-batch $\mathcal{B}_i$ as $g_i(x; \mathcal{B}_i) := \frac{1}{|\mathcal{B}_i|}\sum_{\zeta_i \in \mathcal{B}_i} g_i(x; \zeta_i)$. Let $\mathcal{X}$ be a normed vector space with $\|\cdot\|_2$. For each $i \in [n]$, let $\mathcal{Y}_i \subset \mathbb{R}^m$ be a normed vector space with a general norm $\|\cdot\|$ and $\|\cdot\|_*$ be its dual norm. See Table 3 in the appendix for the full list of notations.

We make the following assumptions throughout the paper.

**Assumption 1.** The domain $\mathcal{X} \subseteq \mathbb{R}^d$ in (1) is a convex and closed set. Besides, the regularizatoin term $r$ is proper, lower-semicontinuous, and $\mu$-convex on $\mathcal{X}$, $\mu \geq 0$.

**Assumption 2.** $g_i$ is convex. Besides, there exists $C_g > 0$ such that $\|g_i(x) - g_i(x')\|_* \leq C_g \|x - x'\|_2, \forall x, x' \in \mathcal{X}$.

**Assumption 3.** $f_i : \mathbb{R}^m \to \mathbb{R}$ is convex. Besides, there exists $C_f > 0$ such that $|f_i(u) - f_i(u')| \leq C_f \|u - u'\|_*$, $\forall u, u' \in \mathcal{Y}_i^*$. If $g_i$ is nonlinear, we assume that $f_i$ is monotonically non-decreasing w.r.t. each coordinate of its input.

Assumption 3 implies that $\|y^{(i)}\| \leq C_f$ for all $y^{(i)} \in \mathcal{Y}_i$ and $\mathcal{Y}_i \subseteq \mathbb{R}_+^m, \forall i \in [n]$. Thus, (1) is equivalent to the

convex-concave problem (2) with a convex and compact $\mathcal{Y} = \mathcal{Y}_1 \times \ldots \times \mathcal{Y}_n$. Note that $f_i$ of all applications in Section 5 and Appendix B satisfy the assumption above.

Although the smoothness of $f_i$ and $g_i$ are not necessary in our work, incorporating them leads to better convergence rates. We say that $g_i : \mathcal{X} \to \mathbb{R}^m$ is $L_g$-smooth if it is differentiable and there exists $L_g > 0$ such that $\|g_i(x_1) - g_i(x_2) - \nabla g_i(x_2)(x_1 - x_2)\|_* \leq \frac{L_g}{2}\|x_1 - x_2\|_2^2, \forall x_1, x_2 \in \mathcal{X}$; Besides, we say that $f_i : \mathbb{R}^m \to \mathbb{R}$ is $L_f$-smooth if it is differentiable and there exists $L_f > 0$ such that $|f_i(u_1) - f_i(u_2) - \langle \nabla f_i(u_2), u_1 - u_2 \rangle| \leq \frac{L_f}{2}\|u_1 - u_2\|_*^2, \forall u_1, u_2 \in \mathcal{Y}_i^*$.

Lastly, we assume that the variances of the zeroth-order and first-order stochastic oracles are bounded.

**Assumption 4.** There exists finite $\sigma_0^2, \sigma_1^2, \delta^2$ such that

$$\mathbb{E}_{\zeta_i}\|g_i(x) - g_i(x; \zeta_i)\|_*^2 \leq \sigma_0^2,$$
$$\mathbb{E}_{\zeta_i}\|[g_i'(x)]^\top - [g_i'(x; \zeta_i)]^\top\|_{\text{op}}^2 \leq \sigma_1^2,$$
$$\frac{1}{n}\sum_{j=1}^n \|[g_j'(x)]^\top y^{(j)} - \frac{1}{n}\sum_{i=1}^n [g_i'(x)]^\top y^{(i)}\|_2^2 \leq \delta^2,$$

for any $x \in \mathcal{X}$, $g_i'(x) \in \partial g_i(x)$, and $y \in \mathcal{Y}$.

Under Assumptions 2, 3, the existence of $\delta^2$ is ensured since $\frac{1}{n}\sum_{j=1}^n \|[g_j'(x)]^\top y^{(j)} - \frac{1}{n}\sum_{i=1}^n [g_i'(x)]^\top y^{(i)}\|_2^2 \leq C_f^2 C_g^2$.

**Algorithm 1** ALEXR

1: **Initialize:** $x_0 \in \mathcal{X} \subseteq \mathbb{R}^d, y_0 \in \mathcal{Y} \subset \mathbb{R}_+^n$
2: **for** $t = 0, 1, \ldots, T - 1$ **do**
3:     Sample a batch $\mathcal{S}_t \subset \{1, \ldots, n\}, |\mathcal{S}_t| = S$
4:     **for** each $i \in \mathcal{S}_t$ **do**
5:        Sample batches $\mathcal{B}_t^{(i)}, \tilde{\mathcal{B}}_t^{(i)}$ of size-$B$ from $\mathbb{P}_i$
6:        Compute stochastic estimator $\tilde{g}_t^{(i)} = g_i(x_t; \mathcal{B}_t^{(i)}) + \theta(g_i(x_t; \mathcal{B}_t^{(i)}) - g_i(x_{t-1}; \mathcal{B}_t^{(i)}))$
7:        Update the $i$-th block of the dual variable $y_{t+1}^{(i)} = \arg\max_{v \in \mathcal{Y}_i} \{v\tilde{g}_t^{(i)} - f_i^*(v) - \tau U_{\psi_i}(v, y_t^{(i)})\}$
8:     **end for**
9:     For each $i \notin \mathcal{S}_t$, $y_{t+1}^{(i)} = y_t^{(i)}$
10:     Compute the stochastic gradient estimator $G_t = \frac{1}{S} \sum_{i \in \mathcal{S}_t} [g_i'(x_t; \tilde{\mathcal{B}}_t^{(i)})]^\top y_{t+1}^{(i)}$ based on the stochastic partial gradient $g_i'(x_t; \tilde{\mathcal{B}}_t^{(i)}) \in \partial g_i(x_t; \tilde{\mathcal{B}}_t^{(i)})$
11:     $x_{t+1} = \arg\min_{x \in \mathcal{X}} \{\langle G_t, x \rangle + r(x) + \frac{\eta}{2} \|x - x_t\|_2^2\}$
12: **end for**

## 3.1. A Primal-Dual Block-Coordinate Algorithm

We propose a stochastic algorithm, ALEXR (refer to Algorithm 1), to efficiently solve the cFCCO problem defined in (1) by leveraging its reformulation in (2). Due to the structure of (1), ALEXR begins each iteration by sampling a mini-batch $\mathcal{S}_t$ of size $S$ from $\{1, \ldots, n\}$ and, for each $i \in \mathcal{S}_t$, sampling two i.i.d. mini-batches $\mathcal{B}_t^i, \tilde{\mathcal{B}}_t^i$ of size $B$ from $\mathbb{P}_i$.

Since stochastic oracles $g_i(x; \zeta_i)$ are only available for those blocks $i \in \mathcal{S}_t$, ALEXR employs a block-coordinate stochastic update for the dual variable $y$, which occurs between line 5 and line 9 in Algorithm 1. For a sampled block $i \in \mathcal{S}_t$, the update of $y^{(i)}$ is based on the extrapolated stochastic gradient estimator $\tilde{g}_t^{(i)}$ of the linear coupling term $y^{(i)} g_i(x_t)$ in Step 6 with $\theta \in [0, 1]$, and a mirror-prox mapping w.r.t. to some strongly convex distance-generating function $\psi_i$. To ensure the proximal mapping in line 7 of Algorithm 1 can be efficiently computed, it is crucial to carefully select $\psi_i$:

• For any smooth outer function $f_i$, we can select $\psi_i = f_i^*$. For $u_t^{(i)} \in \partial f_i^*(y_t^{(i)})$, we can show that (see Lemma 1 in Appendix C):

$$y_{t+1}^{(i)} = \nabla f_i(u_{t+1}^{(i)}), \quad u_{t+1}^{(i)} = \frac{\tau u_t^{(i)} + \tilde{g}_t^{(i)}}{1 + \tau}, \forall i \in \mathcal{S}_t \quad (3)$$

If $f_i$ is a Legendre-type (proper, closed, strictly convex, and essentially smooth) function, ALEXR has a primal-only implementation similar to SOX (Wang & Yang, 2022) and MSVR (Jiang et al., 2022). To be specific, we can derive the following equivalent update of $u$ sequence such that

$$y_t^{(i)} = \nabla f_i(u_t^{(i)}).$$

$$u_{t+1}^{(i)} = \begin{cases} \frac{1}{1+\tau} u_t^{(i)} + \frac{\tau}{1+\tau} \tilde{g}_t^{(i)}, & \text{if } i \in \mathcal{S}_t \\ u_t^{(i)} & \text{o.w.} \end{cases} \quad (4)$$

• For a non-smooth outer function $f_i$, we can choose $\psi_i$ to be the quadratic function $\psi_i(\cdot) = \frac{1}{2} \|\cdot\|_2^2$. This choice requires that the proximal mapping of $f_i^*$ can be efficiently computed, a condition that holds for many simple functions (See Chapters 4 and 6 in Beck, 2017), e.g., the non-smooth function $f_i(\cdot) = \frac{1}{\alpha} \max\{\cdot, 0\}$ in the GDRO problem with Conditional Value at Risk (CVaR) divergence and the pAUC maximization problem described in Section 5.

Then, ALEXR updates the primal variable $x$ based on a stochastic proximal gradient descent update, where $G_t$ is a (sub)gradient estimator of the coupling term $\frac{1}{n} \sum_i y_{t+1}^{(i)} g_i(x_t)$ using an independent mini-batch $\tilde{\mathcal{B}}_t^{(i)}$.

## 3.2. Relation to Existing Algorithms

**Relation to SOX** (Wang & Yang, 2022). By setting $\theta = 0$, $\psi_i = f_i^*$ in ALEXR, the dual update and the gradient estimator become similar to that used in SOX. In particular, the update of $u_{t+1}^{(i)}$ in (3) becomes the moving average estimator, i.e., $u_{t+1}^{(i)} = (1 - \gamma)u_t^{(i)} + \gamma g_i(x_t; \mathcal{B}_t^{(i)})$, where $\gamma = \frac{1}{1+\tau}$. Hence, the updates of ALEXR with $\theta = 0$, $\psi_i = f_i^*$ reduces to SOX without gradient momentum, whose convergence is analyzed in Wang & Yang (2023) for strongly convex FCCO. However, establishing its convergence guarantee for the merely convex problem is still an open problem.

**Relation to MSVR** (Jiang et al., 2022). By setting $\psi_i = f_i^*$, there is only a subtle difference between MSVR and ALEXR, which gives ALEXR an advantage. In particular, the update of $u_{t+1}^{(i)}$ in (3) of ALEXR can be written as $u_{t+1}^{(i)} = (1 - \gamma)u_t^{(i)} + \gamma g_i(x_t; \mathcal{B}_t^{(i)}) + \gamma\theta(g_i(x_t; \mathcal{B}_t^{(i)}) - g_i(x_{t-1}; \mathcal{B}_t^{(i)}))$ with $\gamma = \frac{1}{1+\tau} < 1, \forall i \in \mathcal{S}_t$. This estimator is similar to the one used in MSVR except that the scaling factor before the correction term $(g_i(x_t; \mathcal{B}_t^{(i)}) - g_i(x_{t-1}; \mathcal{B}_t^{(i)}))$ in MSVR is $\beta = \frac{n-S}{S(1-\gamma)} + 1 - \gamma$, which could be much larger than 1. Notably, several existing works have reported better empirical performance using a $\gamma$ less than one (Jiang et al., 2022; Hu et al., 2023a;b), which is consistent with our setting and theory. Another difference between ALEXR and MSVR is that ALEXR does not use the variance-reduction technique (Cutkosky & Orabona, 2019) to compute the gradient estimator of the primal variable, which demands more memory and computational costs, albeit resulting in a worse oracle complexity compared to that of ALEXR for cFCCO.

**Relation to SAPD** (Zhang et al., 2024). When $n = 1$, the reformulation in (2) is a convex-concave saddle point prob-

lem, where SAPD is a representative stochastic algorithm. Both SAPD and ALEXR use an extrapolated estimator to update the dual variable $y$. The key difference is that SAPD updates the entire $y$ without assuming block separability of the dual domain, whereas ALEXR leverages this property to update only $y^{(i)}$ for those sampled blocks $i \in \mathcal{S}_t$. This design addresses the challenge of cFCCO: when $n$ is large, sampling from $\mathbb{P}_i$ and computing stochastic gradient estimator for each $i \in [n]$ is computationally expensive. Although ALEXR can be viewed as a block-coordinate variant of SAPD, its convergence analysis introduces several new challenges that are not present in the analysis of SAPD: (i) The block-coordinate updates of ALEXR lead to new challenges in convergence analysis, such as the dependence issue discussed in Section 3.3; (ii) ALEXR provides more flexibility to choose distance-generating functions $\psi_i$ other than the quadratic one in SAPD for the dual step; (iii) ALEXR and its convergence guarantees also apply to non-smooth problems, whereas SAPD focuses on smooth problems.

### 3.3. Convergence Analysis

For the convenience of analyzing the block-coordinate updates of the dual variable $y$, we define an auxiliary sequence:

$$\bar{y}_{t+1}^{(i)} = \arg\max_{v \in \mathcal{Y}_i} \left\{ v \tilde{g}_t^{(i)} - f_i^*(v) - \tau U_{\psi_i}(v, y_t^{(i)}) \right\}, \quad (5)$$

where $\tilde{g}_t = (\tilde{g}_t^{(1)}, \ldots, \tilde{g}_t^{(n)})^\top, \forall i \in [n]$. Note that only $\tilde{g}_t^{(i)}$ for those blocks $i \in \mathcal{S}_t$ are computed in the $t$-th iteration of Algorithm 1 while $\tilde{g}_t^{(i)}$ for those $i \notin \mathcal{S}_t$ are *virtual*. The reason we introduce the sequence $\{\bar{y}_t\}_{t \geq 0}$ is to decouple the dependence between $y_{t+1}$ and $\mathcal{S}_t$. Besides, for the options of $\psi_i$ listed in Section 3.1, we have $U_{f_i^*}(u, v) \geq \rho U_{\psi_i}(u, v)$ for some $\rho \geq 0$ and any $u, v \in \mathcal{Y}_i$: For example, $\rho = 1$ for a smooth $f_i$ and $\psi_i = f_i^*$ while $\rho = 0$ for a non-smooth $f_i$.

We define the objective function in (2) to be $L(x, y) := \frac{1}{n} \sum_{i=1}^n [g_i(x)^\top y^{(i)} - f_i^*(y^{(i)})] + r(x)$. After the $t$-th iteration of ALEXR, for any $x \in \mathcal{X}, y \in \mathcal{Y}$ we can obtain

$$\frac{\eta + \mu}{2} \|x - x_{t+1}\|_2^2 + \frac{\tau + \rho}{n} U_\psi(y, \bar{y}_{t+1}) \leq \quad (6)$$
$$\frac{\eta}{2} \|x - x_t\|_2^2 + \frac{\tau}{n} U_\psi(y, y_t) - (L(x_{t+1}, y) - L(x, \bar{y}_{t+1}))$$
$$+ R_t$$

where $R_t$ captures the remaining terms in (15). Notably, the term $L(x_{t+1}, y) - L(x, \bar{y}_{t+1})$ can be converted into the objective gap $F(x_{t+1}) - F(x)$ in an ergodic sense. In a standard convergence analysis based on potential functions (Bansal & Gupta, 2019), all terms in the potential function are expected to be non-expansive after a single iteration. However, it is not immediately clear whether the shaded part in (6) is non-expansive, regardless of whether

we choose $U_\psi(y, y_t)$ or $U_\psi(y, \bar{y}_t)$ as part of the potential function. It is worth noting that the issue above does not arise in the analysis of min-max optimization algorithms without block-coordinate updates such as that in Zhang et al. (2024), because $\bar{y}_{t+1} = y_{t+1}$ if the whole $y$ is updated.

#### 3.3.1. SMOOTH AND STRONGLY CONVEX CASE

When the outer function $f_i$ is smooth, we show that ALEXR can achieve the fast $O(\epsilon^{-1})$ rate for a strongly convex cFCCO problem. Under this setting, the min-max problem in (2) is strongly-convex-strongly-concave (SCSC) and a unique saddle point $(x_*, y_*)$ exists for the unique minimizer $x_*$ of the original problem in (1). We define that $\mathcal{G}_t$ is the $\sigma$-algebra generated by $\{\mathcal{B}_0, \mathcal{S}_0, \ldots, \mathcal{B}_{t-1}, \mathcal{S}_{t-1}, \mathcal{B}_t\}$ and $\mathcal{F}_t$ is the $\sigma$-algebra generated by $\{\mathcal{B}_0, \mathcal{S}_0, \ldots, \mathcal{B}_{t-1}, \mathcal{S}_{t-1}, \mathcal{B}_t, \mathcal{S}_t\}$. Note that $\mathcal{G}_t \subset \mathcal{F}_t$ and $y_{t+1}$ is $\mathcal{F}_t$-measurable. Since $x_*, y_*$ are independent of the randomness in the algorithm, we have

$$\mathbb{E}[U_\psi(y_*, y_{t+1}) \mid \mathcal{G}_t] = \frac{S}{n} U_\psi(y_*, \bar{y}_{t+1}) + \frac{n - S}{n} U_\psi(y_*, y_t).$$

Plug the equation above and $x = x_*, y = y_*$ into (6) can establish the contraction needed for potential-function-based convergence analysis. This leads to the following results, which holds for ALEXR with any strongly convex $\psi_i$, including $\psi_i(\cdot) = \frac{1}{2} \|\cdot\|_2^2$ and $\psi_i = f_i^*$ for a smooth $f_i$.

**Theorem 1.** *Suppose that Assumptions 1, 2, 3, 4 hold with $\mu > 0$ and $L_f$-smooth outer function $f_i$.*

- *(i) If $g_i$ is smooth, ALEXR with $\eta = \frac{\mu\theta}{1-\theta}, \tau = \frac{S}{n(1-\theta)}$, and $\theta = 1 - O(\epsilon)$ makes $\frac{\mu}{2} \mathbb{E} \|x_T - x_*\|_2^2 \leq \epsilon$ in $\tilde{O}(\max\{\frac{1}{\mu S}, \frac{1}{\mu B}, \frac{n}{BS}\}\epsilon^{-1})$ iterations;*

- *(ii) If $g_i$ is non-smooth, ALEXR with the same setting of $\eta, \tau$, and $\theta = 1 - O(\epsilon)$ leads to iteration complexity $\tilde{O}(\max\{\frac{1}{\mu}, \frac{n}{BS}\}\epsilon^{-1})$.*

*Remark* 1. On strongly convex cFCCO with smooth $f_i$ and $g_i$, ALEXR achieves $\tilde{O}(\max\{\frac{1}{\mu S \epsilon}, \frac{1}{\mu B \epsilon}, \frac{n}{BS\epsilon}\})$ iteration complexity, which improves upon the previously best-known $O(\frac{n}{\mu\sqrt{BS}\epsilon})$ achieved by MSVR (Jiang et al., 2022). Besides, we also provide the oracle complexity of ALEXR when the inner function $g_i$ is non-smooth, which is absent in previous work.

#### 3.3.2. CONVEX CASE WITH POSSIBLY NON-SMOOTH $f_i$

Now we shift our focus to the cFCCO problem with possibly non-smooth outer function $f_i$. In this case, we require $\psi_i = \frac{1}{2} \|\cdot\|_2^2$.

To derive a bound of the objective gap $\mathbb{E}[F(\bar{x}_T) - F(x_*)]$, $\bar{x}_T = \frac{1}{T} \sum_{t=0}^{T-1} x_t$, we will plug $x = x_*$ and $y = \tilde{y}_T^{(i)} \in \arg\max_{v \in \mathcal{Y}_i} \{v^\top g_i(\bar{x}_T) - f_i^*(v)\} \in \partial f_i(g_i(\bar{x}_T))$ into (6).

Unfortunately, the technique outlined in Section 3.3.1 does not address this issue because $\tilde{y}_T$ also depends on $\mathcal{S}_t$. We address this issue by introducing multiple virtual sequences to transform the shaded part in (6) into telescoping sums of several potential functions of these virtual sequences (See Lemma 9), a technique we extended from Nemirovski et al. (2009); Juditsky et al. (2011); Alacaoglu et al. (2022).

When $g_i$ is non-smooth, ALEXR achieves the same convergence rate for $\theta \in \{0, 1\}$, but choosing $\theta = 0$ saves $S$ inner function evaluations at $x_{t-1}$.

**Theorem 2.** *Suppose Assumptions 1, 2, 3, 4 hold and $g_i$ is non-smooth. ALEXR with $\psi_i = \frac{1}{2}\|\cdot\|_2^2$, $\theta = 0$, $\eta = O(\frac{1}{\epsilon})$, and $\tau = O(\frac{1}{B\epsilon})$ can make $\mathbb{E}[F(\bar{x}_T) - F(x_*)] \leq \epsilon$ in $O(\max\{1, \frac{\Omega_\mathcal{Y}^0}{BS}\}\epsilon^{-2})$ iterations, where $\Omega_\mathcal{Y}^0 := \mathbb{E}[U_\psi(\tilde{y}_T, y_0)] = \sum_{i=1}^n \mathbb{E}[U_{\psi_i}(\tilde{y}_T^{(i)}, y_0^{(i)})]$.*

When $g_i$ is smooth, setting the parameter $\theta = 1$ leverages the extrapolation term and yields a better convergence rate.

**Theorem 3.** *Suppose Assumptions 1, 2, 3, 4 hold and $g_i$ is smooth. ALEXR with $\psi_i = \frac{1}{2}\|\cdot\|_2^2$, $\theta = 1$, $\eta = O(\frac{1}{\epsilon})$, and $\tau = O(\frac{1}{B\epsilon})$ can make $\mathbb{E}[F(\bar{x}_T) - F(x_*)] \leq \epsilon$ in $O(\max\{\frac{1}{S}, \frac{\Omega_\mathcal{Y}^0}{BS}\}\epsilon^{-2})$ iterations, where $\Omega_\mathcal{Y}^0 := \mathbb{E}[U_\psi(\tilde{y}_T, y_0)] = \sum_{i=1}^n \mathbb{E}[U_{\psi_i}(\tilde{y}_T^{(i)}, y_0^{(i)})]$.*

*Remark 2.* The radius $\Omega_\mathcal{Y}^0$ is $O(n)$ in the worst case, but it can be much smaller than $O(n)$ when $\tilde{y}_T$ and $y_0$ exhibit some sparsity structure (an example is provided in Appendix F.1).

We compare the above results with that of MSVR. For non-smooth $f_i$, MSVR is not applicable. Theorem 2 implies that ALEXR can achieve the $O(\max\{\frac{1}{\epsilon^2}, \frac{n}{BS\epsilon^2}\})$ iteration complexity for the merely convex problem even $f_i$ is non-smooth. Furthermore, Theorem 3 also indicates that when both $f_i$ and $g_i$ are smooth, ALEXR achieves the $O(\max\{\frac{1}{S\epsilon^2}, \frac{n}{BS\epsilon^2}\})$ iteration complexity for the merely convex problem, improving upon the $O(\frac{n}{\sqrt{B}S\epsilon^2})$ iteration complexity in (Jiang et al., 2022) of the double-loop algorithm MSVR.

When $f_i$ is non-smooth, the strong convexity of the objective does not result in a better rate compared to the merely convex case, as we will demonstrate in Section 4.

## 4. Lower Complexity Bounds

In the previous section, we introduced the ALEXR algorithm and established the upper bounds of its iteration complexity and oracle complexity (i.e., the number of calls of stochastic oracles). In order to examine whether these bounds of ALEXR are (near-)optimal for the problem in (1), we examine the *lower* bounds by constructing "hard" instances of 1 for the following abstract first-order update scheme that subsumes ALEXR as well as previous algorithms (Zhang et al., 2024; Wang & Yang, 2022; Jiang et al., 2022)[2].

The abstract scheme starts with the initial spaces $\mathfrak{X}_0 = \mathfrak{G}_0 = \{\mathbf{0}_d\}$, $\mathfrak{Y}_0 = \{\mathbf{0}_n\}$, $\mathfrak{g}_0 = \{\mathbf{0}_m\}$ and progresses as follows in the $t$-th iteration: First, it samples a batch $\mathcal{S}_t \subset [n]$ and $\zeta_t^i, \tilde{\zeta}_t^i$ i.i.d. from $\mathbb{P}_i$. For those $i \in \mathcal{S}_t$,

$$\mathfrak{g}_{t+1}^{(i)} = \mathfrak{g}_t^{(i)} + \mathrm{span}\{g_i(\hat{x}; \zeta_t^{(i)}) \mid \hat{x} \in \mathfrak{X}_t\},$$
$$\mathfrak{Y}_{t+1}^{(i)} = \mathfrak{Y}_t^{(i)} + \mathrm{span}\{\mathrm{MP}(\hat{y}_i, \hat{g}_i) \mid \hat{y}_i \in \mathfrak{Y}_t^{(i)}, \hat{g}_i \in \mathfrak{g}_{t+1}^{(i)}\},$$

where "+" refers to the Minkowski sum, $\mathfrak{g}_t^{(i)}, \mathfrak{Y}_t^{(i)}$ are the $i$-th slices of the spaces $\mathfrak{g}_t, \mathfrak{Y}_t$, and $\mathrm{MP}(\hat{y}_i, \hat{g}_i) := \arg\max_v\{v\hat{g}_i - f_i^*(v) - \tau U_{\psi_i}(v, \hat{y}_i)\}$. For those $i \notin \mathcal{S}_t$, the corresponding slices remain unchanged, i.e., $\mathfrak{g}_{t+1}^{(i)} = \mathfrak{g}_t^{(i)}, \mathfrak{Y}_{t+1}^{(i)} = \mathfrak{Y}_t^{(i)}$. The spaces $\mathfrak{G}_t, \mathfrak{X}_t$ are updated as

$$\mathfrak{G}_{t+1} = \mathfrak{G}_t + \mathrm{span}\{G(\hat{x}, \hat{y}) \mid \hat{x} \in \mathfrak{X}_t, \hat{y} \in \mathfrak{Y}_{t+1}\},$$
$$\mathfrak{X}_{t+1} = \mathfrak{X}_t + \mathrm{span}\{\mathrm{QP}(\hat{x}, \hat{G}) \mid \hat{x} \in \mathfrak{X}_t, \hat{G} \in \mathfrak{G}_{t+1}\},$$

where we define $G(\hat{x}, \hat{y}) := \frac{1}{S}\sum_{i \in \mathcal{S}_t}[\nabla g_i(\hat{x}; \tilde{\zeta}_t^{(i)})]^\top \hat{y}^{(i)}$ and $\mathrm{QP}(\hat{x}, \hat{G}) := \arg\min_x\{x^\top \hat{G} + r(x) + \frac{\eta}{2}\|x - \hat{x}\|_2^2\}$.

For the problem with smooth outer function $f_i$, we construct a hard instance of (1) by setting $f_i$ to a variant of the Huber function and inner function $g_i$ to a linear function with some Bernoulli distributed noise. For the problem with non-smooth outer function $f_i$, we construct a hard instance by replacing the smooth Huber function $f_i$ with a monotonically non-decreasing hinge function. Details of the constructions and the proof are provided in Appendix E.

The construction of the noise and the hinge function $f_i$ for the non-smooth problem is adapted from Zhang & Lan (2020). Our contributions are twofold: First, we design an abstract scheme that supports block-coordinate updates and characterize how the optimal oracle complexity depends on the total number of blocks $n$; Second, we also construct a hard instance to prove the lower bound for the strongly convex cFCCO problem with a smooth outer function $f_i$, which highlights the near-optimality of our ALEXR in this setting and its significant improvement over previous algorithms.

**Theorem 4.** *For the $\mu$-strongly cFCCO problem in (1) with a smooth outer function $f_i$, any algorithm within the abstract scheme described above requires at least $\Omega(\max\{\frac{S}{\mu}, n\}\epsilon^{-1})$ oracles calls to find an $\bar{x}$ such that $\frac{\mu}{2}\mathbb{E}\|\bar{x} - x_*\|_2^2 \leq \epsilon$; Besides, For the cFCCO problem in (1) (whether merely convex or strongly convex) with a non-smooth outer function $f_i$, any algorithm within the abstract scheme described above requires at least $\Omega(n\epsilon^{-2})$ oracles calls to find an $\bar{x}$ such that $\mathbb{E}[F(\bar{x}) - F(x_*)] \leq \epsilon$.*

---

[2]It covers SOX (Wang & Yang, 2022) and MSVR (Jiang et al., 2022) when $\psi_i = f_i^*$. Besides, it also covers SAPD (Zhang et al., 2024) when $S = n$ and $\psi_i = \frac{1}{2}\|\cdot\|_2^2$.

Theorem 4 demonstrates that ALEXR is near-optimal in both cases. Furthermore, it also shows that the upper bounds established in Theorem 1 and Theorem 3 are tight.

# 5. Experiments

In this section, we present two main applications of the cFCCO problem: Group Distributionally Robust Optimization (GDRO) and Partial AUC Maximization (pAUC) with a restricted True Positive Rate (TPR). We then evaluate the empirical performance of our proposed ALEXR algorithm against previous baselines in these applications. More applications are discussed in Appendix B while additional details and experimental results can be found in Appendix G.

## 5.1. Group Distributionally Robust Optimization

The Group Distributionally Robust Optimization (GDRO) framework aims to train machine learning models that are robust across predefined subgroups (Sagawa et al., 2019). Suppose that there are $n$ predefined groups and the data distribution of the $i$-th group is $\mathbb{P}_i$. The $\phi$-divergence penalized GDRO can be formulated as

$$\min_w \max_{q \in \Delta_n} \left\{ \sum_{i=1}^{n} \left( q^{(i)} R_i(w) - \frac{\lambda}{n} \phi(nq_i) \right) \right\} + r(w), \quad (7)$$

where $\Delta_n$ is the $(n-1)$-dimensional probability simplex, $w$ is the model parameter, $R_i(w) := \mathbb{E}_{z \sim \mathbb{P}_i}[\ell(w; z)]$ is the risk of the $i$-th group, and $\phi : \mathbb{R}_+ \to \mathbb{R} \cup \{+\infty\}, \phi(1) = 0$.

Prior work (Sagawa et al., 2019; Zhang et al., 2023) discarded the $\phi$-divergence penalty, i.e., $\lambda = 0$ in (7), and consider the problem $\min_w \max_{i \in [n]} R_i(w)$, which minimizes the risk of the *worst* group. However, the model trained through worst-group risk minimization may be vacuous if the worst group is an outlier. Moreover, the sizes of groups may follow a long-tailed distribution such that multiple rare groups exist. To resolve these issues, we choose $\lambda > 0$ and consider the penalized GDRO problem with CVaR divergence $\phi = \mathbb{I}_{[0, \alpha^{-1}]}$ or $\chi^2$-divergence $\phi(t) = \frac{1}{2}(t-1)^2$.

The challenges of directly solving (7) using stochastic minmax algorithms lie in estimating the stochastic gradient of $q$ and controlling its variance when $n$ is large (Zhang et al., 2023). Alternatively, we can transform the above problem into an equivalent problem by duality (Levy et al., 2020):

$$\min_{w, c \in \mathbb{R}} \frac{\lambda}{n} \sum_{i=1}^{n} \phi^* \left( \frac{R_i(w) - c}{\lambda} \right) + c + r(w), \quad (8)$$

where $\phi^*$ is monotonically non-decreasing, e.g., $\phi^*(u) = \frac{1}{\alpha}(u)_+$ for CVaR divergence and $\phi^*(u) = \frac{1}{4}(u+2)_+^2 - 1$ for $\chi^2$-divergence. The dual formulation in (8) is recognized as a difficult open problem in Sagawa et al. (2019) due to the biased stochastic estimator (refer to footnote 4 in

their paper). When $R_i(w)$ is convex, we can solve the problem in (8) by viewing it as a cFCCO problem with a convex outer function $f_i(\cdot) = \lambda \phi^*(\cdot)$ and an inner function $g_i(x) = (R_i(w) - c)/\lambda$ that is jointly convex to $x = (w, c)$. In Appendix F, we compare the convergence rates and periteration costs of ALEXR with previous GDRO algorithms.

First, we empirically compare our proposed ALEXR with baseline methods on the GDRO problem in (7) with the CVaR divergence for the binary classification task. We consider the linear model $w$ and the logistic loss $\ell(w; z)$.

**Datasets.** We perform experiments on two datasets: a tabular dataset Adult (Becker & Kohavi, 1996) and an image dataset CelebA (Liu et al., 2015). For the Adult dataset, we construct 83 groups according to features such as race and the task is to predict the income. For the CelebA dataset, we constructed 160 groups based on binary attributes such as sex and the task is to determine whether a person possesses blonde hair. Please see Appendix G.1 for more details.

**Baselines.** We compare ALEXR with previous algorithms on the FCCO problem including BSGD (Hu et al., 2020), SOX (Wang & Yang, 2022), and MSVR (Jiang et al., 2022)[3]. Besides, we also compare ALEXR with previous algorithms for the GDRO problem, which include OOA (Sagawa et al., 2019) and SGD with up-weighting (SGD-UW) (Buda et al., 2018). OOA was originally proposed for the GDRO problem without a penalty term and we extend it to the CVaR-penalized GDRO based on an efficient algorithm for projection onto the capped simplex (Lim & Wright, 2016), where we use the implementation in Blondel (2019). To show the benefit of GDRO, we also include SGD based on empirical risk minimization (ERM) as a baseline, which neglects the group information. We do not compare with some other GDRO algorithms (Zhang et al., 2023; Soma et al., 2022) that do not support group sampling or do not apply to the CVaR-penalized problem. Moreover, algorithms for distributionally robust optimization (DRO) (Levy et al., 2020; Meng & Gower, 2023) are not applicable to the GDRO problem due to the stochastic oracles of per-group risk $R_i(w)$. We execute all algorithms for 5 runs with different random seeds. For a fair comparison, each algorithm samples 64 data points in each iteration. For SGD, these data points are sampled from the entire training dataset, whereas for other algorithms, we sample 8 groups and 8 data points per group. We tune the step sizes of all algorithms in the range $\{2, 5, 10\} \times 10^{\{-3, -2, -1\}}$. For SOX and MSVR, we also tune the momentum parameter $\gamma$ in the range $\{0.1, 0.5, 0.9\}$. For ALEXR, we choose the extrapolation parameter $\theta \in \{0.1, 1.0\}$ and $\psi_i(\cdot) = \frac{1}{2}(\cdot)^2$. For all algorithms, we choose the weight decay parameter 0.05 on

---

[3]MSVR was designed for FCCO problems with smooth $f_i$. We replace the gradient $\nabla f_i$ in MSVR with a subgradient to make it applicable to this GDRO problem with a non-smooth $f_i$.

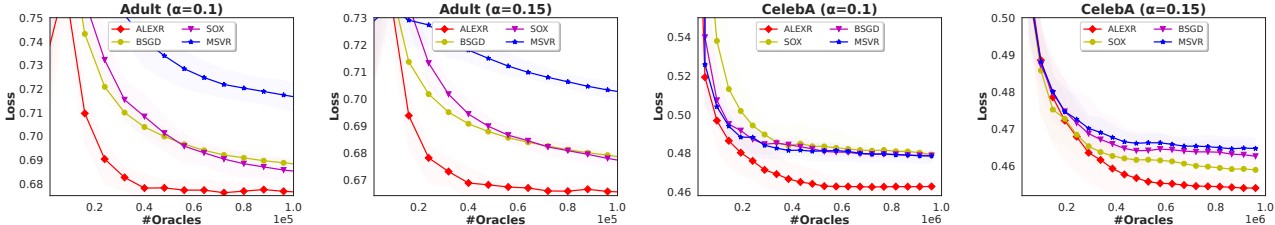

*Figure 1.* GDRO loss curves evaluated on the validation datasets during the training process with $\alpha = 0.1$ and $0.15$.

*Table 2.* Test accuracy (%) on the worst-$(\alpha n)$ groups with $\alpha = 0.1$ and $0.15$. The best accuracy is highlighted in black.

| | **Adult** | | **CelebA** | | Mean |
|---|---|---|---|---|---|
| Methods | $\alpha = 0.1$ | $\alpha = 0.15$ | $\alpha = 0.1$ | $\alpha = 0.15$ | |
| SGD | 0.71±0.20 | 1.87±0.25 | 2.75±0.08 | 4.89±0.10 | 2.56 |
| SGD-UW | 23.70±1.01 | 26.26±1.06 | 73.70±0.13 | 74.18±0.13 | 49.46 |
| OOA | 51.46±2.21 | 54.12±2.04 | 66.40±6.37 | 73.43± 0.79 | 61.35 |
| BSGD | 55.81±0.70 | **58.58±0.61** | 75.30±0.27 | 76.16±0.12 | 66.46 |
| SOX | 56.34 ±1.15 | 58.36±0.44 | 75.04±0.20 | 76.10±0.30 | 66.46 |
| MSVR | 47.78±1.06 | 49.49±0.95 | 75.34±0.28 | 76.17±0.09 | 62.20 |
| ALEXR | **56.58±0.69** | 58.52±0.71 | **75.79±0.05** | **76.29±0.07** | **66.80** |

the Adult dataset and 0.1 on the CelebA dataset.

**Results.** In Table 2, we report test accuracy for all algorithms on the worst-$(\alpha n)$ groups with $\alpha = 0.1$ and $0.15$. Besides, we plot the validation loss curves for FCCO algorithms sharing the same objective function (8) in Figure 1. First, we notice that the vanilla SGD performs poorly on the worst-$(\alpha n)$ groups' data. While the up-weighting trick offers some improvement for SGD, its effectiveness still falls short of Group DRO algorithms. Among GDRO algorithms, our proposed ALEXR exhibits faster convergence compared to baseline methods. ALEXR also achieves superior test accuracy compared to baseline methods in most cases.

### 5.2. Partial AUC Maximization with Restricted TPR

The Area Under the ROC Curve (AUC) is a more informative metric than accuracy for assessing the performance of binary classifiers in the context of imbalanced data (Yang & Ying, 2022). In scenarios influenced by diagnostic or monetary considerations, the primary objective may be to maximize the partial AUC (pAUC) with a specified lower bound $\alpha$ for the true positive rate (TPR). As shown in Zhu et al. (2022), a surrogate objective for maximizing pAUC with restricted TPR is formulated as

$$\min_{w \in \mathbb{R}^d} \frac{1}{n_+ n_-} \sum_{a_i \in \mathcal{S}_+^{\uparrow}[1, n_+(1-\alpha)]} \sum_{a_j \in \mathcal{S}_-} L(w; a_i, a_j), \quad (9)$$

Here $\mathcal{S}_+, \mathcal{S}_-$ are the sets of positive/negative data, $w$ refers to the model and $L(w; a_i, a_j) = \ell(h_w(a_j) - h_w(a_i))$ represents a continuous pairwise surrogate loss, where $h_w(a_i)$ de-

notes the prediction score for data $a_i$. Additionally, $\mathcal{S}_+^{\uparrow}[1, k]$ the bottom-$k$ positive data based on the prediction scores. In particular, $\ell$ is a convex and monotonically non-decreasing function, ensuring the consistency of the surrogate objective (Gao & Zhou, 2015). Based on the duality (Levy et al., 2020), the problem in (9) is equivalent to

$$\min_{w, s \in \mathbb{R}} \frac{1}{n_+(1-\alpha)} \sum_{a_i \in \mathcal{S}_+} \left[ \frac{1}{n_-} \sum_{a_j \in \mathcal{S}_-} L(w; a_i, a_j) - s \right]_+$$
$$+ s,$$

which is a cFCCO problem with $f_i = (\cdot)_+$ and $g_i(w, s) = \frac{1}{n_-} \sum_{a_j \in \mathcal{S}_-} L(w; a_i, a_j) - s$ jointly convex to $(w, s)$.

In our experiments, we consider linear prediction model $w$ and two different lower bounds $\alpha$ of TPR: 0.5 and 0.75.

**Baselines.** Apart from BSGD, SOX, and MSVR, we also include SGD with over-sampling for the cross-entropy (CE) loss and the SOTA algorithm (Zhu et al., 2022) as baselines. In each iteration, each algorithm samples an equal number of positive and negative data points (16 for each), which is based on the convergence theory in Zhu et al. (2022).

**Datasets.** We perform experiments on four datasets: Covtype, Cardiomegaly, Lung-mass, and Higgs. The Covtype and Higgs datasets are from the LibSVM repository (Chang & Lin, 2011). To create imbalanced datasets, we randomly remove 99.5% positive data from Covtype and 99.9% positive data from Higgs. For Covtype, we randomly allocate 75% of the data for training and 25% for validation. For Higgs, we randomly select 500,000 data points for vali-

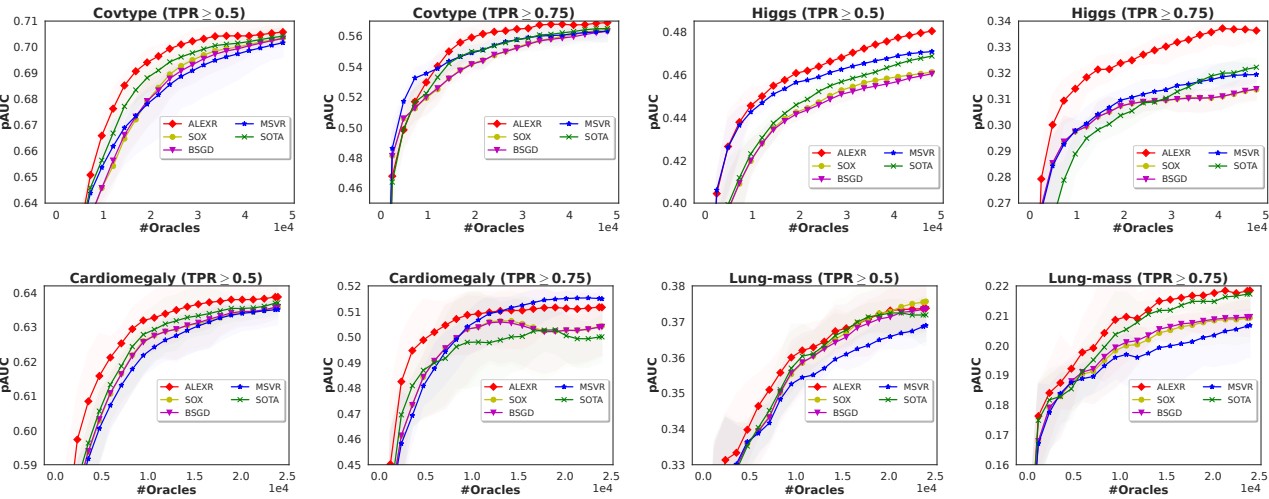

*Figure 2.* Partial AUC evaluated on the validation datasets during the training process under TPR$\geq 0.5$ and TPR$\geq 0.75$.

dation and the rest as training data. Cardiomegaly and Lung-mass are two imbalanced datasets that share the same collection of Chest X-ray images and different label annotations from the MedMNIST repository (Yang et al., 2023), where we use the default splits. We vectorize each 28×28 image in Cardiomegaly/Lung-mass datasets as a data point. Statistics of datasets are listed in Table 5 of the appendix.

**Results.** In Figure 2, we compare the pAUC curves during training. First, the results suggest that optimizing the surrogate loss in (9) outperforms optimizing the CE loss for maximizing pAUC with a restricted TPR. Moreover, ALEXR demonstrates overall superior performance when compared to other baselines including the SOTA algorithm specifically designed for pAUC maximization.

## 6. Conclusion and Discussion

In this paper, we study a class of convex FCCO problems, called cFCCO, via its min-max reformulation (2). We propose a single-loop primal-dual block-coordinate stochastic algorithm called ALEXR, which achieves improved iteration complexities compared to previous works on both merely and strongly convex cFCCO problems with smooth $f_i$ and $g_i$. We also establish the iteration complexities of ALEXR when either $f_i$ or $g_i$ is non-smooth. Furthermore, we present lower complexity bounds to show that the convergence rate of ALEXR is near-optimal among first-order stochastic methods for cFCCO problems. Finally, we note that it remains an open problem to prove similar complexities as in this work for cFCCO with concave outer functions $f_i$ such as the logarithmic function, which has broad applications in machine learning.

## Acknowledgments

We are deeply grateful to Guanghui Lan for his invaluable feedback on this paper. We are also thankful to Stephen J. Wright for bringing the analysis of the duality gap for PureCD to our attention. We thank Guanghui Wang for the initial discussion of the problem. We also thank anonymous reviewers for their comments. BW and TY were partially supported by by National Science Foundation Award #2306572 and #2147253.

## Impact Statement

This paper presents work whose goal is to advance the field of Machine Learning. There are many potential societal consequences of our work, none of which we feel must be specifically highlighted here.

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

*Table 3.* Notations we use throughout the paper.

| | **Basic** | |
|:---:|:---|:---:|
| $d$ | Number of the model parameters | |
| $n$ | Number of summands in cFCCO | (1) |
| $\mathbb{R}_+$ | Set of non-negative real numbers | Below (2) |
| $(x)_+$ | $\max\{x, 0\}$ | |
| $[n]$ | Set $\{1, 2, \ldots, n\}$ | |
| $y^{(i)}$ | The $i$-th block of size $m$ in the vector $y \in \mathbb{R}^{nm}$ | |
| $a \vee b$ | $\max(a, b)$ for $a, b \in \mathbb{R}$ | |
| $a \wedge b$ | $\min(a, b)$ for $a, b \in \mathbb{R}$ | |
| $a \asymp b$ | There exists $c, C > 0$ such that $cb \le a \le Cb$ for $a, b > 0$ | |
| $\mathbb{I}_E$ | $= 1$ if the event $E$ is true and $= 0$ otherwise | Appendix E |
| | **Standard** | |
| $f^*$ | The convex conjugate of a function $f$ | |
| $\mu$ | The strong convexity constant | Assumption 1 |
| $\psi_i$ | Strictly convex and differentiable $\psi_i : \mathbb{R}^m \to \mathbb{R}$ | |
| $\mathcal{X}$ | Convex and closed domain of the model $x$ | (1) |
| $\mathcal{Y}$ | The decomposable domain of the dual variable $y$, $\mathcal{Y} = \mathcal{Y}_1 \times \ldots \times \mathcal{Y}_n, \mathcal{Y}_i \subseteq \mathbb{R}_+^m$ | (2) |
| $\mathcal{Y}_i$ | A normed vector space in $\mathbb{R}_+^m$ with norm $\|\cdot\|$ | (2) |
| $y^{(i)}$ | The $i$-th size-$m$ block of a vector $y \in \mathbb{R}^{nm}$ | |
| $\mathcal{Y}_i^*$ | Dual space of $\mathcal{Y}_i$ with norm $\|\cdot\|_*$ | |
| $U_{\psi_i}(u, v)$ | Bregman divergence $\psi_i(u) - \psi_i(v) - \langle \nabla \psi_i(v), u - v \rangle$ for $u, v \in \mathbb{R}^m$ associated with $\psi_i$ | |
| $U_\psi(y_1, y_2)$ | Defined as $\sum_{i=1}^n U_{\psi_i}(y_1^{(i)}, y_2^{(i)})$ for $y_1, y_2 \in \mathbb{R}^{nm}$ | |
| $D_{\psi_i, \mathcal{Y}_i}$ | The diameter $[\max_{v \in \mathcal{Y}_i} \psi_i(v) - \min_{v \in \mathcal{Y}_i} \psi_i(v)]^{1/2}$ of a set $\mathcal{Y}_i$ w.r.t. $\psi_i$ | |
| $D_\mathcal{Y}$ | Defined as $[\sum_{i=1}^n D_{\psi_i, \mathcal{Y}_i}^2]^{1/2}$ | |
| $D_\mathcal{X}$ | $D_{\psi_i, \mathcal{X}}$ with $\psi_i = \frac{1}{2} \|\cdot\|_2^2$ | |
| $\|T_i\|_{\mathrm{op}}$ | Operator norm $\sup_{x \in \mathcal{X}} \left\{ \frac{\|T_i x\|_*}{\|x\|_2} \right\}$ for a linear operator $T_i : \mathcal{X} \to \mathcal{Y}_i^*$ | |
| $\|T_i^*\|_{\mathrm{op}}$ | Operator norm $\sup_{x \in \mathcal{X}} \left\{ \frac{\|T_i x\|_*}{\|x\|_2} \right\}$ of the adjoint operator $T_i^* : \mathcal{Y}_i \to \mathcal{X}$ | |
| $\mathrm{span}(S)$ | Linear span of a set $S$ of vectors | |
| $\Delta_n$ | The $(n-1)$-dimensional probability simplex $\Delta_n \subset \mathbb{R}_+^n$ | |
| $C_g$ | Lipschitz constant of inner function $g_i$ | Assumption 2 |
| $C_f$ | Lipschitz constant of outer function $f_i$ | Assumption 3 |
| $\sigma_0^2, \sigma_1^2$ | Variance upper bounds of zero-th and first order oracles of $g_i$ | Assumption 4 |
| $\delta^2$ | Variance upper bound of compositional stochastic gradient | Assumption 4 |
| | **Algorithm** | |
| $\mathcal{S}_t$ | Batch $\mathcal{S}_t \subset [n]$ of size $S$ sampled in the $t$-th iteration | Algorithm 1 |
| $\mathcal{B}_t^{(i)}, \tilde{\mathcal{B}}_t^{(i)}$ | Two i.i.d. batches of size-$B$ sampled from $\mathbb{P}_i$ in the $t$-th iteration. Batches $\{\mathcal{B}_t^{(i)}\}_{i \in \mathcal{S}_t}$ are actually sampled in Algorithm 1; Batches $\{\mathcal{B}_t^{(i)}\}_{i \notin \mathcal{S}_t}$ are virtual and only for analysis | |
| $g_i(x; \mathcal{B}^{(i)})$ | Stochastic estimator $\frac{1}{B} \sum_{\zeta_i \in \mathcal{B}^{(i)}} g(x; \zeta_i)$ based on the mini-batch $\mathcal{B}^{(i)}$ sampled from $\mathbb{P}_i$ | for $g_i(x)$ in (1) |
| $\eta, \tau, \theta$ | Hyperparameters of ALEXR | Algorithm 1 |
| $\tilde{g}_t^{(i)}$ | Extrapolated stochastic estimator for the inner function value | Line 6 in Algorithm 1 |
| | **Analysis** | |
| $F(x)$ | Objective function of cFCCO | (1) |
| $L(x, y)$ | Objective function of the min-max reformulation | (2) |
| $x_*$ | A minimizer of $F(x)$ in (1) | |
| $(x_*, y_*)$ | Unique saddle point of $L(x, y)$ in (2) when $f_i$ is smooth and $r$ is strongly convex | |
| $\bar{y}_t$ | Dual auxiliary sequence defined for the convenience of convergence analysis | (5) |
| $\bar{x}_T$ | Time-average primal iterate $\frac{1}{T} \sum_{t=0}^{T-1} x_{t+1}$ | |
| $\tilde{y}_T$ | $\tilde{y}_T^{(i)} \in \arg\max_{v \in \mathcal{Y}_i} \{v^\top g_i(\bar{x}_T) - f_i^*(v)\} \in \partial f_i(g_i(\bar{x}_T))$ | |
| $\hat{y}_t$ | Virtual sequence defined in the proofs of Lemma 6 and Lemma 10 | Below (20) and (34) |
| $\Gamma_t$ | Defined as $\frac{1}{n} \sum_{i=1}^n (g_i(x_t) - g_i(x_{t-1}))^\top (y^{(i)} - y_t^{(i)})$ | Lemma 6 |
| $\Upsilon_t^x$ | Defined as $\frac{1}{2} \mathbb{E} \|x_* - x_t\|_2^2$ | Section D.1.2 |
| $\Upsilon_t^y$ | Defined as $\frac{1}{S} \mathbb{E} U_\psi(y_*, y_t)$ | Section D.1.2 |
| $\hat{y}_t$ | Virtual sequence constructed in Lemma 9 | Below (6) |
| $\hat{\check{y}}_t, \check{y}_t$ | Virtual sequences constructed in Lemma 10 | Below (33) |
| $\bar{\bar{y}}_T$ | Time-average dual auxiliary iterate $\frac{1}{T} \sum_{t=0}^{T-1} \bar{y}_{t+1}$ | Below (39) |

# A. Other Related Work

The min-max reformulation of cFCCO in (2) is closely related to the following prior work.

**Convex-Concave Saddle Point (SP) Problem.** The saddle point (SP) problem $\min_{x \in \mathcal{X}} \max_{y \in \mathcal{Y}} L(x, y)$ that is $\mu_x$-convex in $x$ and $\mu_y$-concave in $y$ ($\mu_x, \mu_y \geq 0$) has been thoroughly studied. We refer to the SP problem with $\mu_x, \mu_y > 0$ as a strongly-convex-strongly-concave (SCSC) problem while those with $\mu_x, \mu_y = 0$ as a convex-concave (CC) problem. A saddle point $(x_*, y_*)$, if it exists, satisfies the condition $L(x_*, y) \leq L(x_*, y_*) \leq L(x, y_*), \forall (x, y) \in \mathcal{X} \times \mathcal{Y}$. Besides, the SP problem is closely related to the more general monotone variational inequalities (VI), which involve finding a point $z_* = (x_*, y_*)$ such that $\langle \Phi(z_*), z - z_* \rangle \geq 0, \Phi(z) = (\partial_x L(x, y), -\partial_y L(x, y)), \forall z \in \mathcal{X} \times \mathcal{Y}$. To assess the optimality of any point $(x_{\text{out}}, y_{\text{out}}) \in \mathcal{X} \times \mathcal{Y}$, we can employ the concept of the duality gap, defined as $\text{Gap}(x_{\text{out}}, y_{\text{out}}) := \max_{x,y} \{L(x_{\text{out}}, y) - L(x, y_{\text{out}})\}$, and for SCSC problems, we can also use $D(x_{\text{out}}, y_{\text{out}}) := \frac{\mu_x}{2} \|x_{\text{out}} - x_*\|_2^2 + \frac{\mu_y}{2} \|y_{\text{out}} - y_*\|_2^2$. The convergence rate is quantified by measuring the number of iterations required to find an $\epsilon$-approximate saddle point or an $\epsilon$-approximate solution to the VI, satisfying one of the following conditions: (i) $\text{Gap}(x_{\text{out}}, y_{\text{out}}) \leq \epsilon$; (ii) $D(x_{\text{out}}, y_{\text{out}}) \leq \epsilon$; (iii) $\langle \Phi(z_{\text{out}}), z_{\text{out}} - z \rangle \leq \epsilon$.

Accessing exact oracles such as $\nabla_x L$ and $\nabla_y L$ may not be feasible in many real-world scenarios. Instead, the available resources provide only unbiased stochastic estimators, denoted as $\tilde{\nabla}_x L$ and $\tilde{\nabla}_y L$, with variances bounded by $\sigma^2$. This limitation has prompted the development of numerous algorithms tailored for addressing the stochastic saddle point problem (SPP) and the more general stochastic variational inequalities (SVIs). For instance, the stochastic mirror descent (SMD) method (Nemirovski et al., 2009) achieves the optimal convergence rate of $O(\frac{1}{\epsilon^2})$ for non-Lipschitz SVIs. For Lipschitz monotone SVIs, the stochastic mirror-prox (SMP) method (Juditsky et al., 2011) attains the optimal rate of $O(\frac{1}{\epsilon} + \frac{\sigma^2}{\epsilon^2})$. For SCSC and non-smooth SP problems, Yan et al. (2020) establish the $\tilde{O}(\frac{1}{\epsilon} + \frac{1}{\mu_x \epsilon} \vee \frac{1}{\mu_y \epsilon})$ rate with probability $1 - p$. Hsieh et al. (2019) propose a single-call stochastic extragradient (SSEG) method that achieves a rate of $O(\frac{1}{\epsilon} + \frac{\sigma^2}{\mu_x \epsilon} \vee \frac{\sigma^2}{\mu_y \epsilon})$ for Lipschitz and strongly monotone SVIs. More recently, several works have devised stochastic algorithms for both the SSP and SVI problems, achieving (near-)optimal deterministic and stochastic convergence rates simultaneously. Zhang et al. (2024) introduce the SAPD algorithm, which reaches a convergence rate of $\tilde{O}(\frac{1}{\mu_x} \vee \frac{1}{\mu_y} + \frac{1}{\sqrt{\mu_x \mu_y}} + \frac{\sigma^2}{\mu_x \epsilon} \vee \frac{\sigma^2}{\mu_y \epsilon})$ for the SCSC problem and $O(\frac{1}{\epsilon} + \frac{\sigma^2}{\epsilon^2})$ for the CC problem. These algorithms cannot be directly applied to the min-max reformulation of cFCCO in (2) because the dual variable $y$ could be very high-dimensional, making it computationally infeasible to update the entire $y$. This challenge motivates the block-coordinate stochastic update in our algorithm ALEXR.

**Coordinate Methods for the Block-Separable Deterministic SP Problem.** A special class of bilinearly-coupled SP problem is in the form $\min_x \max_y L(x, y) := \frac{1}{n} \sum_{i=1}^n [y^{(i)} a_i^\top x - \phi_i(y^{(i)})] + r(x)$, where $L(x, y)$ is block-separable w.r.t. the dual variable $y$. One illustrative example is the primal-dual reformulation of the (regularized) empirical risk minimization (ERM) problem, denoted as $\min_x F(x)$, where $F(x)$ is defined as $F(x) := \frac{1}{n} \sum_{i=1}^n \ell(a_i^\top x) + r(x)$. This reformulation applies to data-label pairs $(a_i, b_i)_{i=1}^n$ in the context of a linear model. Particularly in scenarios with a significantly large value of $n$, the computational overhead of computing $\nabla_y L(x, y)$ and updating $y$ can become prohibitively expensive. In such cases, randomized coordinate methods offer a viable solution by reducing the per-iteration oracle cost from $O(n)$ to $O(1)$. The SPDC method (Zhang & Xiao, 2015) leads to $\tilde{O}(n + \sqrt{\frac{n}{\mu_x \mu_y}})$ convergence rate to make $\mathbb{E}[D(\bar{x}, \bar{y}] \leq \epsilon$ for the SCSC problem and $\tilde{O}\left(n + \frac{\sqrt{n}}{\epsilon}\right)$ rate to make $\mathbb{E}[F(\bar{x}) - F(x_*)] \leq \epsilon$ for the CC problem. Recently, Alacaoglu et al. (2022) extended the Pure-CD originally proposed in Alacaoglu et al. (2020) to incorporate importance sampling and exploit the potential sparsity in $A$. For the CC problem with dense $A$, Pure-CD not only achieves an improved rate of $O(n + \frac{\sqrt{n}}{\epsilon})$ to guarantee $\mathbb{E}[F(\bar{x}) - F(x_*)] \leq \epsilon$ but also attains a rate of $\tilde{O}(\frac{n}{\epsilon})$ to ensure $\mathbb{E}[\text{Gap}(\bar{x}, \bar{y})] \leq \epsilon$. It is worth noting that $\mathbb{E}[\text{Gap}(\bar{x}, \bar{y})] \leq \epsilon$ serves as a sufficient but not necessary condition for $\mathbb{E}[F(\bar{x}) - F(x_*)] \leq \epsilon$.

In addition to addressing the bilinearly-coupled block-separable saddle point (SP) problem, Hamedani et al. (2023) have extended their focus to the more general Convex-Concave (CC) problem, defined as $L(x, y) = \Phi(x, y) - \phi(y) + \sum_{i=1}^m h_i(x^{(i)})$. Their work establishes a convergence rate of $O\left(\frac{m}{\epsilon}\right)$ for a randomized block-coordinate primal-dual method, ensuring that $\mathbb{E}[\text{Gap}(\bar{x}, \bar{y})] \leq \epsilon$. Furthermore, Jalilzadeh et al. (2019) have delved into scenarios where $L(x, y)$ exhibits block-separability to both $x$ and $y$. In this context, $L(x, y)$ is defined as $L(x, y) = \Phi(x, y) - \sum_{i=1}^n \phi_i(y^{(i)}) + \sum_{j=1}^m h_j(x^{(j)})$. They introduce a doubly-randomized block-coordinate method to address such problems. It is worth emphasizing that all the works mentioned in this section (Zhang & Xiao, 2015; Alacaoglu et al., 2020; 2022; Jalilzadeh et al., 2019) rely on the assumption of having access to the exact $\nabla_x \Phi(x, y)$ and $\nabla_y \Phi(x, y)$. In contrast, our work addresses the more challenging problem where only stochastic oracles are available.

## B. Others Applications of cFCCO

In Section 5, we introduce two applications of cFCCO: Group Distributionally Robust Optimization (GDRO) and Partial AUC (pAUC) Maximization with a Restricted TPR. Here we provide more applications of cFCCO in machine learning.

**Robust Logistic Regression.** Consider a collection of data-label pairs, denoted as $(a_i, b_i)_{i=1}^n$. We can formulate the robust logistic regression problem as $\min_{x \in \mathcal{X}} \frac{1}{n} \sum_{i=1}^n \log(1 + \exp b_i \mathbb{E}[\mathcal{A}(a_i)^\top x \mid a_i]) + r(x)$. In this formulation, $\mathcal{A}(a_i)$ represents the perturbed data generated from an underlying distribution $\mathbb{P}_i$. This is a special case of (1), where the functions $f_i(\cdot)$ are convex and monotonically non-decreasing given by $f_i(\cdot) = \log(1 + \exp(b_i \cdot))$, and $g_i(x) = \mathbb{E}_{A(a_i) \sim \mathbb{P}_i}[\mathcal{A}(a_i)^\top x]$.

**Bellman Residual Minimization.** The task of approximating the value function, denoted as $V^\pi(s)$, for each state $s$ under policy $\pi$ using a linear mapping can be expressed as $\min_{x \in \mathcal{X}} \sum_{s=1}^S (\phi_s^\top x - \sum_{s'} \mathbb{P}_{s,s'}^\pi [r_{s,s'} + \gamma \cdot \phi_{s'}^\top x])^2$. In this formulation, $\phi_s$ and $\phi_{s'}$ are feature vectors representing states $s$ and $s'$, respectively. Additionally, $r_{s,s'}$ represents the random reward obtained during the transition from state $s$ to $s'$, $\gamma < 1$ is the discount factor, $\pi$ denotes the policy, and $\mathbb{P}_{s,s'}^\pi$ represents the probability of transitioning from state $s$ to $s'$ under policy $\pi$. This problem can be formulated as (1), where the functions $f_s(\cdot)$ are convex and given by $f_s(\cdot) = \frac{1}{S}(\cdot)^2$, and the affine function $g_s(x) = \phi_s^\top x - \sum_{s'} \mathbb{P}_{s,s'}^\pi [r_{s,s'} + \gamma \cdot \phi_{s'}^\top x]$.

**Bipartite Ranking.** Imbalanced data classification is usually tackled in the context of the bipartite ranking problem. There is often a desire to penalize those positive examples with lower scores. One approach is the $p$-norm push, introduced by Rudin (2009). It formulates the problem as $\min_{x \in \mathcal{X}} \frac{1}{n_+} \sum_{a_i \in \mathcal{D}_+} \left( \frac{1}{n_-} \sum_{a_j \in \mathcal{D}_-} \ell(s_x(a_j) - s_x(a_i)) \right)^p + r(x), p \geq 1$. Here, $\mathcal{D}_+$ and $\mathcal{D}_-$ represent positive and negative data sets. The function $s_x(a)$ denotes the ranking score of data point $a$, which is determined by a linear model parameterized by $x$. The loss function $\ell$ is non-negative, convex, and monotonically non-decreasing, for instance, $\ell(\cdot) = \exp(\cdot)$. The $p$-norm push method is in a special case of (1), where the functions $f_i(\cdot)$ are convex and monotonically non-decreasing and given by $f_i(\cdot) = (\cdot)^p$, and the convex function $g_i(x) = \frac{1}{n_+} \sum_{a_j \in \mathcal{D}_+} \ell(s_x(a_j) - s_x(a_i))$. One popular approach for retrieval problems is maximizing the precision or recall at top $k$ positions (prec/rec@k). Yang (2022) has formulated the problem as $\min_{x \in \mathcal{X}} \frac{1}{n_+} \sum_{a_i \in \mathcal{D}_+} \ell_1(\sum_{a_j \in \mathcal{D}_+ \cup \mathcal{D}_-} \ell_2(s_x(a_j) - s_x(a_i) - k)) + r(x)$, where $\ell_1, \ell_2$ are monotonically non-decreasing convex surrogate losses of the zero-one loss. Hence, maximizing precision or recall at top $k$ positions with a convex model $s_x(a)$ is covered by (1).

**Multi-Task GDRO.** GDRO can be extended to the multi-task setting. Consider a scenario with $n$ tasks and $m$ groups. We represent the data distribution for the $i$-th task and the $j$-th group as $\mathbb{P}_{i,j}$. Additionally, let $\ell(x; z)$ be the loss function associated with parameter $x$ on data point $z$. The Multi-Task GDRO, with a regularization term $r$, is formulated as $\min_{x \in \mathcal{X}} \frac{1}{n} \sum_{i=1}^n \max_{j \in [m]} \mathbb{E}[\ell(x; z_{ij})] + r(x)$. In this formulation, the functions $f_i(\cdot)$ are defined as $f_i(g_i) = \max_{j \in [m]}(g_{ij})$, and $g_{ij}(x) = \mathbb{E}[\ell(x; z_{ij})]$, where $g_i(x) = [g_{i1}(x), \ldots, g_{im}(x)]$. Alternatively, we may consider the smooth $f_i(g_i) = \log \sum_{j \in [m]} \exp(g_{ij})$. This problem is particularly relevant for the scenario featuring a substantial number $n$ of tasks, such as identity prediction in human faces, with a limited number $m$ of groups (e.g., lightning conditions).

## C. Basic Lemmas

**Lemma 1.** *Suppose that* $y_0^{(i)} = f_i'(u_0^{(i)}) \in \partial f_i(u_0^{(i)})$ *for some* $u_0^{(i)} \in \mathbb{R}$ *and* $u_{t+1}^{(i)} = \begin{cases} \frac{\tau}{1+\tau} u_t^{(i)} + \frac{1}{1+\tau} \tilde{g}_t^{(i)}, & i \in \mathcal{S}_t \\ u_t^{(i)}, & i \notin \mathcal{S}_t. \end{cases}$

*Algorithm 1 with* $\psi_i = f_i^*$ *satisfies that* $y_t^{(i)} = f_i'(u_t^{(i)}) \in \partial f_i(u_t^{(i)})$ *for all* $i \in \{1, \ldots, n\}$ *and* $t \geq 0$.

*Proof.* We prove it by induction. The base case follows from the premise. Assume that $y_t^{(i)} = f_i'(u_t^{(i)}) \in \partial f_i(u_t^{(i)})$.

• Case I ($i \notin \mathcal{S}_t$): Note that $y_{t+1}^{(i)} = y_t^{(i)}$ and $u_{t+1}^{(i)} = u_t^{(i)}$. Thus, $y_{t+1}^{(i)} = f_i'(u_{t+1}^{(i)}) \in \partial f_i(u_{t+1}^{(i)})$.

• Case II ($i \in \mathcal{S}_t$): This part resembles Lemma 2 in Zhang & Lan (2020). Based on the update rule and the premise $y_t^{(i)} \in \partial f_i(u_t^{(i)})$, we have

$$
\begin{aligned}
y_{t+1}^{(i)} &= \arg\max_{y^{(i)}} \left\{ y^{(i)} \tilde{g}_t^{(i)} - f_i^*(y^{(i)}) - \tau \left( f_i^*(y^{(i)}) - (f_i^*)'(y_t^{(i)}) \cdot y^{(i)} \right) \right\} \\
&= \arg\max_{y^{(i)}} \left\{ \left( \frac{1}{1+\tau} \tilde{g}_t^{(i)} + \frac{\tau}{1+\tau} u_t^{(i)} \right) \cdot y^{(i)} - f_i^*(y^{(i)}) \right\} \in \partial f_i \left( \frac{1}{1+\tau} \tilde{g}_t^{(i)} + \frac{\tau}{1+\tau} u_t^{(i)} \right) = \partial f_i(u_{t+1}^{(i)}).
\end{aligned}
$$

$\square$

The following lemma is well-known and similar ideas have been used in Nemirovski et al. (2009); Juditsky et al. (2011).

**Lemma 2.** *Consider a martingale difference sequence $\Delta_t$ adapted to $\mathcal{F}_t$. Define a sequence $\{\hat{\pi}_t\}_t$:*

$$\hat{\pi}_0 = 0, \quad \hat{\pi}_{t+1} = \arg\min_v \{\langle -\Delta_t, v \rangle + \alpha U_\psi(v, \hat{\pi}_t)\},$$

*where we also assume that $\psi$ is $\mu_\psi$-strongly convex w.r.t. $\|\cdot\|$ ($\mu_\psi > 0$). For any $v$ (that possibly depends on $\Delta_t$) we have*

$$\mathbb{E}\left[\langle \Delta_t, v \rangle\right] \leq \mathbb{E}\left[\alpha U_\psi(v, \hat{\pi}_t) - \alpha U_\psi(v, \hat{\pi}_{t+1})\right] + \frac{1}{2\alpha\mu_\psi}\mathbb{E}\|\Delta_t\|_*^2.$$

*Proof.* Use the three-point inequality:

$$\langle -\Delta_t, \hat{\pi}_{t+1} - v \rangle \leq \alpha U_\psi(v, \hat{\pi}_t) - \alpha U_\psi(v, \hat{\pi}_{t+1}) - \alpha U_\psi(\hat{\pi}_{t+1}, \hat{\pi}_t).$$

Add $\langle -\Delta_t, \hat{\pi}_t - \hat{\pi}_{t+1} \rangle$ to both sides and use Young's inequality.

$$\langle -\Delta_t, \hat{\pi}_t - v \rangle \leq \alpha U_\psi(v, \hat{\pi}_t) - \alpha U_\psi(v, \hat{\pi}_{t+1}) - \alpha U_\psi(\hat{\pi}_{t+1}, \hat{\pi}_t) + \langle \Delta_t, \hat{\pi}_{t+1} - \hat{\pi}_t \rangle$$
$$\leq \alpha U_\psi(v, \hat{\pi}_t) - \alpha U_\psi(v, \hat{\pi}_{t+1}) - \alpha U_\psi(\hat{\pi}_{t+1}, \hat{\pi}_t) + \frac{\alpha\mu_\psi}{2}\|\hat{\pi}_{t+1} - \hat{\pi}_t\|^2 + \frac{1}{2\alpha\mu_\psi}\|\Delta_t\|_*^2.$$

If $\psi$ is $\mu_\psi$-strongly convex, we have $-U_\psi(\hat{\pi}_{t+1}, \hat{\pi}_t) \leq -\frac{\mu_\psi}{2}\|\hat{\pi}_{t+1} - \hat{\pi}_t\|^2$. Lastly, $\mathbb{E}_t[\Delta_t, \hat{\pi}_t] = 0$. $\square$

The following lemma combines Lemma 4 in Juditsky et al. (2011) and Lemma 7 in Zhang & Lan (2020).

**Lemma 3.** *Let $\Pi \subset \mathbb{R}^m$ be a non-empty closed and convex set and function $u(\pi)$ be $\mu$-strongly convex on $\Pi$ w.r.t. $\|\cdot\|$. Let $\hat{\pi}$ be generated via a prox-mapping with the argument $g + \delta$, $\hat{\pi} \leftarrow \arg\min_{\pi \in \Pi}\{\langle \pi, g + \delta - u'(\underline{\pi})\rangle + u(\pi)\}$ for some $\underline{\pi} \in \Pi$, where $\delta$ denotes a noise term with $\mathbb{E}[\delta] = 0$ and $\mathbb{E}[\|\delta\|_*^2] \leq \sigma_0^2$. Then, for $\bar{\pi}$ generated via a prox-mapping with the argument $g$, $\bar{\pi} \leftarrow \arg\min_{\pi \in \Pi}\{\langle \pi, g - u'(\underline{\pi})\rangle + u(\pi)\}$, we have*

$$\|\hat{\pi} - \bar{\pi}\| \leq \|\delta\|_* / \mu, \tag{10}$$
$$|\mathbb{E}\langle \hat{\pi}, \delta \rangle| \leq \sigma_0^2 / \mu. \tag{11}$$

For completeness, we present the proof of the lemma above. We do not claim any novelty here.

*Proof.* By the optimality condition of prox-mapping, we have

$$\langle u'(\hat{\pi}) - u'(\underline{\pi}) + g + \delta, \hat{\pi} - \pi \rangle \leq 0, \quad \forall \pi \in \Pi, \tag{12}$$
$$\langle u'(\bar{\pi}) - u'(\underline{\pi}) + g, \bar{\pi} - \pi \rangle \leq 0, \quad \forall \pi \in \Pi. \tag{13}$$

Choose $\pi = \bar{\pi}$ in (12) and $\pi = \hat{\pi}$ in (13). By combining (12) and (13), we have

$$\|\delta\|_* \|\hat{\pi} - \bar{\pi}\| \geq \langle \delta, \hat{\pi} - \bar{\pi} \rangle \geq \langle u'(\hat{\pi}) - u'(\bar{\pi}), \hat{\pi} - \bar{\pi} \rangle.$$

Since $u$ is $\mu$-strongly convex, we have $\langle u'(\hat{\pi}) - u'(\bar{\pi}), \hat{\pi} - \bar{\pi} \rangle \geq \mu \|\hat{\pi} - \bar{\pi}\|^2$. Thus, $\|\hat{\pi} - \bar{\pi}\| \leq \|\delta\|_* / \mu$.

Moreover, the triangle inequality leads to $|\mathbb{E}\langle \hat{\pi}, \delta \rangle| \leq |\mathbb{E}\langle \hat{\pi} - \bar{\pi}, \delta \rangle| + |\mathbb{E}\langle \bar{\pi}, \delta \rangle|$. Note that $\mathbb{E}\langle \bar{\pi}, \delta \rangle = 0$. Moreover, Cauchy-Schwartz inequality and (10) leads to

$$|\mathbb{E}\langle \hat{\pi}, \delta \rangle| \leq |\mathbb{E}\langle \hat{\pi} - \bar{\pi}, \delta \rangle| \leq \mathbb{E}[\|\hat{\pi} - \bar{\pi}\| \|\delta\|_*] \leq \mathbb{E}\|\delta\|_*^2 / \mu \leq \sigma_0^2 / \mu.$$

$\square$

Next, we present a basic inequality about the mirror proximal update. Similar results have been widely used in the literature, e.g., Lemma 3.8 in Lan (2020) and Lemma 7.1 in Hamedani & Aybat (2021).

**Lemma 4.** *Suppose that the function $\phi : \mathcal{X} \to \mathbb{R}$ is on a convex closed domain $\mathcal{X}$ and $\phi$ is $\mu$-convex ($\mu \geq 0$) with respect to a prox-function $U_\psi(x,y) := \psi(x) - \psi(y) - \langle \psi'(y), x - y \rangle$ for any $x, y \in \mathcal{X}$ with a generating function $\psi$, i.e., $\phi(x) \geq \phi(y) + \langle \phi'(y), x - y \rangle + \mu U_\psi(x,y), \forall x, y \in \mathcal{X}$. For $\hat{x} = \arg\min_{x \in \mathcal{X}} \{ \phi(x) + \eta U_\psi(\underline{x}, x) \}$, we have*

$$\phi(\hat{x}) - \phi(x) \leq \eta U_\psi(x, \underline{x}) - (\eta + \mu) U_\psi(x, \hat{x}) - \eta U_\psi(\hat{x}, \underline{x}), \quad \forall x \in \mathcal{X}. \tag{14}$$

*Proof.* By the definition of the prox-function $U_\psi(x, y)$, we have

$$U_\psi(x, \underline{x}) - U_\psi(x, \hat{x}) - U_\psi(\hat{x}, \underline{x})$$
$$= \psi(x) - \psi(\underline{x}) - \langle \psi'(\underline{x}), x - \underline{x} \rangle - \psi(x) + \psi(\hat{x}) + \langle \psi'(\hat{x}), x - \hat{x} \rangle - \psi(\hat{x}) + \psi(\underline{x}) + \langle \psi'(\underline{x}), \hat{x} - \underline{x} \rangle$$
$$= \langle \psi'(\hat{x}) - \psi'(\underline{x}), x - \hat{x} \rangle.$$

By the strong convexity of $\phi$ with respect to $\psi$, we have $\phi(x) - \phi(\hat{x}) \geq \langle \phi'(\hat{x}), x - \hat{x} \rangle + \mu U_\psi(x, \hat{x})$. The optimality condition of the prox-mapping implies that $\langle \phi'(\hat{x}) + \eta(\psi'(\hat{x}) - \psi'(\underline{x})), x - \hat{x} \rangle \geq 0$ for any $x \in \mathcal{X}$. Thus, we obtain $\langle \phi'(\hat{x}), x - \hat{x} \rangle \geq -\eta \langle \psi'(\hat{x}) - \psi'(\underline{x}), x - \hat{x} \rangle$ such that

$$\phi(x) - \phi(\hat{x}) \geq \langle \phi'(\hat{x}), x - \hat{x} \rangle + \mu U_\psi(x, \hat{x})$$
$$\geq \eta \langle \psi'(\underline{x}) - \psi'(\hat{x}), x - \hat{x} \rangle + \mu U_\psi(x, \hat{x}) \geq -\eta U_\psi(x, \underline{x}) + (\eta + \mu) U_\psi(x, \hat{x}) + U_\psi(\hat{x}, \underline{x}).$$

$\square$

## D. Convergence Analysis

The following lemma is the complete version of (6), which is the starting point for the convergence analysis of ALEXR. Recall that in all cases we have $U_{f_i^*}(u, v) \geq \rho U_{\psi_i}(u, v)$ for some $\rho \geq 0$ and any $u, v \in \mathcal{Y}_i$.

**Lemma 5.** *Under Assumption 1, the following holds for any $x \in \mathcal{X}, y \in \mathcal{Y}$ after the $t$-th iteration of Algorithm 1.*

$$L(x_{t+1}, y) - L(x, \bar{y}_{t+1}) \tag{15}$$
$$\leq \frac{\tau}{n} U_\psi(y, y_t) - \frac{\tau + \rho}{n} U_\psi(y, \bar{y}_{t+1}) - \frac{\tau}{n} U_\psi(\bar{y}_{t+1}, y_t) + \frac{1}{n} \sum_{i=1}^{n} (g_i(x_{t+1}) - \tilde{g}_t^{(i)})^\top (y^{(i)} - \bar{y}_{t+1}^{(i)}) + \frac{\eta}{2} \|x - x_t\|_2^2$$
$$- \frac{\eta + \mu}{2} \|x - x_{t+1}\|_2^2 - \frac{\eta}{2} \|x_{t+1} - x_t\|_2^2 + \frac{1}{n} \sum_{i=1}^{n} (g_i(x_{t+1}) - g_i(x))^\top \bar{y}_{t+1}^{(i)} - \langle G_t, x_{t+1} - x \rangle.$$

*Remark* 3. Note that Algorithm 1 only samples $\mathcal{B}_t^{(i)}$ for those $i \in \mathcal{S}_t$ and computes the extrapolated stochastic estimator $\tilde{g}_t^{(i)}$ for those $i \in \mathcal{S}_t$ in the $t$-th iteration. For those $i \notin \mathcal{S}_t$, the stochastic estimators $\{\tilde{g}_t^{(i)}\}_{i \notin \mathcal{S}_t}$ and the batches $\{\mathcal{B}_t^{(i)}\}_{i \notin \mathcal{S}_t}$ are virtual and introduced solely for the convenience of analysis. They are not required in the actual execution of Algorithm 1.

*Proof.* According to Lemma 4, the primal update rule implies that

$$- \langle G_t, x - x_{t+1} \rangle + r(x_{t+1}) - r(x) \leq \frac{\eta}{2} \|x - x_t\|_2^2 - \frac{\eta + \mu}{2} \|x - x_{t+1}\|_2^2 - \frac{\eta}{2} \|x_{t+1} - x_t\|_2^2. \tag{16}$$

Similarly, for all $i \in [n]$ the dual update rule implies that

$$(y^{(i)} - \bar{y}_{t+1}^{(i)})^\top \tilde{g}_t^{(i)} + f_i^*(\bar{y}_{t+1}^{(i)}) - f_i^*(y^{(i)}) \leq \tau U_{\psi_i}(y^{(i)}, y_t^{(i)}) - (\tau + \rho) U_{\psi_i}(y^{(i)}, \bar{y}_{t+1}^{(i)}) - \tau U_{\psi_i}(\bar{y}_{t+1}^{(i)}, y_t^{(i)}).$$

Average this equation over $i = 1, \ldots, n$.

$$\frac{1}{n} \sum_{i=1}^{n} \left( (y^{(i)} - \bar{y}_{t+1}^{(i)})^\top \tilde{g}_t^{(i)} + f_i^*(\bar{y}_{t+1}^{(i)}) - f_i^*(y^{(i)}) \right) \leq \frac{\tau}{n} U_\psi(y, y_t) - \frac{\tau + \rho}{n} U_\psi(y, \bar{y}_{t+1}) - \frac{\tau}{n} U_\psi(\bar{y}_{t+1}, y_t). \tag{17}$$

By the definition of $L(x, y)$ in (2), we have

$$
L(x_{t+1}, y) - L(x, \bar{y}_{t+1})
$$

$$
= \frac{1}{n} \sum_{i=1}^{n} g_i(x_{t+1})^\top y^{(i)} - \frac{1}{n} \sum_{i=1}^{n} f_i^*(y^{(i)}) + r(x_{t+1}) - \frac{1}{n} \sum_{i=1}^{n} g_i(x)^\top \bar{y}_{t+1}^{(i)} + \frac{1}{n} \sum_{i=1}^{n} f_i^*(\bar{y}_{t+1}^{(i)}) - r(x)
$$

$$
= \frac{1}{n} \sum_{i=1}^{n} \left( g_i(x_{t+1})^\top (y^{(i)} - \bar{y}_{t+1}^{(i)}) + f_i^*(\bar{y}_{t+1}^{(i)}) - f_i^*(y^{(i)}) + (g_i(x_{t+1}) - g_i(x))^\top \bar{y}_{t+1}^{(i)} \right) + r(x_{t+1}) - r(x).
$$

Combine the equation above with (16) and (17).

$$
L(x_{t+1}, y) - L(x, \bar{y}_{t+1})
$$

$$
\leq \frac{\tau}{n} U_\psi(y, y_t) - \frac{\tau + \rho}{n} U_\psi(y, \bar{y}_{t+1}) - \frac{\tau}{n} U_\psi(\bar{y}_{t+1}, y_t) + \frac{1}{n} \sum_{i=1}^{n} (g_i(x_{t+1}) - \tilde{g}_t^{(i)})^\top (y^{(i)} - \bar{y}_{t+1}^{(i)}) + \frac{\eta}{2} \|x - x_t\|_2^2
$$

$$
- \frac{\eta + \mu}{2} \|x - x_{t+1}\|_2^2 - \frac{\eta}{2} \|x_{t+1} - x_t\|_2^2 + \frac{1}{n} \sum_{i=1}^{n} (g_i(x_{t+1}) - g_i(x))^\top \bar{y}_{t+1}^{(i)} - \langle G_t, x_{t+1} - x \rangle. \tag{18}
$$

$\square$

### D.1. Convergence Analysis of the Smooth and Strongly Convex Case (Section 3.3.1)

First, we present several lemmas that upper-bound different terms in (15) with $(x, y) = (x_*, y_*)$, where $(x_*, y_*)$ is the unique saddle point of the strongly-convex-strongly-concave objective $L(x, y)$ of (2) when (1) is strongly convex and $f_i$ is smooth. We present our result with a general $\mu_\psi$-strongly convex distance-generating function $\psi_i$. For example, we have $\mu_\psi = \frac{1}{L_f}$ for $L_f$-smooth $f_i$ and $\psi_i = f_i^*$; Besides, we have $\mu_\psi = 1$ for non-smooth $f_i$ and $\psi_i = \frac{1}{2} \|\cdot\|_2^2$.

#### D.1.1. SUPPORTING LEMMAS

**Lemma 6.** *Under Assumptions 2 and 4, the following inequality holds for Algorithm 1 with $\theta < 1$ and any $\lambda_2, \lambda_3 > 0$.*

$$
\frac{1}{n} \sum_{i=1}^{n} \mathbb{E}(g_i(x_{t+1}) - \tilde{g}_t^{(i)})^\top (y_*^{(i)} - \bar{y}_{t+1}^{(i)}) \tag{19}
$$

$$
\leq \Gamma_{t+1} - \theta \Gamma_t + \frac{C_g^2 \|x_{t+1} - x_t\|_2^2}{2\lambda_2} + \frac{\theta C_g^2 \|x_t - x_{t-1}\|_2^2}{2\lambda_3} + \frac{(\lambda_2 + \lambda_3 \theta) U_\psi(\bar{y}_{t+1}, y_t)}{\mu_\psi n} + \frac{2(1 + 2\theta)\sigma_0^2}{B\mu_\psi(\rho + \tau)},
$$

*where $\Gamma_t := \frac{1}{n} \sum_{i=1}^{n} (g_i(x_t) - g_i(x_{t-1}))^\top (y_*^{(i)} - y_t^{(i)})$.*

*Proof.* The $\frac{1}{n} \sum_{i=1}^{n} \mathbb{E}(g_i(x_{t+1}) - \tilde{g}_t^{(i)})^\top (y_*^{(i)} - \bar{y}_{t+1}^{(i)})$ term can be decomposed as

$$
\diamond = \frac{1}{n} \sum_{i=1}^{n} (g_i(x_{t+1}) - \tilde{g}_t^{(i)})^\top (y_*^{(i)} - \bar{y}_{t+1}^{(i)}) \tag{20}
$$

$$
= \underbrace{\frac{1 + \theta}{n} \sum_{i=1}^{n} (g_i(x_t) - g_i(x_t; \mathcal{B}_t^{(i)}))^\top (y_*^{(i)} - \bar{y}_{t+1}^{(i)})}_{\text{I}} + \underbrace{\frac{1}{n} \sum_{i=1}^{n} (g_i(x_{t+1}) - g_i(x_t))^\top (y_*^{(i)} - \bar{y}_{t+1}^{(i)})}_{\text{II}}
$$

$$
+ \underbrace{\frac{\theta}{n} \sum_{i=1}^{n} (g_i(x_{t-1}) - g_i(x_t))^\top (y_*^{(i)} - \bar{y}_{t+1}^{(i)})}_{\text{III}} + \underbrace{\frac{\theta}{n} \sum_{i=1}^{n} (g_i(x_{t-1}; \mathcal{B}_t^{(i)}) - g_i(x_{t-1}))^\top (y_*^{(i)} - \bar{y}_{t+1}^{(i)})}_{\text{IV}}.
$$

Taking conditional expectations of terms I and IV leads to $\mathbb{E}[(g_i(x_t) - g_i(x_t; \mathcal{B}_t^{(i)}))^\top y_*^{(i)} \mid \mathcal{F}_{t-1}] = 0$ and $\mathbb{E}[(g_i(x_{t-1}) - g_i(x_{t-1}; \mathcal{B}_t^{(i)}))^\top y_*^{(i)} \mid \mathcal{F}_{t-1}] = 0$. $\forall i \in [n]$, define $\dot{y}_{t+1}^{(i)} := \arg\max_{v \in \mathcal{Y}_i} \{v^\top \bar{g}_t^{(i)} - f_i^*(v) - \tau U_{\psi_i}(v, y_t^{(i)})\}$ and $\bar{g}_t^{(i)} :=$

$g_i(x_t) + \theta(g_i(x_t) - g_i(x_{t-1}))$. Note that $\dot{y}_{t+1}^{(i)}$ is independent of $\mathcal{B}_t^{(i)}$ such that $\mathbb{E}[(g_i(x_t; \mathcal{B}_t^{(i)}) - g_i(x_t))^\top \dot{y}_{t+1}^{(i)} \mid \mathcal{F}_{t-1}] = 0$.

$$\mathbb{E}\left[(g_i(x_t; \mathcal{B}_t^{(i)}) - g_i(x_t))^\top \bar{y}_{t+1}^{(i)}\right] = \mathbb{E}\left[(g_i(x_t; \mathcal{B}_t^{(i)}) - g_i(x_t))^\top (\bar{y}_{t+1}^{(i)} - \dot{y}_{t+1}^{(i)})\right]$$

$$\overset{\text{Lemma 3}}{\leq} \frac{1}{\mu_\psi(\rho + \tau)} \mathbb{E}\|g_i(x_t) - g_i(x_t; \mathcal{B}_t^{(i)})\|_* \|(1 + \theta)(g_i(x_t) - g_i(x_t; \mathcal{B}_t^{(i)})) - \theta(g_i(x_{t-1}) - g_i(x_{t-1}; \mathcal{B}_t^{(i)}))\|_*$$

$$= \frac{(1 + \theta)\mathbb{E}\|g_i(x_t) - g_i(x_t; \mathcal{B}_t^{(i)})\|_*^2}{\mu_\psi(\rho + \tau)} + \frac{\theta\mathbb{E}\|g_i(x_t) - g_i(x_t; \mathcal{B}_t^{(i)})\|_* \|g_i(x_{t-1}) - g_i(x_{t-1}; \mathcal{B}_t^{(i)})\|_*}{\mu_\psi(\rho + \tau)}$$

$$\leq \frac{(1 + 1.5\theta)}{\mu_\psi(\rho + \tau)} \mathbb{E}\|g_i(x_t) - g_i(x_t; \mathcal{B}_t^{(i)})\|_*^2 + \frac{0.5\theta}{\mu_\psi(\rho + \tau)}\|g_i(x_{t-1}) - g_i(x_{t-1}; \mathcal{B}_t^{(i)})\|_*^2 \leq \frac{(1 + 2\theta)\sigma_0^2}{B\mu_\psi(\rho + \tau)},$$

$$\mathbb{E}\left[(g_i(x_{t-1}; \mathcal{B}_t^{(i)}) - g_i(x_{t-1}))^\top \bar{y}_{t+1}^{(i)}\right] = \mathbb{E}\left[(g_i(x_{t-1}; \mathcal{B}_t^{(i)}) - g_i(x_{t-1}))^\top (\bar{y}_{t+1}^{(i)} - \dot{y}_{t+1}^{(i)})\right] \leq \frac{(1 + 2\theta)\sigma_0^2}{B\mu_\psi(\rho + \tau)}.$$

Define $\Gamma_t := \frac{1}{n} \sum_{i=1}^n (g_i(x_t) - g_i(x_{t-1}))^\top (y_*^{(i)} - y_t^{(i)})$. II + III in (20) can be rewritten as

$$\text{II} + \text{III} = \frac{1}{n} \sum_{i=1}^n g_i(x_{t+1})^\top (y_*^{(i)} - \bar{y}_{t+1}^{(i)}) - \frac{1}{n} \sum_{i=1}^n g_i(x_t)^\top (y_*^{(i)} - \bar{y}_{t+1}^{(i)}) + \frac{\theta}{n} \sum_{i=1}^n (g_i(x_{t-1}) - g_i(x_t))^\top (y_*^{(i)} - \bar{y}_{t+1}^{(i)})$$

$$= \Gamma_{t+1} - \theta\Gamma_t + \frac{1}{n} \sum_{i=1}^n (g_i(x_{t+1}) - g_i(x_t))^\top (y_{t+1}^{(i)} - \bar{y}_{t+1}^{(i)}) + \frac{\theta}{n} \sum_{i=1}^n (g_i(x_{t-1}) - g_i(x_t))^\top (y_t^{(i)} - \bar{y}_{t+1}^{(i)})$$

$$\leq \Gamma_{t+1} - \theta\Gamma_t + \frac{1}{n} \sum_{i=1}^n \|g_i(x_{t+1}) - g_i(x_t)\|_* \|y_{t+1}^{(i)} - \bar{y}_{t+1}^{(i)}\| + \frac{\theta}{n} \sum_{i=1}^n \|g_i(x_{t-1}) - g_i(x_t)\|_* \|y_t^{(i)} - \bar{y}_{t+1}^{(i)}\|$$

$$\leq \Gamma_{t+1} - \theta\Gamma_t + \frac{C_g^2 \|x_{t+1} - x_t\|_2^2}{2\lambda_2} + \frac{\theta C_g^2 \|x_t - x_{t-1}\|_2^2}{2\lambda_3} + \frac{(\lambda_2 + \lambda_3\theta)U_\psi(\bar{y}_{t+1}, y_t)}{\mu_\psi n}.$$

$\square$

**Lemma 7.** *When $g_i$ is $L_g$-smooth and Assumptions 1, 2, 3, 4 hold, the following holds for Algorithm 1.*

$$\frac{1}{n}\mathbb{E} \sum_{i=1}^n (g_i(x_{t+1}) - g_i(x_*))^\top \bar{y}_{t+1}^{(i)} - \mathbb{E} \langle G_t, x_{t+1} - x_* \rangle \leq \frac{\frac{C_f^2\sigma_1^2}{B} + \frac{\delta^2}{S}}{\eta + \mu} + \frac{L_g C_f}{2} \|x_{t+1} - x_t\|_2^2. \tag{21}$$

*Proof.* We define $\Delta_t := \frac{1}{S} \sum_{i \in \mathcal{S}_t} [\nabla g_i(x_t; \tilde{\mathcal{B}}_t^{(i)})]^\top y_{t+1}^{(i)} - \frac{1}{n} \sum_{i=1}^n [\nabla g_i(x_t)]^\top \bar{y}_{t+1}^{(i)}$.

$$\frac{1}{n} \sum_{i=1}^n (g_i(x_{t+1}) - g_i(x_*))^\top \bar{y}_{t+1}^{(i)} - \langle G_t, x_{t+1} - x_* \rangle$$

$$= \frac{1}{n} \sum_{i=1}^n (g_i(x_{t+1}) - g_i(x_t))^\top \bar{y}_{t+1}^{(i)} + \frac{1}{n} \sum_{i=1}^n (g_i(x_t) - g_i(x_*))^\top \bar{y}_{t+1}^{(i)} + (\frac{1}{n} \sum_{i=1}^n [\nabla g_i(x_t)]^\top \bar{y}_{t+1}^{(i)} + \Delta_t)^\top (x_* - x_{t+1})$$

$$\overset{\diamond}{\leq} \frac{1}{n} \sum_{i=1}^n (g_i(x_{t+1}) - g_i(x_t))^\top \bar{y}_{t+1}^{(i)} + (\frac{1}{n} \sum_{i=1}^n [\nabla g_i(x_t)]^\top \bar{y}_{t+1}^{(i)})^\top (x_t - x_*) + (\frac{1}{n} \sum_{i=1}^n [\nabla g_i(x_t)]^\top \bar{y}_{t+1}^{(i)} + \Delta_t)^\top (x_* - x_{t+1})$$

$$= \frac{1}{n} \sum_{i=1}^n (g_i(x_{t+1}) - g_i(x_t))^\top \bar{y}_{t+1}^{(i)} + (\frac{1}{n} \sum_{i=1}^n [\nabla g_i(x_t)]^\top \bar{y}_{t+1}^{(i)})^\top (x_t - x_{t+1}) + \langle \Delta_t, x_* - x_{t+1} \rangle, \tag{22}$$

where $\diamond$ is due to the convexity of $g_i$ and $\mathcal{Y}_i \subseteq \mathbb{R}_+^m$. The first two terms in (22) can be bounded by the Lipschitz continuity

of $f_i$ and $\nabla g_i$.

$$
\frac{1}{n}\sum_{i=1}^{n}(g_i(x_{t+1}) - g_i(x_t))^\top \bar{y}_{t+1}^{(i)} + (\frac{1}{n}\sum_{i=1}^{n}[\nabla g_i(x_t)]^\top \bar{y}_{t+1}^{(i)})^\top (x_t - x_{t+1})
$$

$$
= \frac{1}{n}\sum_{i=1}^{n}(g_i(x_{t+1}) - g_i(x_t) - \nabla g_i(x_t)(x_{t+1} - x_t))^\top \bar{y}_{t+1}^{(i)}
$$

$$
\leq \frac{1}{n}\sum_{i=1}^{n}\left\|\bar{y}_{t+1}^{(i)}\right\| \|g_i(x_{t+1}) - g_i(x_t) - \nabla g_i(x_t)(x_{t+1} - x_t)\|_* \leq \frac{C_f}{n}\sum_{i=1}^{n}\|g_i(x_{t+1}) - g_i(x_t) - \nabla g_i(x_t)(x_{t+1} - x_t)\|_*.
$$

Due to the $L_g$-smoothness of $g_i$, we have

$$
\|g_i(x_{t+1}) - g_i(x_t) - \nabla g_i(x_t)(x_{t+1} - x_t)\|_* \leq \frac{L_g}{2}\|x_{t+1} - x_t\|_2^2.
$$

Thus, the first two terms in (22) can be upper bounded by

$$
\frac{1}{n}\sum_{i=1}^{n}(g_i(x_{t+1}) - g_i(x_t))^\top \bar{y}_{t+1}^{(i)} + (\frac{1}{n}\sum_{i=1}^{n}[\nabla g_i(x_t)]^\top \bar{y}_{t+1}^{(i)})^\top (x_t - x_{t+1}) \leq \frac{L_g C_f}{2}\|x_{t+1} - x_t\|_2^2. \tag{23}
$$

Besides, we have $\mathbb{E}[\langle \Delta_t, x_* \rangle \mid \mathcal{F}_{t-1}] = 0$. By the Lipschitz continuity of $f_i$ and the definition of the operator norm, we have $\|([\nabla g_i(x_t)]^\top - [\nabla g_i(x_t; \tilde{B}_t^{(i)})]^\top)\bar{y}_{t+1}^{(i)}\|_2 \leq \|[\nabla g_i(x_t)]^\top - [\nabla g_i(x_t; \tilde{B}_t^{(i)})]^\top\|_{\text{op}}\|\bar{y}_{t+1}^{(i)}\| \leq C_f\|[\nabla g_i(x_t)]^\top - [\nabla g_i(x_t; \tilde{B}_t^{(i)})]^\top\|_{\text{op}}$. According to Lemma 3 and Assumption 4, we can derive that

$$
-\mathbb{E}[\langle x_{t+1}, \Delta_t \rangle] \leq \frac{\mathbb{E}\|\Delta_t\|_2^2}{\mu + \eta} \leq \frac{1}{\mu + \eta}(\frac{\delta^2}{S} + \mathbb{E}\|\frac{1}{S}\sum_{i \in \mathcal{S}_t}([\nabla g_i(x_t)]^\top - [\nabla g_i(x_t; \tilde{B}_t^{(i)})]^\top)\bar{y}_{t+1}^{(i)}\|_2^2) \leq \frac{\frac{C_f^2\sigma_1^2}{B} + \frac{\delta^2}{S}}{\mu + \eta}. \tag{24}
$$

Then, combining (22), (23) and (24) leads to

$$
\frac{1}{n}\mathbb{E}\sum_{i=1}^{n}(g_i(x_{t+1}) - g_i(x_*))^\top \bar{y}_{t+1}^{(i)} - \mathbb{E}\langle G_t, x_{t+1} - x \rangle \leq \frac{\frac{C_f^2\sigma_1^2}{B} + \frac{\delta^2}{S}}{\mu + \eta} + \frac{L_g C_f}{2}\|x_{t+1} - x_t\|_2^2.
$$

$\square$

**Lemma 8.** *When $g_i$ is non-smooth and Assumptions 1, 2, 3, 4 hold, the following holds for Algorithm 1.*

$$
\frac{1}{n}\mathbb{E}\sum_{i=1}^{n}(g_i(x_{t+1}) - g_i(x_*))^\top \bar{y}_{t+1}^{(i)} - \mathbb{E}\langle G_t, x_{t+1} - x \rangle \leq \frac{\frac{C_f^2\sigma_1^2}{B} + \frac{\delta^2}{S} + 4C_f^2C_g^2}{\mu + \eta} + \frac{\eta + \mu}{4}\|x_{t+1} - x_t\|_2^2. \tag{25}
$$

*Proof.* Note that (22) and (24) still hold. Since $g_i$ is non-smooth, we need to bound the left-hand side of (23) in a different way. Based on the definition of the operator norm and the Lipschitz continuity of $g_i$, we have $\|g_i'(x_t)(x_t - x_{t+1})\|_* \leq \|g_i'(x_t)\|_{\text{op}}\|x_t - x_{t+1}\|_2 \leq C_g\|x_t - x_{t+1}\|_2$ such that

$$
\frac{1}{n}\sum_{i=1}^{n}(g_i(x_{t+1}) - g_i(x_t))^\top \bar{y}_{t+1}^{(i)} + \frac{1}{n}\sum_{i=1}^{n}([g_i'(x_t)]^\top \bar{y}_{t+1}^{(i)})^\top (x_t - x_{t+1})
$$

$$
= \frac{1}{n}\sum_{i=1}^{n}(g_i(x_{t+1}) - g_i(x_t))^\top \bar{y}_{t+1}^{(i)} + \frac{1}{n}\sum_{i=1}^{n}(g_i'(x_t)(x_t - x_{t+1}))^\top \bar{y}_{t+1}^{(i)}
$$

$$
\leq \frac{1}{n}\sum_{i=1}^{n}\|\bar{y}_{t+1}^{(i)}\|(\|g_i(x_{t+1}) - g_i(x_t)\|_* + \|g_i'(x_t)(x_t - x_{t+1})\|_*)
$$

$$
\leq 2C_f C_g\|x_{t+1} - x_t\|_2 \leq \frac{4C_f^2 C_g^2}{\eta + \mu} + \frac{\eta + \mu}{4}\|x_{t+1} - x_t\|_2^2, \tag{26}
$$

where $g_i'(x_t) \in \partial g_i(x_t)$. Merge (22), (24), and (26).

$$\frac{1}{n}\mathbb{E}\sum_{i=1}^{n}(g_i(x_{t+1}) - g_i(x_*))^\top \bar{y}_{t+1}^{(i)} - \mathbb{E}\langle G_t, x_{t+1} - x\rangle \leq \frac{\frac{C_f^2\sigma_1^2}{B} + \frac{\delta^2}{S} + 4C_f^2C_g^2}{\mu + \eta} + 0.25(\eta + \mu)\|x_{t+1} - x_t\|_2^2.$$

$\square$

### D.1.2. PROOF OF THEOREM 1

*Proof.* If $g_i$ is smooth, we combine (15), (19), and (21).

$$\mathbb{E}[L(x_{t+1}, y_*) - L(x_*, \bar{y}_{t+1})] \leq \frac{\tau + \rho\left(1 - \frac{S}{n}\right)}{S}\mathbb{E}[U_\psi(y_*, y_t)] - \frac{\tau + \rho}{S}\mathbb{E}[U_\psi(y_*, y_{t+1})] + \frac{\eta}{2}\mathbb{E}\|x_* - x_t\|_2^2$$

$$- \frac{\eta + \mu}{2}\mathbb{E}\|x_* - x_{t+1}\|_2^2 - \left(\frac{\tau}{n} - \frac{\lambda_2 + \lambda_3\theta}{\mu_\psi n}\right)\mathbb{E}\left[U_\psi(\bar{y}_{t+1}, y_t)\right] - \left(\frac{\eta}{2} - \frac{C_g^2}{2\lambda_2} - \frac{L_gC_f}{2}\right)\mathbb{E}\|x_{t+1} - x_t\|_2^2$$

$$+ \frac{\theta C_g^2}{2\lambda_3}\mathbb{E}\|x_t - x_{t-1}\|_2^2 + \mathbb{E}[\Gamma_{t+1} - \theta\Gamma_t] + \frac{2(1 + 2\theta)\sigma_0^2}{B\mu_\psi(\rho + \tau)} + \frac{\frac{C_f^2\sigma_1^2}{B} + \frac{\delta^2}{S}}{\eta + \mu}. \tag{27}$$

Define $\Upsilon_t^x := \frac{1}{2}\mathbb{E}\|x_* - x_t\|_2^2$ and $\Upsilon_t^y = \frac{1}{S}\mathbb{E}U_\psi(y_*, y_t)$. Note that $L(x_{t+1}, y_*) - L(x_*, \bar{y}_{t+1}) \geq 0$. Multiply both sides of (27) by $\theta^{-t}$ and do telescoping sum from $t = 0$ to $T - 1$. Add $\eta\theta^{-T}\Upsilon_T^x$ to both sides.

$$\eta\theta^{-T}\Upsilon_T^x \leq \sum_{t=0}^{T-1}\theta^{-t}((\eta\Upsilon_t^x + (\tau + \rho(1 - \frac{S}{n}))\Upsilon_t^y - \theta\mathbb{E}\Gamma_t) - ((\eta + \mu)\Upsilon_{t+1}^x + (\tau + \rho)\Upsilon_{t+1}^y - \mathbb{E}\Gamma_{t+1}))$$

$$+ \eta\theta^{-T}\Upsilon_T^x + \left(\frac{2(1 + 2\theta)\sigma_0^2}{\mu_\psi B(\rho + \tau)} + \frac{\frac{C_f^2\sigma_1^2}{B} + \frac{\delta^2}{S}}{\eta + \mu}\right)\sum_{t=0}^{T-1}\theta^{-t} - \sum_{t=0}^{T-1}\theta^{-t}\left(\frac{\tau}{n} - \frac{(\lambda_2 + \lambda_3\theta)}{\mu_\psi n}\right)\mathbb{E}[U_\psi(\bar{y}_{t+1}, y_t)]$$

$$- \sum_{t=0}^{T-1}\theta^{-t}\left(\frac{\eta}{2} - \frac{L_gC_f}{2} - \frac{C_g^2}{2\lambda_2} - \frac{C_g^2}{2\lambda_3}\right)\mathbb{E}\|x_{t+1} - x_t\|_2^2.$$

Let $\eta = \frac{\mu\theta}{1-\theta}$ such that $\theta = \frac{\eta}{\eta+\mu}$ and $\tau = \frac{\rho S}{n(1-\theta)} - \rho$ (where $\tau > 0$ if $\theta > 1 - \frac{S}{n}$) such that $\theta = \frac{\tau + \rho\left(1 - \frac{S}{n}\right)}{\tau + \rho}$. Then,

$$\sum_{t=0}^{T-1}\theta^{-t}((\eta\Upsilon_t^x + (\tau + \rho(1 - \frac{S}{n}))\Upsilon_t^y - \theta\mathbb{E}\Gamma_t) - ((\eta + \mu)\Upsilon_{t+1}^x + (\tau + \rho)\Upsilon_{t+1}^y - \mathbb{E}\Gamma_{t+1}))$$

$$= \eta\Upsilon_0^x + (\tau + \rho(1 - \frac{S}{n}))\Upsilon_0^y - \theta\mathbb{E}\Gamma_0 - \theta^{-T+1}\left((\eta + \mu)\Upsilon_T^x + (\tau + \rho)\Upsilon_T^y - \mathbb{E}\Gamma_T\right).$$

By setting $x_{-1} = x_0$, we have $\Gamma_0 = 0$. Besides, we have $-\Gamma_T \leq \frac{1}{n}\sum_{i=1}^{n}\|g_i(x_T) - g_i(x_{T-1})\|_* \|y_*^{(i)} - y_t^{(i)}\| \leq \frac{C_g}{n}\|x_T - x_{T-1}\|_2 \|y_* - y_T\|$. Thus,

$$\eta\theta^{-T}\Upsilon_T^x \leq \eta\Upsilon_0^x + (\tau + \rho(1 - \frac{S}{n}))\Upsilon_0^y - \theta^{-T+1}((\eta + \mu)\Upsilon_T^x + (\tau + \rho)\Upsilon_T^y - \frac{\eta}{\theta}\Upsilon_T^x - \frac{C_g}{n}\|x_T - x_{T-1}\|_2 \|y_* - y_T\|)$$

$$- \sum_{t=1}^{T-1}\theta^{-t+1}\underbrace{(((\eta + \mu)\Upsilon_{t+1}^x + (\tau + \rho)\Upsilon_{t+1}^y - \mathbb{E}\Gamma_{t+1}) - (\frac{\eta}{\theta}\Upsilon_t^x + (\tau + \rho(1 - \frac{S}{n}))/\theta\Upsilon_t^y - \mathbb{E}\Gamma_t))}_{\heartsuit}$$

$$+ \left(\frac{2(1 + 2\theta)\sigma_0^2}{\mu_\psi B(\rho + \tau)} + \frac{\frac{C_f^2\sigma_1^2}{B} + \frac{\delta^2}{S}}{\eta + \mu}\right)\sum_{t=0}^{T-1}\theta^{-t} - \sum_{t=0}^{T-1}\theta^{-t}\underbrace{\left(\frac{\tau}{n} - \frac{(\lambda_2 + \lambda_3\theta)}{\mu_\psi n}\right)}_{\heartsuit}\mathbb{E}[U_\psi(\bar{y}_{t+1}, y_t)]$$

$$- \sum_{t=0}^{T-1}\theta^{-t}\underbrace{\left(\frac{\eta}{2} - \frac{L_gC_f}{2} - \frac{C_g^2}{2\lambda_2} - \frac{C_g^2}{2\lambda_3}\right)}_{\heartsuit}\mathbb{E}\|x_{t+1} - x_t\|_2^2. \tag{28}$$

Note that $\eta + \mu - \frac{\eta}{\theta} = 0 \Leftrightarrow \theta = \frac{\eta}{\eta+\mu}$ such that $(\eta + \mu)\Upsilon_T^x - \frac{\eta}{\theta}\Upsilon_T^x \geq 0$ and $\frac{C_g}{n}\|x_T - x_{T-1}\|_2 \|y - y_T\| \leq \frac{C_g^2}{2\lambda_2}\|x_T - x_{T-1}\|_2^2 + \frac{\lambda_2}{2\mu_\psi n^2}U_\psi(y_*, y_T)$. To make the $\heartsuit$ terms in (28) be non-negative, we choose $\lambda_2 \asymp \frac{C_g\sqrt{S\rho\mu_\psi}}{\sqrt{n\mu}}$, $\lambda_3 \asymp \frac{C_g\sqrt{S\rho\mu_\psi}}{\sqrt{n\mu}}$ while ensuring that

$$1/\tau \leq O\left(\frac{\sqrt{n\mu\mu_\psi}}{C_g\sqrt{S\rho}}\right), \quad 1/\eta \leq O\left(\frac{\sqrt{S\rho\mu_\psi}}{C_g\sqrt{n\mu}} \wedge \frac{1}{L_g C_f}\right). \tag{29}$$

Since $\tau = \frac{\rho S}{n(1-\theta)} - \rho \Leftrightarrow \theta = \frac{\tau+\rho(1-\frac{S}{n})}{\tau+\rho}$, we have $\tau + \rho\left(1 - \frac{S}{n}\right) = \theta(\tau + \rho)$ and $(\tau + \rho)(1 - \theta) = \frac{\rho S}{n}$.

$$\mu\Upsilon_T^x \leq \mu\theta^T\Upsilon_0^x + \frac{\left(\tau + \rho\left(1 - \frac{S}{n}\right)\right)(1-\theta)}{\theta}\theta^T\Upsilon_0^y + \left(\frac{2(1+2\theta)\sigma_0^2}{\mu_\psi B(\rho+\tau)} + \frac{\frac{C_f^2\sigma_1^2}{B} + \frac{\delta^2}{S}}{\eta+\mu}\right)$$

$$= \mu\theta^T\Upsilon_0^x + (\tau+\rho)(1-\theta)\theta^T\Upsilon_0^y + \left(\frac{2(1+2\theta)\sigma_0^2}{\mu_\psi B(\rho+\tau)} + \frac{\frac{C_f^2\sigma_1^2}{B} + \frac{\delta^2}{S}}{\eta+\mu}\right)$$

$$= \mu\theta^T\Upsilon_0^x + \frac{\rho S}{n}\theta^T\Upsilon_0^y + \left(\frac{2(1+2\theta)\sigma_0^2}{\mu_\psi B(\rho+\tau)} + \frac{\frac{C_f^2\sigma_1^2}{B} + \frac{\delta^2}{S}}{\eta+\mu}\right).$$

We select $\theta = 1 - O\left(\frac{S}{n} \wedge \frac{\mu}{L_g C_f} \wedge \sqrt{\frac{\mu\rho\mu_\psi S}{C_g^2 n}} \wedge \frac{\mu_\psi B\rho S\epsilon}{\sigma_0^2 n} \wedge \frac{B\mu\epsilon}{C_f^2\sigma_1^2} \wedge \frac{S\mu\epsilon}{\delta^2}\right)$ to make (29) hold and

$$\frac{2(1+2\theta)\sigma_0^2}{\mu_\psi B(\rho+\tau)} + \frac{\frac{C_f^2\sigma_1^2}{B} + \frac{\delta^2}{S}}{\eta+\mu} \leq \frac{2(1+2\theta)(1-\theta)\sigma_0^2 n}{\mu_\psi B\rho S} + \frac{(1-\theta)\left(\frac{C_f^2\sigma_1^2}{B} + \frac{\delta^2}{S}\right)}{\mu} \leq \epsilon.$$

Since $\frac{1}{L_f} = \mu_\psi\rho$ when $f_i$ is $L_f$-smooth, the number of iterations needed by Algorithm 1 to make $\mu\Upsilon_T^x \leq \epsilon$ is

$$T = \tilde{O}\left(\frac{n}{S} + \frac{L_g C_f}{\mu} + \frac{C_g\sqrt{nL_f}}{\sqrt{S\mu}} + \frac{nL_f\sigma_0^2}{BS\epsilon} + \frac{C_f^2\sigma_1^2}{\mu B\epsilon} + \frac{\delta^2}{\mu S\epsilon}\right),$$

where $\tilde{O}(\cdot)$ hides the polylog$(1/\epsilon)$ factor. In the case that $g_i$ is non-smooth, we utilize (25) instead of (21). Correspondingly, we need to replace the blue term $\frac{L_g C_f}{2}$ in (27) by $0.25(\eta + \mu)$. Additionally, there is a $\frac{4C_f^2 C_g^2}{\eta+\mu}$ term on the right-hand side of (27). Following similar steps, we can get the iteration complexity to make $\mu\Upsilon_T^x \leq \epsilon$ is

$$T = \tilde{O}\left(\frac{n}{S} + \frac{C_g\sqrt{nL_f}}{\sqrt{S\mu}} + \frac{nL_f\sigma_0^2}{BS\epsilon} + \frac{C_f^2\sigma_1^2}{\mu B\epsilon} + \frac{\delta^2}{\mu S\epsilon} + \frac{C_f^2 C_g^2}{\mu\epsilon}\right).$$

$\square$

## D.2. Convergence Analysis of the Convex Case with Possibly Non-smooth $f_i$ (Section 3.3.2)

As discussed in Section 3.1, we can choose $\psi_i = \frac{1}{2}\|\cdot\|_2^2$ for the cFCCO problem with non-smooth $f_i$. We present our result with a general $\mu_\psi$-strongly convex and $L_\psi$-smooth distance-generating function $\psi_i$, which subsumes the quadratic one. The following lemma extends Lemma A.2 in Alacaoglu et al. (2022) to mini-batch sampling and a general smooth and strongly convex distance-generating function $\psi_i$.

**Lemma 9.** *The following holds for Algorithm 1 with $L_\psi$-smooth and $\mu_\psi$-strongly convex $\psi_i$ and any $\lambda_1 > 0$ satisfies that*

$$\mathbb{E}\left[\frac{\tau}{n}\left(U_\psi(y, y_t) - U_\psi(y, \bar{y}_{t+1})\right) - \frac{\tau}{n}U_\psi(\bar{y}_{t+1}, y_t)\right] \tag{30}$$

$$\leq \mathbb{E}\left[\frac{\tau}{S}(U_\psi(y, y_t) - U_\psi(y, y_{t+1})) + \frac{\tau\lambda_1}{S}(U_\psi(y, \hat{y}_t) - U_\psi(y, \hat{y}_{t+1}))\right] - \frac{\tau}{n}\left(1 - \frac{L_\psi^2}{\lambda_1\mu_\psi^2 S}\right)\mathbb{E}\left[U_\psi(\bar{y}_{t+1}, y_t)\right],$$

*where $\hat{y}_{t+1}^{(i)} = \arg\min_v\{-v^\top\Delta_t^{(i)} + \frac{n\lambda_1}{S}U_{\psi_i}(v, \hat{y}_t^{(i)})\}$ and $\Delta_t^{(i)} := -\frac{n}{S}\nabla\psi_i(y_{t+1}^{(i)}) + \nabla\psi_i(\bar{y}_{t+1}^{(i)}) + \frac{n-S}{S}\nabla\psi_i(y_t^{(i)})$.*

*Proof.* First, we can make the following decomposition.

$$\frac{\tau}{n}U_\psi(y, y_t) - \frac{\tau}{n}U_\psi(y, \bar{y}_{t+1}) - \frac{\tau}{n}U_\psi(\bar{y}_{t+1}, y_t) \tag{31}$$

$$= \frac{\tau}{S}U_\psi(y, y_t) - \frac{\tau}{S}U_\psi(y, y_{t+1}) - \frac{\tau}{n}U_\psi(\bar{y}_{t+1}, y_t) + \frac{\tau}{S}U_\psi(y, y_{t+1}) - \frac{\tau}{n}U_\psi(y, \bar{y}_{t+1}) + \frac{(S-n)\tau}{nS}U_\psi(y, y_t).$$

We rewrite the last three terms above as follows.

$$\frac{\tau}{S}U_\psi(y, y_{t+1}) - \frac{\tau}{n}U_\psi(y, \bar{y}_{t+1}) + \frac{(S-n)\tau}{nS}U_\psi(y, y_t)$$

$$= \frac{\tau}{S}\sum_{i=1}^n (\psi_i(y^{(i)}) - \psi_i(y_{t+1}^{(i)}) - (y^{(i)} - y_{t+1}^{(i)})^\top \nabla\psi_i(y_{t+1}^{(i)})) - \frac{\tau}{n}\sum_{i=1}^n (\psi_i(y^{(i)}) - \psi_i(\bar{y}_{t+1}^{(i)}) - (y^{(i)} - \bar{y}_{t+1}^{(i)})^\top \nabla\psi_i(\bar{y}_{t+1}^{(i)}))$$

$$+ \frac{(S-n)\tau}{nS}\sum_{i=1}^n (\psi_i(y^{(i)}) - \psi_i(y_t^{(i)}) - (y^{(i)} - y_t^{(i)})^\top \nabla\psi_i(y_t^{(i)}))$$

$$= \frac{\tau}{n}\sum_{i=1}^n \left(\psi_i(\bar{y}_{t+1}^{(i)}) - \frac{n}{S}\psi_i(y_{t+1}^{(i)}) + \frac{n-S}{S}\psi_i(y_t^{(i)})\right) + \underbrace{\frac{\tau}{n}\sum_{i=1}^n (-\frac{n}{S}\nabla\psi_i(y_{t+1}^{(i)}) + \nabla\psi_i(\bar{y}_{t+1}^{(i)}) + \frac{n-S}{S}\nabla\psi_i(y_t^{(i)}))^\top y^{(i)}}_{\sharp}$$

$$+ \frac{\tau}{S}\sum_{i=1}^n \left\langle \nabla\psi_i(y_{t+1}^{(i)}), y_{t+1}^{(i)}\right\rangle - \frac{\tau}{n}\sum_{i=1}^n \left\langle \nabla\psi_i(\bar{y}_{t+1}^{(i)}), \bar{y}_{t+1}^{(i)}\right\rangle + \frac{(S-n)\tau}{nS}\sum_{i=1}^n \left\langle \nabla\psi_i(y_t^{(i)}), y_t^{(i)}\right\rangle.$$

Note that both $\bar{y}_{t+1}^{(i)}$ and $y_t^{(i)}$ are independent of $\mathcal{S}_t$ such that

$$\mathbb{E}[\psi_i(y_{t+1}^{(i)}) \mid \mathcal{G}_t] = \frac{S}{n}\psi_i(\bar{y}_{t+1}^{(i)}) + \frac{n-S}{n}\psi_i(y_t^{(i)}),$$

$$\mathbb{E}\left[\left\langle \nabla\psi_i(y_{t+1}^{(i)}), y_{t+1}^{(i)}\right\rangle \mid \mathcal{G}_t\right] = \frac{S}{n}\left\langle \nabla\psi_i(\bar{y}_{t+1}^{(i)}), \bar{y}_{t+1}^{(i)}\right\rangle + \frac{n-S}{n}\left\langle \nabla\psi_i(y_t^{(i)}), y_t^{(i)}\right\rangle,$$

$$\mathbb{E}\left[\nabla\psi_i(y_{t+1}^{(i)}) \mid \mathcal{G}_t\right] = \frac{S}{n}\nabla\psi_i(\bar{y}_{t+1}^{(i)}) + \frac{n-S}{n}\nabla\psi_i(y_t^{(i)}).$$

Apply Lemma 2 to $\sharp$ with $\Delta_t^{(i)} := -\frac{n}{S}\nabla\psi_i(y_{t+1}^{(i)}) + \nabla\psi_i(\bar{y}_{t+1}^{(i)}) + \frac{n-S}{S}\nabla\psi_i(y_t^{(i)})$, $\hat{y}_{t+1}^{(i)} = \arg\min_v\{-v^\top\Delta_t^{(i)} + \alpha U_{\psi_i}(v, \hat{y}_t^{(i)})\}$ ($\alpha$ to be determined) such that

$$\mathbb{E}\left[\left\langle \Delta_t^{(i)}, y^{(i)}\right\rangle\right] \leq \mathbb{E}\left[\alpha U_{\psi_i}(y^{(i)}, \hat{y}_t^{(i)}) - \alpha U_{\psi_i}(y^{(i)}, \hat{y}_{t+1}^{(i)})\right] + \frac{1}{2\mu_\psi\alpha}\mathbb{E}\left[\left\|\Delta_t^{(i)}\right\|_*^2\right].$$

Sum both sides from 1 to $n$ and divide $n$ on both sides

$$\mathbb{E}[\sharp] \leq \mathbb{E}\left[\frac{\alpha\tau}{n}(U_\psi(y, \hat{y}_t) - U_\psi(y, \hat{y}_{t+1}))\right] + \frac{\tau}{2n\mu_\psi\alpha}\mathbb{E}\left[\sum_{i=1}^n \left\|\Delta_t^{(i)}\right\|_*^2\right].$$

Note that $\mathbb{E}[(\nabla\psi_i(y_{t+1}^{(i)}) - \nabla\psi_i(y_t^{(i)})) \mid \mathcal{G}_t] = \frac{S}{n}(\nabla\psi_i(\bar{y}_{t+1}^{(i)}) - \nabla\psi_i(y_t^{(i)}))$ such that

$$\mathbb{E}\left[\|\Delta_t^{(i)}\|_*^2\right] = \mathbb{E}\left\|(\nabla\psi_i(\bar{y}_{t+1}^{(i)}) - \nabla\psi_i(y_t^{(i)})) - \frac{n}{S}(\nabla\psi_i(y_{t+1}^{(i)}) - \nabla\psi_i(y_t^{(i)}))\right\|_*^2 \leq \frac{n^2}{S^2}\mathbb{E}\left\|\nabla\psi_i(y_{t+1}^{(i)}) - \nabla\psi_i(y_t^{(i)})\right\|_*^2.$$

Thus, we have

$$\mathbb{E}[\sharp] \leq \mathbb{E}\left[\frac{\alpha\tau}{n}(U_\psi(y, \hat{y}_t) - U_\psi(y, \hat{y}_{t+1}))\right] + \frac{\tau n}{2\mu_\psi\alpha S^2}\sum_{i=1}^n \mathbb{E}\left\|\nabla\psi_i(y_{t+1}^{(i)}) - \nabla\psi_i(y_t^{(i)})\right\|_*^2. \tag{32}$$

The last term above can be upper bounded as

$$\frac{\tau n}{2\mu_\psi\alpha S^2}\sum_{i=1}^n \mathbb{E}\left\|\nabla\psi_i(y_{t+1}^{(i)}) - \nabla\psi_i(y_t^{(i)})\right\|_*^2 \leq \frac{\tau n L_\psi^2}{2\mu_\psi\alpha S^2}\sum_{i=1}^n \mathbb{E}\left\|y_{t+1}^{(i)} - y_t^{(i)}\right\|^2.$$

Choose $\alpha = \frac{n\lambda_1}{S}$ for some $\lambda_1 > 0$. According to (31) and (32) and $\mathbb{E}[\|y_{t+1} - y_t\|^2 \mid \mathcal{G}_t] = \frac{S}{n}\|\bar{y}_{t+1} - y_t\|^2 \leq \frac{2S}{n\mu_\psi}U_\psi(\bar{y}_{t+1}, y_t)$, we can finish the proof. $\qquad\square$

### D.2.1. A SUPPORTING LEMMA

**Lemma 10.** *Under Assumptions 2 and 4, the following holds for Algorithm 1 with $\theta = 1$ and any $\lambda_2, \lambda_3, \lambda_4, \lambda_5 > 0$, $y \in \mathcal{Y}$.*

$$\frac{1}{n}\sum_{i=1}^{n}\mathbb{E}(g_i(x_{t+1}) - \tilde{g}_t^{(i)})^\top(y^{(i)} - \bar{y}_{t+1}^{(i)}) \tag{33}$$

$$= \mathbb{E}[\Gamma_{t+1} - \Gamma_t] + \frac{2\lambda_2}{n}\mathbb{E}[U_\psi(y, \hat{y}_t) - U_\psi(y, \hat{y}_{t+1})] + \frac{\lambda_5}{n}\mathbb{E}[U_\psi(y, \check{y}_t) - U_\psi(y, \check{y}_{t+1})]$$

$$+ \frac{(\lambda_3 + \lambda_4)\mathbb{E}[U_\psi(\bar{y}_{t+1}, y_t)]}{\mu_\psi n} + \frac{C_g^2\mathbb{E}\|x_{t+1} - x_t\|^2}{2\lambda_3} + \frac{C_g^2\mathbb{E}\|x_t - x_{t-1}\|^2}{2\lambda_4} + \frac{9\sigma_0^2}{\tau\mu_\psi B} + \frac{\sigma_0^2}{\lambda_2\mu_\psi B} + \frac{\sigma_0^2}{2\lambda_5\mu_\psi B}.$$

*where $\Gamma_t := \frac{1}{n}\sum_{i=1}^{n}(g_i(x_t) - g_i(x_{t-1}))^\top(y^{(i)} - y_t^{(i)})$, $\{\hat{y}_t\}_{t\geq 0}$, $\{\check{y}_t\}_{t\geq 0}$ are virtual sequences constructed as $\hat{y}_{t+1}^{(i)} = \arg\min_{v\in\mathcal{Y}_i}\{v^\top(g_i(x_t;\mathcal{B}_t^{(i)}) - g_i(x_t)) + \lambda_2 U_{\psi_i}(v, \hat{y}_t^{(i)})\}$, $\check{y}_{t+1}^{(i)} = \arg\min_{v\in\mathcal{Y}_i}\{v^\top(g_i(x_{t-1};\mathcal{B}_t^{(i)}) - g_i(x_{t-1})) + \lambda_2 U_{\psi_i}(v, \check{y}_t^{(i)})\}$ for each $i \in [n]$.*

*Proof.* The term $\frac{1}{n}\sum_{i=1}^{n}(g_i(x_{t+1}) - \tilde{g}_t^{(i)})^\top(y^{(i)} - \bar{y}_{t+1}^{(i)})$ can be decomposed as

$$\frac{1}{n}\sum_{i=1}^{n}(g_i(x_{t+1}) - \tilde{g}_t^{(i)})^\top(y^{(i)} - \bar{y}_{t+1}^{(i)}) \tag{34}$$

$$= \underbrace{\frac{1+\theta}{n}\sum_{i=1}^{n}(g_i(x_t) - g_i(x_t;\mathcal{B}_t^{(i)}))^\top(y^{(i)} - \bar{y}_{t+1}^{(i)})}_{\text{I}} + \underbrace{\frac{1}{n}\sum_{i=1}^{n}g_i(x_{t+1})^\top(y^{(i)} - \bar{y}_{t+1}^{(i)}) - \frac{1}{n}\sum_{i=1}^{n}g_i(x_t)^\top(y^{(i)} - \bar{y}_{t+1}^{(i)})}_{\text{II}}$$

$$+ \underbrace{\frac{\theta}{n}\sum_{i=1}^{n}(g_i(x_{t-1}) - g_i(x_t))^\top(y^{(i)} - \bar{y}_{t+1}^{(i)}) + \frac{\theta}{n}\sum_{i=1}^{n}(g_i(x_{t-1};\mathcal{B}_t^{(i)}) - g_i(x_{t-1}))^\top(y^{(i)} - \bar{y}_{t+1}^{(i)})}_{\text{IV}}.$$

Note that our ALEXR (Algorithm 1) only samples $\mathcal{B}_t^{(i)}$ for those $i \in \mathcal{S}_t$ in the $t$-th iteration. For those $i \notin \mathcal{S}_t$, the batches $\{\mathcal{B}_t^{(i)}\}_{i\notin\mathcal{S}_t}$ are virtual and introduced solely for the convenience of analysis, which are not required in the actual execution of Algorithm 1. For each $i \in [n]$, define $\dot{y}_{t+1}^{(i)} := \arg\max_{v\in\mathcal{Y}_i}\{v^\top\bar{g}_t^{(i)} - f_i^*(v) - \tau U_{\psi_i}(v, y_t^{(i)})\}$ and $\bar{g}_t^{(i)} := g_i(x_t) + \theta(g_i(x_t) - g_i(x_{t-1}))$. We decompose the I term in (34) as

$$\text{I} = \frac{1+\theta}{n}\sum_{i=1}^{n}(g_i(x_t) - g_i(x_t;\mathcal{B}_t^{(i)}))^\top(y^{(i)} - \bar{y}_{t+1}^{(i)})$$

$$= \frac{1+\theta}{n}\sum_{i=1}^{n}\left\{(g_i(x_t) - g_i(x_t;\mathcal{B}_t^{(i)}))^\top(\dot{y}_{t+1}^{(i)} - \bar{y}_{t+1}^{(i)}) + (g_i(x_t) - g_i(x_t;\mathcal{B}_t^{(i)}))^\top y^{(i)} - (g_i(x_t) - g_i(x_t;\mathcal{B}_t^{(i)}))^\top\dot{y}_{t+1}^{(i)}\right\}.$$

Since $f_i^* + \tau U_{\psi_i}(y^{(i)}, y_t^{(i)})$ is $\tau\mu_\psi$-strongly convex, Lemma 3 implies that

$$\frac{1}{n}\mathbb{E}\sum_{i=1}^{n}(g_i(x_t) - g_i(x_t;\mathcal{B}_t^{(i)}))^\top(\dot{y}_{t+1}^{(i)} - \bar{y}_{t+1}^{(i)}) \leq \frac{1}{n}\sum_{i=1}^{n}\mathbb{E}\|g_i(x_t) - g_i(x_t;\mathcal{B}_t^{(i)})\|_*\|\dot{y}_{t+1} - \bar{y}_{t+1}\|$$

$$\leq \frac{1}{n\tau\mu_\psi}\sum_{i=1}^{n}\mathbb{E}\left[\|g_i(x_t) - g_i(x_t;\mathcal{B}_t^{(i)})\|_*((1+\theta)\|g_i(x_t) - g_i(x_t;\mathcal{B}_t^{(i)})\|_* + \theta\|g_i(x_{t-1}) - g_i(x_{t-1};\mathcal{B}_t^{(i)})\|_*)\right]$$

$$\leq \frac{1}{n\tau\mu_\psi}\sum_{i=1}^{n}\mathbb{E}\left[(1+1.5\theta)\|g_i(x_t) - g_i(x_t;\mathcal{B}_t^{(i)})\|_*^2 + 0.5\theta\|g_i(x_{t-1}) - g_i(x_{t-1};\mathcal{B}_t^{(i)})\|_*^2\right] \leq \frac{(1+2\theta)\sigma_0^2}{\tau B\mu_\psi}.$$

Apply Lemma 2 to the term $\frac{1}{n}\sum_{i=1}^{n}(g_i(x_t) - g_i(x_t;\mathcal{B}_t^{(i)}))^\top y^{(i)}$. For any $\lambda_2 > 0$ and the auxiliary sequence $\{\hat{y}_t\}_{t\geq 0}$ constructed as $\hat{y}_{t+1}^{(i)} = \arg\min_{v\in\mathcal{Y}_i}\{v^\top(g_i(x_t;\mathcal{B}_t^{(i)}) - g_i(x_t)) + \lambda_2 U_{\psi_i}(v, \hat{y}_t^{(i)})\}$ for each $i \in [n]$, we have

$$\frac{1}{n}\sum_{i=1}^{n}(g_i(x_t) - g_i(x_t;\mathcal{B}_t^{(i)}))^\top y^{(i)} \leq \frac{\lambda_2}{n}\mathbb{E}[U_\psi(y, \hat{y}_t) - U_\psi(y, \hat{y}_{t+1})] + \frac{1}{2\lambda_2\mu_\psi n}\mathbb{E}\|g(x_t) - g(x_t;\mathcal{B}_t^{(i)})\|_*^2.$$

Lastly, $\mathbb{E}[(g_i(x_t) - g_i(x_t; \mathcal{B}_t^{(i)}))^\top \dot{y}_{t+1}^{(i)} \mid \mathcal{F}_{t-1}] = 0$. Choose $\theta = 1$. Then, the I term in (34) can be bounded as

$$\mathbb{E}[\mathrm{I}] \leq \frac{2\lambda_2}{n} \mathbb{E}[U_\psi(y, \hat{y}_t) - U_\psi(y, \hat{y}_{t+1})] + \frac{\sigma_0^2}{\lambda_2 \mu_\psi B} + \frac{6\sigma_0^2}{\tau \mu_\psi B}. \tag{35}$$

Define $\Gamma_t := \frac{1}{n} \sum_{i=1}^n (g_i(x_t) - g_i(x_{t-1}))^\top (y^{(i)} - y_t^{(i)})$. For any $\lambda_3, \lambda_4 > 0$, II + III can be rewritten as

$$\mathrm{II + III} = \frac{1}{n} \sum_{i=1}^n g_i(x_{t+1})^\top (y^{(i)} - \bar{y}_{t+1}^{(i)}) - \frac{1}{n} \sum_{i=1}^n g_i(x_t)^\top (y^{(i)} - \bar{y}_{t+1}^{(i)}) + \frac{1}{n} \sum_{i=1}^n (g_i(x_{t-1}) - g_i(x_t))^\top (y^{(i)} - \bar{y}_{t+1}^{(i)})$$

$$= \Gamma_{t+1} - \Gamma_t + \frac{1}{n} \sum_{i=1}^n (g_i(x_{t+1}) - g_i(x_t))^\top (y_{t+1}^{(i)} - \bar{y}_{t+1}^{(i)}) + \frac{1}{n} \sum_{i=1}^n (g_i(x_{t-1}) - g_i(x_t))^\top (y_t^{(i)} - \bar{y}_{t+1}^{(i)})$$

$$\leq \Gamma_{t+1} - \Gamma_t + \frac{C_g^2 \|x_{t+1} - x_t\|_2^2}{2\lambda_3} + \frac{\lambda_3 \|y_{t+1} - \bar{y}_{t+1}\|^2}{2n} + \frac{C_g^2 \|x_t - x_{t-1}\|_2^2}{2\lambda_4} + \frac{\lambda_4 \|y_t - \bar{y}_{t+1}\|^2}{2n}$$

Note that $y_{t+1}^{(i)} = \bar{y}_{t+1}^{(i)}$ if $i \in \mathcal{S}_t$ and $y_{t+1}^{(i)} = y_t^{(i)}$ otherwise. Then, $\|y_{t+1} - \bar{y}_{t+1}\|^2 \leq \|y_t - \bar{y}_{t+1}\|^2$ such that

$$\mathrm{II + III} \leq \Gamma_{t+1} - \Gamma_t + \frac{C_g^2 \|x_{t+1} - x_t\|_2^2}{2\lambda_3} + \frac{C_g^2 \|x_t - x_{t-1}\|_2^2}{2\lambda_4} + \frac{(\lambda_3 + \lambda_4) U_\psi(\bar{y}_{t+1}, y_t)}{\mu_\psi n}. \tag{36}$$

We decompose the IV term in (34) as

$$\mathrm{IV} = \frac{1}{n} \sum_{i=1}^n (g_i(x_{t-1}; \mathcal{B}_t^{(i)}) - g_i(x_{t-1}))^\top (y^{(i)} - \bar{y}_{t+1}^{(i)})$$

$$= \frac{1}{n} \sum_{i=1}^n \left\{ (g_i(x_{t-1}; \mathcal{B}_t^{(i)}) - g_i(x_{t-1}))^\top (\dot{y}_{t+1}^{(i)} - \bar{y}_{t+1}^{(i)}) \right.$$

$$\left. + (g_i(x_{t-1}; \mathcal{B}_t^{(i)}) - g_i(x_{t-1}))^\top y^{(i)} - (g_i(x_{t-1}; \mathcal{B}_t^{(i)}) - g_i(x_{t-1}))^\top \dot{y}_{t+1}^{(i)} \right\}.$$

By the Cauchy-Schwarz inequality, we have

$$\frac{1}{n} \sum_{i=1}^n \mathbb{E}\left[ (g_i(x_{t-1}; \mathcal{B}_t^{(i)}) - g_i(x_{t-1}))^\top (\dot{y}_{t+1}^{(i)} - \bar{y}_{t+1}^{(i)}) \right] \leq \frac{1}{n} \sum_{i=1}^n \mathbb{E}\left[ \|g_i(x_{t-1}; \mathcal{B}_t^{(i)}) - g_i(x_{t-1})\|_* \|\dot{y}_{t+1}^{(i)} - \bar{y}_{t+1}^{(i)}\| \right].$$

Since $f_i^*(y^{(i)}) + \tau U_{\psi_i}(y^{(i)}, y_t^{(i)})$ is $\tau\mu_\psi$-strongly convex to $y^{(i)}$, Lemma 3 implies that

$$\|\dot{y}_{t+1}^{(i)} - \bar{y}_{t+1}^{(i)}\| \leq \frac{(1+\theta)\|g_i(x_t) - g_i(x_t; \mathcal{B}_t^{(i)})\|_* + \theta\|g_i(x_{t-1}) - g_i(x_{t-1}; \mathcal{B}_t^{(i)})\|_*}{\tau\mu_\psi}.$$

Similar to (35), the following holds for any $\lambda_5 > 0$ and the auxiliary sequence $\{\breve{y}_t\}_{t\geq 0}$ that is constructed as $\breve{y}_{t+1}^{(i)} = \arg\min_{v \in \mathcal{Y}_i}\{v^\top (g_i(x_{t-1}; \mathcal{B}_t^{(i)}) - g_i(x_{t-1})) + \lambda_2 U_{\psi_i}(v, \breve{y}_t^{(i)})\}$ for each $i \in [n]$.

$$\frac{1}{n} \sum_{i=1}^n \mathbb{E}\left[ (g_i(x_{t-1}; \mathcal{B}_t^{(i)}) - g_i(x_{t-1}))^\top y^{(i)} \right] \leq \frac{\lambda_5}{n} \mathbb{E}[U_\psi(y, \breve{y}_t) - U_\psi(y, \breve{y}_{t+1})] + \frac{\sigma_0^2}{2\lambda_5 \mu_\psi B}.$$

Consider that $\frac{1}{n} \sum_{i=1}^n \mathbb{E}[(g_i(x_{t-1}; \mathcal{B}_t^{(i)}) - g_i(x_{t-1}))^\top \dot{y}_{t+1}^{(i)}] = 0$.

$$\mathbb{E}[\mathrm{IV}] \leq \frac{\lambda_5}{n} \mathbb{E}[U_\psi(y, \hat{y}_t) - U_\psi(y, \hat{y}_{t+1})] + \frac{\sigma_0^2}{2\lambda_5 \mu_\psi B} + \frac{3\sigma_0^2}{\tau \mu_\psi B}. \tag{37}$$

Combine (35), (36), and (37).

$$\frac{1}{n} \sum_{i=1}^n \mathbb{E}(g_i(x_{t+1}) - \tilde{g}_t^{(i)})^\top (y^{(i)} - \bar{y}_{t+1}^{(i)})$$

$$\leq \mathbb{E}[\Gamma_{t+1} - \Gamma_t] + \frac{2\lambda_2}{n} \mathbb{E}[U_\psi(y, \hat{y}_t) - U_\psi(y, \hat{y}_{t+1})] + \frac{\lambda_5}{n} \mathbb{E}[U_\psi(y, \breve{y}_t) - U_\psi(y, \breve{y}_{t+1})]$$

$$+ \frac{(\lambda_3 + \lambda_4) \mathbb{E}[U_\psi(\bar{y}_{t+1}, y_t)]}{\mu_\psi n} + \frac{C_g^2 \mathbb{E}\|x_{t+1} - x_t\|^2}{2\lambda_3} + \frac{C_g^2 \mathbb{E}\|x_t - x_{t-1}\|^2}{2\lambda_4} + \frac{9\sigma_0^2}{\tau\mu_\psi B} + \frac{\sigma_0^2}{\lambda_2\mu_\psi B} + \frac{\sigma_0^2}{2\lambda_5\mu_\psi B}.$$

$\square$

### D.2.2. PROOF OF THEOREM 3

*Proof.* If $g_i$ is smooth, we combine (15), (21), (30), (33). Set $x = x_*$ and $x_0 = x_{-1}$.

$$
\mathbb{E}[L(x_{t+1}, y_{t+1}) - L(x_*, \bar{y}_{t+1})]
$$

$$
\leq \frac{\tau}{S}\mathbb{E}[U_\psi(y, y_t) - U_\psi(y, y_{t+1})] + \frac{\tau\lambda_1}{S}\mathbb{E}[U_\psi(y, \hat{y}_t) - U_\psi(y, \hat{y}_{t+1})] + \frac{\eta}{2}\mathbb{E}\|x_* - x_t\|_2^2 - \frac{\eta}{2}\mathbb{E}\|x_* - x_{t+1}\|_2^2
$$

$$
+ \mathbb{E}[\Gamma_{t+1} - \Gamma_t] + \frac{2\lambda_2}{n}\mathbb{E}[U_\psi(y, \hat{\hat{y}}_t) - U_\psi(y, \hat{\hat{y}}_{t+1})] + \frac{\lambda_5}{n}\mathbb{E}[U_\psi(y, \check{y}_t) - U_\psi(y, \check{y}_{t+1})]
$$

$$
- \left(\frac{\tau}{n} - \frac{\tau L_\psi^2}{n\lambda_1\mu_\psi^2 S} - \frac{\lambda_3 + \lambda_4}{\mu_\psi n}\right)\mathbb{E}[U_\psi(\bar{y}_{t+1}, y_t)] - \left(\frac{\eta}{2} - \frac{C_g^2}{2\lambda_3} - \frac{L_g C_f}{2}\right)\mathbb{E}\|x_{t+1} - x_t\|_2^2 + \frac{C_g^2}{2\lambda_4}\mathbb{E}\|x_t - x_{t-1}\|_2^2
$$

$$
+ \frac{9\sigma_0^2}{\tau\mu_\psi B} + \frac{\sigma_0^2}{\lambda_2\mu_\psi B} + \frac{\sigma_0^2}{2\lambda_5\mu_\psi B} + \frac{C_f^2\sigma_1^2}{\eta B} + \frac{\delta^2}{\eta S}. \tag{38}
$$

Do telescoping sum from $t = 0$ to $T - 1$ for the equation above.

$$
\sum_{t=0}^{T-1}\mathbb{E}[L(x_{t+1}, y) - L(x_*, \bar{y}_{t+1})]
$$

$$
\leq \frac{\eta\mathbb{E}\|x_* - x_0\|_2^2}{2} + \frac{\tau}{S}\mathbb{E}[U_\psi(y, y_0)] + \frac{\tau\lambda_1}{S}\mathbb{E}[U_\psi(y, \hat{y}_0)] + \frac{2\lambda_2}{n}\mathbb{E}[U_\psi(y, \hat{\hat{y}}_0)] + \frac{\lambda_5}{n}\mathbb{E}[U_\psi(y, \check{y}_0)]
$$

$$
- \left(\frac{\tau}{n} - \frac{\tau L_\psi^2}{n\lambda_1\mu_\psi^2 S} - \frac{\lambda_3 + \lambda_4}{\mu_\psi n}\right)\sum_{t=0}^{T-1}\mathbb{E}[U_\psi(\bar{y}_{t+1}, y_t)] - \left(\frac{\eta}{2} - \frac{L_g C_f}{2} - \frac{C_g^2}{2\lambda_3} - \frac{C_g^2}{2\lambda_4}\right)\sum_{t=0}^{T-1}\mathbb{E}\|x_{t+1} - x_t\|^2
$$

$$
+ \mathbb{E}[\Gamma_T] - \frac{\tau}{S}\mathbb{E}[U_\psi(y, y_T)] + \left(\frac{C_f^2\sigma_1^2}{B\eta} + \frac{\delta^2}{S\eta}\right)T + \frac{9\sigma_0^2 T}{\tau\mu_\psi B} + \frac{\sigma_0^2 T}{\lambda_2\mu_\psi B} + \frac{\sigma_0^2 T}{2\lambda_5\mu_\psi B}.
$$

Note that $\Gamma_0 = 0$, $\Gamma_T \leq \frac{1}{n}\sum_{i=1}^n \|g_i(x_T) - g_i(x_{T-1})\|_* \left\|y^{(i)} - y_T^{(i)}\right\| \leq \frac{C_g^2}{2\lambda_3}\|x_T - x_{T-1}\|_2^2 + \frac{\lambda_3}{2n\mu_\psi}U_\psi(y, y_T)$. Choose $\lambda_1 \asymp \frac{L_\psi^2}{S\mu_\psi^2}$, $\lambda_2 \asymp \frac{n\tau}{S}$, $\lambda_3 \asymp \frac{C_g\sqrt{S}}{\sqrt{n}}$, $\lambda_4 \asymp \frac{C_g\sqrt{S}}{\sqrt{n}}$, $\lambda_5 \asymp \frac{n\tau}{S}$, and let $1/\tau = O\left(\frac{\sqrt{n}\mu_\psi}{C_g\sqrt{S}}\right)$ and $1/\eta = O\left(\frac{\sqrt{S}}{C_g\sqrt{n}}\right)$. Since $L(x, y)$ is convex in $x$ and linear in $y$, we have

$$
\mathbb{E}\max_y[L(\bar{x}_T, y) - L(x_*, \bar{\bar{y}}_T)] \leq \mathbb{E}\max_y \frac{1}{T}\sum_{t=0}^{T-1}[L(x_{t+1}, y) - L(x_*, \bar{y}_{t+1})], \tag{39}
$$

where $\bar{x}_T = \frac{1}{T}\sum_{t=0}^{T-1} x_{t+1}$, $\bar{\bar{y}}_T = \frac{1}{T}\sum_{t=0}^{T-1} \bar{y}_{t+1}$. Now work on the LHS.

$$
L(\bar{x}_T, y) - L(x_*, \bar{\bar{y}}_T) = \frac{1}{n}\sum_{i=1}^n \left(y^{(i)}g_i(\bar{x}_T) - f_i^*(y^{(i)})\right) + r(\bar{x}_T) - \frac{1}{n}\sum_{i=1}^n \left(\bar{\bar{y}}_T^{(i)}g_i(x_*) - f_i^*(\bar{\bar{y}}_T^{(i)})\right) - r(x_*)
$$

Choose $y^{(i)} = \tilde{y}_T^{(i)} \in \arg\max_v\{vg_i(\bar{x}_T) - f_i^*(v)\} \Leftrightarrow g_i(\bar{x}_T) \in \partial f_i^*(\tilde{y}_T^{(i)}) \Leftrightarrow \tilde{y}_T^{(i)} \in \partial f_i(g_i(\bar{x}_T))$ such that $\tilde{y}_T^{(i)}g_i(\bar{x}_T) - f_i^*(\tilde{y}_T^{(i)}) = f_i(g_i(\bar{x}_T))$. By Fenchel-Young, $-\bar{\bar{y}}_T^{(i)}g_i(x_*) + f_i^*(\bar{\bar{y}}_T^{(i)}) \geq -f_i(g_i(x_*))$. Thus, $\mathbb{E}[F(\bar{x}_T) - F(x_*)] \leq \mathbb{E}\max_y[L(\bar{x}_T, y) - L(x_*, \bar{\bar{y}}_T)]$. Thus, we can make $\mathbb{E}[F(\bar{x}_T) - F(x_*)] \leq \epsilon$ after $T = O\left(\frac{L_g C_f D_\mathcal{X}^2}{\epsilon} + \frac{\sqrt{n}C_g D_\mathcal{X}^2}{\sqrt{S}\epsilon} + \frac{C_g(1+L_\psi^2/(S\mu_\psi^2))D_\mathcal{Y}^2}{\mu_\psi\sqrt{n}S\epsilon} + \frac{D_\mathcal{X}^2\delta^2}{S\epsilon^2} + \frac{D_\mathcal{X}^2 C_f^2\sigma_1^2}{B\epsilon^2} + \frac{\sigma_0^2(1+L_\psi^2/(S\mu_\psi^2))D_\mathcal{Y}^2}{\mu_\psi BS\epsilon^2}\right)$ iterations by setting $\theta = 1$, $\tau = O\left(\frac{\sqrt{S}C_g}{\mu_\psi\sqrt{n}} \vee \frac{\sigma_0^2}{\mu_\psi B\epsilon}\right)$, $\eta = O\left(L_g C_f \vee \frac{\sqrt{n}C_g}{\sqrt{S}} \vee \frac{\delta^2}{S\epsilon} \vee \frac{C_f^2\sigma_1^2}{B\epsilon}\right)$. $\square$

### D.2.3. PROOF OF THEOREM 2

*Proof.* If $g_i$ is non-smooth, we can use ALEXR with $\theta = 0$, where $\tilde{g}_t^{(i)} = g_i(x_t; \mathcal{B}_t^{(i)})$. Then, for any $\lambda > 0$ we have

$$\frac{1}{n} \sum_{i=1}^{n} \mathbb{E} \left\langle g_i(x_{t+1}) - \tilde{g}_t, y^{(i)} - \bar{y}_{t+1}^{(i)} \right\rangle = \frac{1}{n} \sum_{i=1}^{n} \mathbb{E} \left\langle g_i(x_{t+1}) - g_i(x_t; \mathcal{B}_t^{(i)}), y^{(i)} - \bar{y}_{t+1}^{(i)} \right\rangle$$

$$= \frac{1}{n} \sum_{i=1}^{n} \mathbb{E} \left[ \left\langle g_i(x_{t+1}) - g_i(x_t), y^{(i)} - \bar{y}_{t+1}^{(i)} \right\rangle \right] + \frac{1}{n} \sum_{i=1}^{n} \mathbb{E} \left[ \left\langle g_i(x_t) - g_i(x_t; \mathcal{B}_t^{(i)}), y^{(i)} - \bar{y}_{t+1}^{(i)} \right\rangle \right]$$

$$\leq \frac{1}{n} \sum_{i=1}^{n} \mathbb{E} \left[ \|g_i(x_{t+1}) - g_i(x_t)\|_* \left\| y^{(i)} - \bar{y}_{t+1}^{(i)} \right\| \right] + \frac{1}{n} \sum_{i=1}^{n} \mathbb{E} \left[ \left\langle g_i(x_t) - g_i(x_t; \mathcal{B}_t^{(i)}), y^{(i)} - \bar{y}_{t+1}^{(i)} \right\rangle \right]$$

$$\leq \frac{C_g^2 \mathbb{E} \|x_{t+1} - x_t\|_2^2}{2\lambda_4} + 2\lambda_4 \frac{1}{n} \sum_{i=1}^{n} \mathbb{E}[\|y^{(i)}\|^2 + \|\bar{y}_{t+1}^{(i)}\|^2] + \frac{1}{n} \sum_{i=1}^{n} \mathbb{E} \left[ \left\langle g_i(x_t) - g_i(x_t; \mathcal{B}_t^{(i)}), y^{(i)} - \bar{y}_{t+1}^{(i)} \right\rangle \right]$$

$$\leq \frac{C_g^2 \mathbb{E} \left[ \|x_{t+1} - x_t\|_2^2 \right]}{2\lambda_4} + 4\lambda_4 C_f^2 + \frac{1}{n} \sum_{i=1}^{n} \mathbb{E} \left[ \left\langle g_i(x_t) - g_i(x_t; \mathcal{B}_t^{(i)}), y^{(i)} - \bar{y}_{t+1}^{(i)} \right\rangle \right].$$

The last term above can be decomposed as

$$\frac{1}{n} \sum_{i=1}^{n} \mathbb{E} \left[ \left\langle g_i(x_t) - g_i\left(x_t; \mathcal{B}_t^{(i)}\right), y^{(i)} - \bar{y}_{t+1}^{(i)} \right\rangle \right]$$

$$= \frac{1}{n} \sum_{i=1}^{n} \mathbb{E}_t \left[ \left\langle g_i(x_t) - g_i\left(x_t; \mathcal{B}_t^{(i)}\right), y^{(i)} - y_t^{(i)} \right\rangle \right] + \frac{1}{n} \sum_{i=1}^{n} \mathbb{E} \left[ \left\langle g_i(x_t) - g_i(x_t; \mathcal{B}_t^{(i)}), y_t^{(i)} - \bar{y}_{t+1}^{(i)} \right\rangle \right]. \quad (40)$$

Note that $\mathbb{E} \left[ \left\langle g_i(x_t) - g_i(x_t; \mathcal{B}_t^{(i)}), y_t^{(i)} \right\rangle \mid \mathcal{F}_{t-1} \right] = 0$. Besides, Lemma (9) implies that for some $\lambda_2 > 0$ and sequence $\{\tilde{y}_t\}_t$

$$\mathbb{E} \left[ \left\langle g_i(x_t) - g_i(x_t; \mathcal{B}_t^{(i)}), y^{(i)} \right\rangle \right] \leq \mathbb{E} \left[ \lambda_2 U_{\psi_i}(y^{(i)}, \tilde{y}_t^{(i)}) - \lambda_2 U_{\psi_i}(y^{(i)}, \tilde{y}_{t+1}^{(i)}) \right] + \frac{1}{2\lambda_2 \mu_\psi} \mathbb{E} \left\| g_i(x_t) - g_i(x_t; \mathcal{B}_t^{(i)}) \right\|_*^2.$$

such that

$$\frac{1}{n} \sum_{i=1}^{n} \mathbb{E} \left[ \left\langle g_i(x_t) - g_i(x_t; \mathcal{B}_t^{(i)}), y^{(i)} \right\rangle \right] \leq \frac{\lambda_2}{n} \mathbb{E} \left[ U_\psi(y, \tilde{y}_t) - U_\psi(y, \tilde{y}_{t+1}) \right] + \frac{\sigma_0^2}{2\lambda_2 B \mu_\psi}.$$

For any $\lambda_3 > 0$, the second term in (40) can be bounded as

$$\frac{1}{n} \sum_{i=1}^{n} \mathbb{E} \left[ \left\langle g_i(x_t) - g_i(x_t; \mathcal{B}_t^{(i)}), y_t^{(i)} - \bar{y}_{t+1}^{(i)} \right\rangle \right] \leq \frac{\lambda_3}{2n} \sum_{i=1}^{n} \mathbb{E} \left[ \left\| g_i(x_t) - g_i(x_t; \mathcal{B}_t^{(i)}) \right\|_*^2 \right] + \frac{\mathbb{E} \left[ \|y_t - \bar{y}_{t+1}\|^2 \right]}{2\lambda_3 n}$$

$$\leq \frac{\lambda_3 \sigma_0^2}{2B} + \frac{\mathbb{E} \left[ U_\psi(\bar{y}_{t+1}, y_t) \right]}{\lambda_3 \mu_\psi n}.$$

Thus, we have

$$\mathbb{E} \left[ \frac{1}{n} \sum_{i=1}^{n} \left\langle g_i(x_{t+1}) - \tilde{g}_t, y^{(i)} - \bar{y}_{t+1}^{(i)} \right\rangle \right] \leq \frac{C_g^2 \mathbb{E} \left[ \|x_{t+1} - x_t\|_2^2 \right]}{2\lambda_4} + 4\lambda_4 C_f^2 + \frac{\lambda_2}{n} \mathbb{E} \left[ U_\psi(y, \tilde{y}_t) - U_\psi(y, \tilde{y}_{t+1}) \right]$$

$$+ \frac{\sigma_0^2}{2B\lambda_2 \mu_\psi} + \frac{\lambda_3 \sigma_0^2}{2B} + \frac{\mathbb{E} \left[ U_\psi(\bar{y}_{t+1}, y_t) \right]}{2\lambda_3 \mu_\psi n}.$$

The other parts are the same as as the proof of Theorem 3.

$\square$

# E. Proof of Theorem 4

We consider a special instance of problem (1) that is separable over the coordinates $i$ and $\mathbb{P}_i = \mathbb{P}$.

$$\min_{x \in [-D,D]^n} F(x), \quad F(x) = \frac{1}{n} \left( \sum_{i=1}^n f(g_i(x)) + \frac{\alpha}{2} \|x\|^2 \right), \tag{41}$$

$$g_i(x) = \mathbb{E}_{\zeta \sim \mathbb{P}}[g_i(x; \zeta)], \ g_i(x; \zeta) = x^{(i)} + \zeta,$$

where the additive noise $\zeta$ follows

$$\zeta = \begin{cases} -\nu & \text{w.p. } 1 - p, \\ \nu(1-p)/p & \text{w.p. } p. \end{cases}, \quad \text{where } p := \frac{\nu^2}{\sigma^2} \in (0, 1).$$

We construct the hard problems for (i) smooth $f_i$; and (ii) non-smooth $f_i$ separately.

**(i) Smooth $f_i$:** First, we can consider the special instance that $f_i$ is the identity mapping and $\delta = 0$ (e.g., all $f_i$'s are identical), $\sigma_0 = 0$. Then, the cFCCO problem in (1) becomes the standard strongly convex minimization problem. Then, we can apply the information-theoretic lower bounds (Agarwal et al., 2009; Nguyen et al., 2019): Any algorithm in the abstract scheme requires at least $\Omega(\frac{1}{\mu\epsilon})$ iterations to find an $\bar{x}$ such that $\frac{\mu}{2}\mathbb{E}\|\bar{x} - x_*\|_2^2 \le \epsilon$.

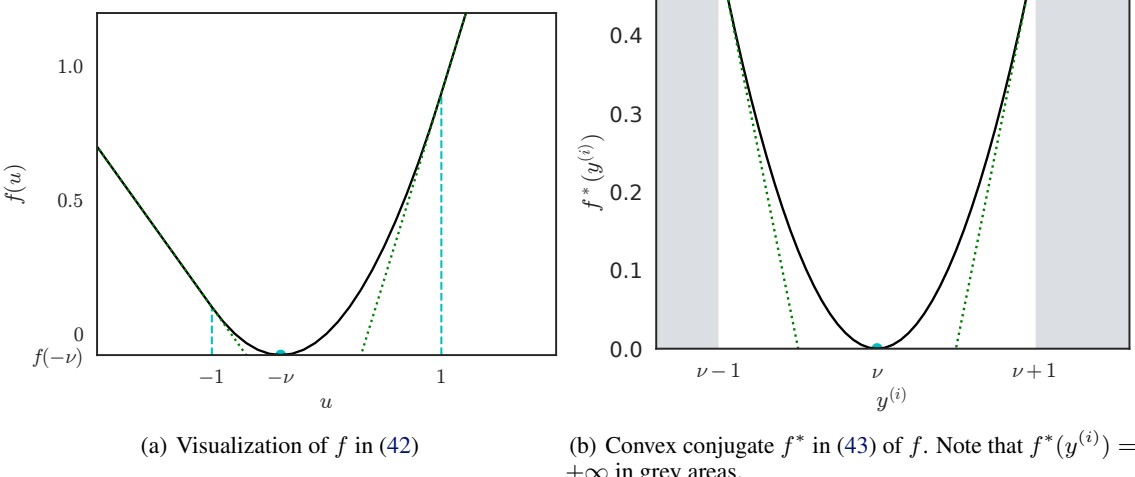

(a) Visualization of $f$ in (42)

(b) Convex conjugate $f^*$ in (43) of $f$. Note that $f^*(y^{(i)}) = +\infty$ in grey areas.

Next, we construct another "hard" instance to derive the second half of the lower bound in this case. Consider the following strongly convex FCCO problem

$$\min_{x \in \mathcal{X}} F(x) = \frac{1}{n} \sum_{i=1}^n f(g_i(x)) + r(x),$$

$$f(u) = \begin{cases} (\nu - 1)u + \frac{1}{2}(\nu - 1)^2 + \nu - 1 - \frac{\nu^2}{2}, & u \in (-\infty, -1) \\ \frac{1}{2}(u + \nu)^2 - \frac{\nu^2}{2}, & u \in [-1, 1] \\ (1 + \nu)u + \frac{1}{2}(1 + \nu)^2 - 1 - \nu - \frac{\nu^2}{2}, & u \in (1, \infty) \end{cases}, \quad r(x) = \frac{1}{4n} \|x\|_2^2 \tag{42}$$

where $\mathcal{X} = [-1, 1]^n$, the outer function $f : \mathbb{R} \to \mathbb{R}$ is smooth and Lipschitz continuous for $\nu < 1$. As stated in Assumption 3, we do not require $f$ to be monotonically non-decreasing when $g_i$ is affine. Choose $\alpha = \frac{1}{2}$ in (41). We define that $F_i(x^{(i)}) := f(g_i(x)) + \frac{1}{4}[x^{(i)}]^2$ such that $F(x) = \frac{1}{n} \sum_{i=1}^n F_i(x^{(i)})$. Thus, the problem $\min_x F(x)$ is equivalent to the problems $\min_{x^{(i)}} F_i(x^{(i)})$ on all coordinates $i \in [n]$. Since the problem is separable over the coordinates, we have $x_*^{(i)} = \arg\min_{x \in [-1,1]} F_i(x^{(i)})$ for $x_* = \arg\min_{x \in \mathcal{X}} F(x)$. Thus, we have $x_*^{(i)} = -\frac{2\nu}{3}$ and $F_i(x_*^{(i)}) = -\frac{\nu^2}{3}$. By the

convex conjugate, for any $y^{(i)} \in \mathbb{R}$ we have

$$
f^*(y^{(i)}) = \max \left\{ \sup_{u < -1} \left\{ uy^{(i)} - \left( (\nu - 1)u + \frac{1}{2}(\nu - 1)^2 + \nu - 1 - \frac{\nu^2}{2} \right) \right\}, \sup_{-1 \le u \le 1} \left\{ uy^{(i)} - \frac{1}{2}(u + \nu)^2 + \frac{\nu^2}{2} \right\}, \right.
$$
$$
\left. \sup_{u > 1} \left\{ uy^{(i)} - \left( (1 + \nu)u + \frac{1}{2}(1 + \nu)^2 - 1 - \nu - \frac{\nu^2}{2} \right) \right\} \right\}
$$
$$
= \begin{cases} +\infty, & y^{(i)} \in (-\infty, \nu - 1) \cup (\nu + 1, \infty) \\ \frac{1}{2}(y^{(i)} - \nu)^2, & y^{(i)} \in [\nu - 1, \nu + 1]. \end{cases} \tag{43}
$$

Since $\mathbb{P}_i = \mathbb{P}$ in the "hard" problem (41), the abstract scheme only needs to sample shared $\zeta_t, \tilde{\zeta}_t \sim \mathbb{P}$ for all coordinates $i \in \mathcal{S}_t$ in the $t$-th iteration. For an $i \in [n]$, suppose that $\mathfrak{g}_\tau^{(i)} = \{0\}$ or $\{-\nu\}, \mathfrak{Y}_\tau^{(i)} = \{0\}, \mathfrak{X}_\tau^{(i)} = \{0\}$ for all $\tau \le t$. Then,

- If $i \notin \mathcal{S}_t$, the abstract scheme leads to

$$
\mathfrak{g}_{t+1}^{(i)} = \{0\} \text{ or } \{-\nu\}, \quad \mathfrak{Y}_{t+1}^{(i)} = \{0\}, \quad \mathfrak{X}_{t+1}^{(i)} = \{0\}.
$$

- If $i \in \mathcal{S}_t$ and $\zeta_t = -\nu$, the abstract scheme proceeds as

$$
\mathfrak{g}_{t+1}^{(i)} = \mathfrak{g}_t^{(i)} + \text{span} \left\{ \hat{x}^{(i)} + \zeta_t \mid \hat{x}^{(i)} \in \mathfrak{X}_t^{(i)} \right\},
$$
$$
\mathfrak{Y}_{t+1}^{(i)} = \mathfrak{Y}_t^{(i)} + \text{span} \left\{ \underset{y^{(i)} \in [\nu - 1, \nu + 1]}{\arg\max} \left\{ y^{(i)}(\hat{g}^{(i)} + \nu) - \frac{1}{2}(y^{(i)})^2 - \tau U_{\psi_i}(y^{(i)}, \hat{y}^{(i)}) \right\} \mid \hat{g}^{(i)} \in \mathfrak{g}_{t+1}^{(i)}, \hat{y}^{(i)} \in \mathfrak{Y}_t^{(i)} \right\},
$$
$$
\mathfrak{X}_{t+1}^{(i)} = \mathfrak{X}_t^{(i)} + \text{span} \left\{ \underset{x^{(i)} \in [-1, 1]}{\arg\min} \left\{ \frac{1}{S}\hat{y}^{(i)}x^{(i)} + \frac{1}{n}(x^{(i)})^2 + \frac{\eta}{2}(x^{(i)} - \hat{x}^{(i)})^2 \right\} \mid \hat{y}^{(i)} \in \mathfrak{Y}_{t+1}^{(i)}, \hat{x}^{(i)} \in \mathfrak{X}_t^{(i)} \right\}.
$$

In this case, we can derive that $\mathfrak{g}_{t+1}^{(i)} = \{-\nu\}$ such that $y^{(i)}(\hat{g}^{(i)} + \nu) = 0$ for $\hat{g}^{(i)} \in \mathfrak{g}_{t+1}^{(i)}, \forall i \in \mathcal{S}_t$. Since $\frac{1}{2}(y^{(i)})^2 + \tau U_{\psi_i}(y^{(i)}, \hat{y}^{(i)})$ is strongly convex to $y^{(i)}$ and non-negative, we have $\mathfrak{Y}_{t+1}^{(i)} = \{0\}$ for $i \in \mathcal{S}_t$. Then, $\mathfrak{X}_{t+1}^{(i)} = \{0\}$ for $i \in \mathcal{S}_t$.

To sum up, given the event $\bigcap_{\tau=1}^t \{\mathfrak{g}_\tau^{(i)} = \{0\} \text{ or } \{-\nu\}, \mathfrak{Y}_\tau^{(i)} = \{0\}, \mathfrak{X}_\tau^{(i)} = \{0\}\}$, we can make sure that $\{\mathfrak{g}_{t+1}^{(i)} = \{0\} \text{ or } \{-\nu\} \wedge \mathfrak{Y}_{t+1}^{(i)} = \{0\} \wedge \mathfrak{X}_{t+1}^{(i)} = \{0\}\}$ when one of the following mutually exclusive events happens:

- Event I: $i \notin \mathcal{S}_t$;
- Event II: $i \in \mathcal{S}_t$ and $\zeta_t = -\nu$.

Note that the random variable $\zeta_t$ is independent of $\mathcal{S}_t$. Thus, the probability of the event $E_{t+1}^{(i)} := \{\mathfrak{g}_{t+1}^{(i)} = \emptyset \text{ or } \{-\nu\} \wedge \mathfrak{Y}_{t+1}^{(i)} = \{0\} \wedge \mathfrak{X}_{t+1}^{(i)} = \{0\}\}$ conditioned on $\bigcap_{\tau=1}^t E_\tau^{(i)}$ can be bounded as

$$
\mathbb{P}\left[ E_{t+1}^{(i)} \mid \bigcap_{\tau=1}^t E_\tau^{(i)} \right] = \mathbb{P}\left[ \left\{ \mathfrak{g}_{t+1}^{(i)} = \{0\} \text{ or } \{-\nu\} \wedge \mathfrak{Y}_{t+1}^{(i)} = \{0\} \wedge \mathfrak{X}_{t+1}^{(i)} = \{0\} \right\} \mid \bigcap_{\tau=1}^t E_\tau^{(i)} \right]
$$
$$
\ge \mathbb{P}\left[ \{i \notin \mathcal{S}_t\} \right] + \mathbb{P}\left[ \{\{i \in \mathcal{S}_t\} \wedge \{\zeta_t = -\nu\}\} \right]
$$
$$
= \mathbb{P}\left[ \{i \notin \mathcal{S}_t\} \right] + \mathbb{P}\left[ \{i \in \mathcal{S}_t\} \right] \mathbb{P}\left[ \{\zeta_t = -\nu\} \right] = \left( 1 - \frac{S}{n} \right) + \frac{S}{n}(1 - p) = 1 - \frac{Sp}{n}.
$$

Since $\mathcal{S}_t$ and $\zeta_t$ in different iterations $t$ are mutually independent, we have

$$
\mathbb{P}\left[ E_T^{(i)} \right] \ge \mathbb{P}\left[ \bigcap_{t=0}^{T-1} E_{t+1}^{(i)} \right] = \prod_{t=0}^{T-1} \mathbb{P}\left[ E_{t+1}^{(i)} \mid \bigcap_{t=1}^t E_t^{(i)} \right] = \left( 1 - \frac{Sp}{n} \right)^T > 3/4 - \frac{TSp}{n}.
$$

Thus, letting $T < \frac{n}{4Sp}$ can make $\mathbb{P}[E_T^{(i)}] > \frac{1}{2}$. Choose $\nu = 3\sqrt{2\epsilon}$, and $\sigma = \sigma_0$ such that $p = \frac{\nu^2}{\sigma^2} = \frac{18\epsilon}{\sigma_0^2}$. For any $i \in [n]$ and any output $\tilde{x}_T^{(i)} \in \mathfrak{X}_T^{(i)}$, we have

$$
\mathbb{E}[(\tilde{x}_T^{(i)} - x_*^{(i)})^2] = \mathbb{E}[\mathbb{I}_{E_T^{(i)}}(\tilde{x}_T^{(i)} - x_*^{(i)})^2 + \mathbb{I}_{\overline{E_T^{(i)}}}(\tilde{x}_T^{(i)} - x_*^{(i)})^2] \ge \mathbb{E}[\mathbb{I}_{E_T^{(i)}}(\tilde{x}_T^{(i)} - x_*^{(i)})^2]
$$
$$
= \mathbb{E}[\mathbb{I}_{E_T^{(i)}}(x_*^{(i)})^2] = \mathbb{P}[E_T^{(i)}](x_*^{(i)})^2 > \frac{2\nu^2}{9} = 4\epsilon.
$$

Since the derivations above hold for arbitrary $i \in [S]$ and the $r(x)$ in (42) is $\frac{1}{2n}$-strongly convex ($\mu = \frac{1}{2n}$), we have

$$\frac{\mu}{2}\mathbb{E}\|\tilde{x}_T^{(i)} - x_*\|_2^2 = \frac{1}{4n}\mathbb{E}\|\tilde{x}_T^{(i)} - x_*\|_2^2 = \frac{1}{4n}\sum_{i=1}^n \mathbb{E}[(\tilde{x}_T^{(i)} - x_*^{(i)})^2] > \epsilon.$$

Thus, to find an output $\tilde{x}_T$ such that $\frac{\mu}{2}\mathbb{E}\|\tilde{x}_T - x_*\|_2^2 \leq \epsilon$, the abstract scheme requires at least $T \geq \frac{n}{4Sp} = \frac{n\sigma_0^2}{72S\epsilon}$ iterations.

**(ii) Non-smooth $f_i$:** We borrow the construction $f_i(\cdot) = \beta \max\{\cdot, -\nu\}$ from Zhang & Lan (2020). We define that $F_i(x^{(i)}) := f(g_i(x)) + \frac{\alpha}{2}[x^{(i)}]^2 = \beta \max\{x^{(i)}, -\nu\} + \frac{\alpha}{2}[x^{(i)}]^2$ such that $F(x) = \frac{1}{n}\sum_{i=1}^n F_i(x^{(i)})$. Let the domain $\mathcal{X}$ be $[-2\nu, 2\nu]^n$. Since the problem is separable over the coordinates, we have $x_*^{(i)} = \arg\min_{x\in[-2\nu,2\nu]} F_i(x^{(i)}) = \arg\min_{x\in[-2\nu,2\nu]}\left\{\beta\max\{x^{(i)}, -\nu\} + \frac{\alpha}{2}(x^{(i)})^2\right\}$ for $x_* = \arg\min_{x\in\mathcal{X}} F(x)$. We have

$$x_*^{(i)} = \begin{cases} -\beta/\alpha & \text{if } \alpha > \beta/\nu \\ -\nu & \text{if } \alpha \in \frac{\beta}{\nu}[0,1] \end{cases}, \quad F_i(x_*^{(i)}) \leq \begin{cases} -\beta^2/(2\alpha) & \text{if } \alpha > \beta/\nu \\ -\beta\nu/2 & \text{if } \alpha \in \frac{\beta}{\nu}[0,1]. \end{cases}$$

Since $F_i(0) = 0$, we can derive that $F_i(0) - F_i(x_*^{(i)}) \geq \frac{1}{2}\min\{\beta\nu, \beta^2/\alpha\}$. By the convex conjugate, we have

$$f(\hat{g}^{(i)}) = \max_{y^{(i)}\in[0,\beta]}\{y^{(i)}\hat{g}^{(i)} - \nu(\beta - y^{(i)})\}.$$

Due to similar reason as in the smooth $f_i$ case, the probability of the event $E_T^{(i)} := \{\mathfrak{g}_T^{(i)} = \{0\} \text{ or } \{-\nu\} \wedge \mathfrak{Y}_T^{(i)} = \{0\} \wedge \mathfrak{X}_T^{(i)} = \{0\}\}$ can be lower bounded as

$$\mathbb{P}[E_T^{(i)}] \geq \mathbb{P}\left[\bigcap_{t=0}^{T-1} E_{t+1}^{(i)}\right] = \prod_{t=0}^{T-1}\mathbb{P}\left[E_{t+1}^{(i)} \mid \bigcap_{t=1}^t E_t^{(i)}\right] = \left(1 - \frac{Sp}{n}\right)^T > 3/4 - \frac{TSp}{n}.$$

Thus, letting $T < \frac{n}{4Sp}$ can make $\mathbb{P}[E_T^{(i)}] > \frac{1}{2}$. Choose $\beta = C_f$, $\nu = \frac{4\epsilon}{C_f}$, and $\sigma = \sigma_0$ such that $p := \frac{\nu^2}{\sigma^2} = \frac{16\epsilon^2}{C_f^2\sigma_0^2}$. For any $i \in [n]$ and any output $\tilde{x}_T^{(i)} \in \mathfrak{X}_T^{(i)}$, we have

$$\mathbb{E}[F_i(\tilde{x}_T^{(i)}) - F_i(x_*^{(i)})] = \mathbb{E}\left[\mathbb{I}_{E_T^{(i)}}\left(F_i(\tilde{x}_T^{(i)}) - F_i(x_*^{(i)})\right) + \mathbb{I}_{\overline{E_T^{(i)}}}\left(F_i(\tilde{x}_T^{(i)}) - F_i(x_*^{(i)})\right)\right] \geq \mathbb{E}\left[\mathbb{I}_{E_T^{(i)}}\left(F_i(\tilde{x}_T^{(i)}) - F_i(x_*^{(i)})\right)\right]$$

$$= \mathbb{E}\left[\mathbb{I}_{E_T^{(i)}}\left(F_i(0) - F_i(x_*^{(i)})\right)\right] = \mathbb{P}[E_T^{(i)}]\left(F_i(0) - F_i(x_*^{(i)})\right) > \min\{\beta\nu, \beta^2/\alpha\}/4 = \epsilon.$$

Since the derivations above hold for arbitrary $i \in [S]$, we can derive that

$$\mathbb{E}[F(\tilde{x}_T) - F(x_*)] = \frac{1}{n}\sum_{i=1}^n \mathbb{E}[F_i(\bar{x}^{(i)}) - F_i(x_*^{(i)})] > \epsilon.$$

Thus, to find a $\tilde{x}_T$ such that $\mathbb{E}[F(\bar{x}) - F(x_*)] \leq \epsilon$, the abstract scheme requires at least $T \geq \frac{n}{4Sp} = \frac{nC_f^2\sigma_0^2}{64S\epsilon^2}$ iterations.

## F. Application to GDRO with $\phi$-divergence

We discuss two examples of the GDRO problem with $\phi$-divergence: CVaR divergence with a hyper-parameter $\alpha \in (0,1)$ and $\chi^2$-divergence with a hyper-parameter $\lambda > 0$. We compare ALEXR to the following baselines:

● SMD (Nemirovski et al., 2009; Zhang et al., 2023): It can be applied to the GDRO problem in (7) with CVaR divergence, where the dual mirror step with the entropy distance-generating function can be efficiently solved by projection onto the permutahedron (Lim & Wright, 2016). Moreover, SMD can also be applied to the worst-group DRO problem (Sagawa et al., 2019) (i.e., $\lambda = 0$ in (7) or $\alpha = \frac{1}{n}$ in CVaR). The iteration complexity of SMD is $T = O(\frac{\log n}{\epsilon^2})$. Besides, it requires $O(n\log n)$ computational cost for performing the dual projection and $O(n)$ oracles in each iteration. Note that SMD cannot be applied to the GDRO problem in (7) with $\chi^2$-divergence due to the non-linear penalty term.

● OOA (Sagawa et al., 2019): This algorithm can be viewed as a variant of the SMD algorithm with the dual gradient estimator $[0, \ldots, n\ell(w_t; z_t^{(i_t)}), \ldots, 0]^\top$ for some $i_t \in [n]$ such that it only requires $O(1)$ oracles per iteration. But the dual

*Table 4.* Comparison of iteration complexities, dual projection cost, and per-iteration #oracles for achieving $\epsilon$-optimal solution of the GDRO problem in (7) in terms of $\mathbb{E}[F(w_{\text{out}}) - F(w_*)] \leq \epsilon$ in the merely convex case and $\frac{\mu}{2}\mathbb{E}\|w_{\text{out}} - w_*\|_2^2 \leq \epsilon$ in the $\mu$-strongly convex case, where $x_{\text{out}}$ is the output of each algorithm. We hide other constant quantities except for $n$, variances $\sigma_0^2, \sigma_1^2, \delta^2$, and batch sizes $B, S$. Besides, $\tilde{O}$ hides poly $\log(1/\epsilon)$ factors.

| $\phi$-Divergence | Algorithm | Per-Iter #Oracles | Dual Proj. | Iteration Complexity | |
|---|---|---|---|---|---|
| CVaR | SMD | $O(n)$ | $O(n\log n)$ | $O\left(\frac{\log n}{\epsilon^2}\right)$ | |
| | OOA | $O(1)$ | $O(n\log n)$ | $O\left(\frac{n^2\log n}{\epsilon^2}\right)$ | |
| | ALEXR | $O(1)$ | $O(1)$ | $O\left(\frac{\sqrt{n}}{\alpha^2\sqrt{S}\epsilon} + \frac{1}{\alpha^2\epsilon^2} + \frac{\delta^2}{S\epsilon^2} + \frac{\sigma_1^2}{\alpha^2 B\epsilon^2} + \frac{\sigma_0^2\Omega_{\mathcal{Y}}^0}{BS\epsilon^2}\right)^{\dagger}$ | |
| $\chi^2$ | ALEXR | $O(1)$ | $O(1)$ | Merely Convex $r$ | Strongly Convex $r$ |
| | | | | $O\left(\frac{\sqrt{n}}{\lambda\sqrt{S}\epsilon} + \frac{1}{\lambda^2\epsilon^2} + \frac{\delta^2}{S\epsilon^2} + \frac{\sigma_1^2}{B\epsilon^2} + \frac{\sigma_0^2\Omega_{\mathcal{Y}}^0}{BS\epsilon^2}\right)$ | $\tilde{O}\left(\frac{\sqrt{n}}{\lambda\sqrt{S}\mu} + \frac{1}{\mu\lambda^2\epsilon} + \frac{n\sigma_0^2}{BS\epsilon} + \frac{\sigma_1^2}{\mu B\epsilon} + \frac{\delta^2}{\mu S\epsilon}\right)$ |

$\dagger$ The worst-case estimate of $\Omega_{\mathcal{Y}}^0$ is $\frac{n}{2\alpha^2}$, but it could be much smaller than $\frac{n}{2\alpha^2}$ in practice, as explained in Remark F.1.

projection cost in each iteration is still $O(n\log n)$. The iteration complexity of SMD is $T = O(\frac{n^2\log n}{\epsilon^2})$, which is also independent of $\alpha$. OOA is not applicable to the GDRO problem in (7) with $\chi^2$-divergence either.

It comes to our attention that the NOL algorithm (Zhang et al., 2023) designed for the worst-group DRO problem, i.e., $\lambda = 0$ in (7), can achieve $T = O(\frac{n\log n}{\epsilon^2})$ iteration complexity in high probability with per-iteration $O(1)$ oracles. However, this result cannot be extended to the GDRO problem with CVaR or $\chi^2$-divergence, since their proof technique relies on powerful tools for multi-armed bandits. Besides, Soma et al. (2022) also consider the GDRO problem with CVaR divergence but their convergence analysis suffers from dependency issues, as pointed out in Zhang et al. (2023). Recently, Hu et al. (2023c) studied non-smooth weakly convex FCCO problems and proposed an algorithm SONX, which can be applied to solving GDRO with CVaR divergence. However, their algorithm does not leverage the convexity of the inner function and hence suffers from a worse complexity of $O(\frac{n}{S\sqrt{B}\epsilon^6})$.

### F.1. GDRO with CVaR divergence

GDRO with CVaR divergence can be formulated as (1) with $f_i(\cdot) = \alpha^{-1}(\cdot)_+$, $\alpha \in (0,1)$ and $g_i(w,c) = R_i(w) - c$ such that $C_f = \frac{1}{\alpha}$ and $C_g = C_R + 1$, where $C_R$ is the Lipschitz constant of $R_i$. The dual update (7) of ALEXR with $\psi_i(\cdot) = \frac{1}{2}(\cdot)^2$ has the closed-form expression $y_{t+1}^{(i)} = \begin{cases} \text{Proj}_{[0,\alpha^{-1}]}[y_t^{(i)} + (1/\tau)\tilde{g}_t^{(i)}], & i \in \mathcal{S}_t \\ y_t^{(i)}, & i \notin \mathcal{S}_t \end{cases}$.

The worst-case estimate of the $\Omega_{\mathcal{Y}}^0$ term in Theorem 3 is $\Omega_{\mathcal{Y}}^0 \leq \frac{nC_f^2}{2} = \frac{n}{2\alpha^2}$ when $\psi_i = \frac{1}{2}(\cdot)^2$. However, it could be much smaller than $\frac{n}{2\alpha^2}$ in practice since $\tilde{y}^{(i)} = 0$ for those coordinates $i$ that satisfy $R_i(\bar{w}) \leq \bar{c}$, i.e., the ALEXR algorithm can benefit from the "sparsity" of $\tilde{y}^{(i)} \in \partial f_i(g_i(\bar{w}, \bar{c}))$, where $(\bar{w}, \bar{c})$ is the output of the algorithm. In particular, when $(\bar{w}, \bar{c})$ is close to the optimal solution, then roughly about $\alpha n$ number of groups such that $[R_i(\bar{w}) - \bar{c}]_+ > 0$. As a result, $\Omega_{\mathcal{Y}}^0 = \mathbb{E}[\sum_{i=1}^n U_{\psi_i}(\tilde{y}^{(i)}, 0)] \approx \frac{n\alpha C_f^2}{2} = \frac{n}{2\alpha}$.

### F.2. GDRO with $\chi^2$- divergence

GDRO with $\chi^2$- divergence can be formulated as (1) with $f_i(\cdot) = \lambda\left(\frac{1}{4}(\cdot + 2)_+^2 - 1\right)$ and $g_i(w,c) = (R_i(w) - c)/\lambda$ such that $C_f = \frac{\max\{B_R - \underline{c}, B_R + \bar{c}\}}{2}$ and $C_g = \frac{C_R + 1}{\lambda}$, where $B_R := \max_w |R_i(w)|$ and a valid choice of $\underline{c}, \bar{c}$ is $\underline{c} = -\lambda, \bar{c} = B_R$ (See Appendix A.3 in Levy et al. 2020). In this case, the proximal mapping of $f_i^*(y^{(i)}) = \frac{\lambda}{2}(y^{(i)}/\lambda - 1)^2$ with $\psi_i(\cdot) = \frac{1}{2}(\cdot)^2$ can also be efficiently solved. We can also consider the GDRO problem with a convex regularization term $r(x)$. We can choose either $\psi_i = f_i^*$ or $\psi_i(\cdot) = \frac{1}{2}(\cdot)^2$.

### F.3. Comparison with Baselines

In Table 4, we compare our ALEXR to the baseline algorithms OOA and SMD. It is notable that although SMD has a better iteration complexity for CVaR divergence, it requires $O(n)$ oracles at each iteration. In contrast, ALEXR and OOA only require $O(1)$ oracles in each iteration. In the worst case, we have $\Omega_{\mathcal{Y}}^0 = O(n/\alpha^2)$ for CVaR-penalized GDRO, then ALEXR

has a better complexity than OOA when $\frac{1}{\alpha} = o(\sqrt{n \log n})$. In practice, we have $\Omega_{\mathcal{Y}}^0 = O(n/\alpha)$ for CVaR-penalized GDRO, then ALEXR has a better complexity than OOA when $\frac{1}{\alpha} = o(n \log n)$. In addition, OOA cannot enjoy the parallel speedup with respect to the inner batch size $B$ due to its scaled dual gradient estimator. Moreover, we also provide the iteration complexity of ALEXR on this the GDRO problem with $\chi^2$-divergence, with or without a strongly convex regularizer.

# G. More Details of Experiments

All algorithms are implemented using the PyTorch framework. Experiments are conducted on a workstation with the 12th Gen Intel(R) Core(TM) i7-12700K CPU with 20 logical cores.

## G.1. Group Distributionally Robust Optimization

### G.1.1. DATA PREPROCESSING

**Adult dataset:** We construct 83 groups for the Adult dataset according to income ("$>$50K", "$\leq$50K"), race ("white", "black", "other"), sex ("female", "male"), age ("$\leq$30", "30-45", "$>$45"), relationship ("single", "not_single"), and education ("higher", "others"), where we discard those groups with less than 50 data points. Following Platt (1999), we transform both continuous and categorical features into binary features, resulting in a 122-dimensional feature vector for each data.

**CelebA dataset:** We construct 160 groups for this dataset according to 4 binary attributes ("blond hair", "male", "mouth slightly open", "smiling") and 10 types of additive Gaussian noises (means -0.08:0.02:0.1 and variance 0.08) to the images. Each image of the CelebA dataset is resized to 224×224×3, normalized, and center-cropped. Then, we extract 512-dim feature vectors for those preprocessed images from the last convolutional layer of a ResNet18 pre-trained on ImageNet.

### G.1.2. ADDITIONAL RESULTS

The first two columns of Figure 3 show the existence of rare groups in the datasets. The last two columns of Figure 3 demonstrate that the actual value of $\Omega_{\mathcal{Y}}^0$ is indeed much smaller than its worst-case estimate $\frac{n}{2\alpha^2}$, which verifies the claims in the remark below Theorem 3 and Section F.1.

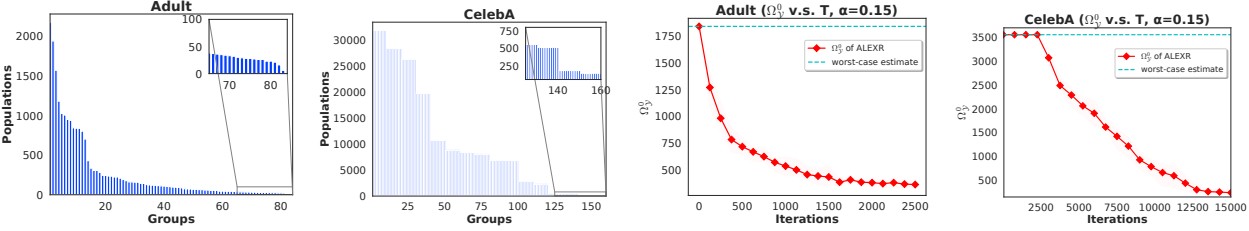

*Figure 3.* Group populations and the computed values of $\Omega_{\mathcal{Y}}^0$.

## G.2. Partial AUC Maximization with Restricted TPR

*Table 5.* Statistics of datasets used in the partial AUC maximization experiments. Here $n_+$ and $n_-$ refer to the numbers of positive and negative data in the train and validation splits.

| Datasets | Train | | Validation | |
|---|---|---|---|---|
| | $n_+$ | $n_-$ | $n_+$ | $n_-$ |
| Covtype | 889 | 178,587 | 252 | 59,573 |
| Higgs | 4,676 | 4,172,030 | 582 | 499,418 |
| Cardiomegaly | 1,950 | 76,518 | 240 | 10,979 |
| Lung-mass | 3,988 | 74,480 | 625 | 10,594 |

