# OpenReview forum: "A Near-Optimal Single-Loop Stochastic Algorithm for Convex Finite-Sum Coupled Compositional Optimization"
_ICML.cc/2025/Conference — ICML 2025 poster_

### Official Review · Reviewer_EbXm · 2025-03-10

**Overall Recommendation:** 4

**Summary:**

The authors study a convex compositional problem with a particular structure and propose a new algorithm called Alexr.

## update after rebuttal
I think the paper deserves to be accepted and I am confident that the authors will make the recommended changes to make the paper even better.

**Claims And Evidence:**

The claims are convincing. But my confidence with respect to the novelty and state of the art is low, as this topic is far from my expertise area.

**Essential References Not Discussed:**

I am not an expert of this class of problems. (2) is a convex-concave saddle-point problem with non-bilinear coupling. The function $g_i$ in the coupling is accessed via a stochastic oracle, and I am not familiar with this setting.

Can you tell me if the following papers are relevant?

* Alacaoglu et al. "Forward‑reflected‑backward method with variance reduction", 2021
* Alacaoglu and Malitsky "Stochastic Variance Reduction for Variational Inequality Methods" 2022
* Zhu et al. "Accelerated Primal-Dual Algorithms for a Class of Convex-Concave Saddle-Point Problems with Non-Bilinear Coupling Term," 2020

**Experimental Designs Or Analyses:**

The experiments look sound.

**Methods And Evaluation Criteria:**

The evaluation setups are good.

**Other Comments Or Suggestions:**

typos:
* regularizatoin
* line 1362: minF(x)

**Other Strengths And Weaknesses:**

Again, this topic is not in my expertise area, but this class of optimization problems seems to have interesting applications in machine learning.

**Questions For Authors:**

Line 60: "By the convex conjugate" do you mean that this follows from some duality framework? Can you give some references on this?

**Relation To Broader Scientific Literature:**

The literature review seems good to me. Some papers I had in mind on saddle point problems are cited.

**Theoretical Claims:**

I did not check the correctness of the proofs.

---

> ### Author Rebuttal · Authors · 2025-03-30
>
> We thank the reviewer for taking the time to read our paper and greatly appreciate your valuable feedback.
>
> > **Q1:** Can you tell me if the following papers are relevant? (1) Alacaoglu et al. "Forward‑reflected‑backward method with variance reduction", 2021; (2) Alacaoglu and Malitsky "Stochastic Variance Reduction for Variational Inequality Methods" 2022; (3) Zhu et al. "Accelerated Primal-Dual Algorithms for a Class of Convex-Concave Saddle-Point Problems with Non-Bilinear Coupling Term," 2020.
>
> **A:** Thanks for bring these interesting works to our attention.
>
> - Zhu et al. study the convex-concave min-max optimization problem with a non-bilinear coupling term, which is similar to the reformulation (2) in our work. However, there are key differences in our problem setting:
>     - i) In our formulation (2), the problem is separable with respect to each coordinate of the dual variable $y$, whereas the problem in Zhu et al. is not;
>     - ii) Zhu et al. focus on the deterministic setting, while our work focuses on the stochastic setting.
> - Both Alacaoglu et al. and Alacaoglu and Malitsky study the monotone variational inequality problem with a finite-sum structure, which includes the concave min-max optimization problem with a finite-sum structure as a special case. Although our problem in (2) also has a finite-sum structure, there are two differences:
>     - i) Our problem (2) is separable with respect to each coordinate of the dual variable $y$, whereas the problems considered in Alacaoglu et al. and Alacaoglu and Malitsky are not;
>     - ii) Alacaoglu et al. and Alacaoglu and Malitsky assume that each summand in the finite sum of losses is directly available. In contrast, our work only requires that each summand be accessible through stochastic oracles.
>
> We will add these discussions in the revision.
>
> > **Q2:** Line 60: "By the convex conjugate" do you mean that this follows from some duality framework? Can you give some references on this?
>
> **A:** Sorry for the confusion. We refer to that the value of a convex and and lower semi-continuous function $f_i$ on any $u$ in its domain can be represented as $f_i(u) = \max_{y^{(i)}\in\mathcal{Y}_i}\{u^\top y^{(i)} - f_i^*(y^{(i)})\}$, where $f_i^*$ is the convex conjugate of $f_i. The formal definition can be found in Section 12 of Rockafellar (1997).
>
> Rockafellar, R.T., 1997. Convex analysis (Vol. 28). Princeton university press.
>
> > **Q3:** typos: regularizatoin; line 1362: minF(x)
>
> **A:** Thanks for pointing these out. We will fix them in the next version of our draft.

---

### Official Review · Reviewer_J4zJ · 2025-03-14

**Overall Recommendation:** 4

**Summary:**

This paper introduces ALEXR, a single-loop, primal-dual block-coordinate algorithm aimed at convex finite-sum coupled compositional problems (both strongly convex and merely convex, even if nonsmooth). By carefully interleaving primal updates with an extrapolated dual variable, the authors achieve improved or near-optimal oracle complexities, bolstered by new lower complexity bounds that highlight the theoretical tightness of their approach. Empirical demonstrations on distributionally robust optimization and partial AUC show consistent performance gains, underscoring the method’s practicality.

**Claims And Evidence:**

The paper’s primary strength lies in its near-optimal theoretical guarantees, achieved through a novel single-loop, primal-dual algorithmic design that replaces more conventional nested structures. This not only simplifies analysis but also clarifies step-by-step convergence arguments, particularly under nonsmooth convex conditions. The authors further bolster the method’s theoretical significance by establishing new lower complexity bounds, underscoring the algorithm’s tightness and rigor. Additionally, they demonstrate clear empirical benefits on group distributionally robust optimization and partial AUC tasks, showing that their approach is both grounded in strong theory and relevant to practical ML scenarios.

**Essential References Not Discussed:**

No

**Experimental Designs Or Analyses:**

The reviewer briefly looked over the numerical experiments to verify their design but did not check the details.

**Methods And Evaluation Criteria:**

Yes

**Other Comments Or Suggestions:**

No

**Other Strengths And Weaknesses:**

No

**Questions For Authors:**

No

**Relation To Broader Scientific Literature:**

The paper builds on and extends prior work on coupled compositional optimization – notably approaches like those by Wang et al. (2022) and Jiang et al. (2022) – by providing a single-loop alternative to the previously more common nested-loop frameworks.

The main motivation is the dual formulation (2) of the primal problem (1), inspired by the duality result of Levy et al. (2020). This duality applies to both GDRO and partial AUC. The applications in GDRO and partial AUC are in line with the growing interest in robust and fair optimization problems.

**Theoretical Claims:**

No

---

> ### Author Rebuttal · Authors · 2025-03-30
>
> We thank the reviewer for taking the time to read our paper and greatly appreciate your valuable feedback.

---

### Official Review · Reviewer_tEp6 · 2025-03-18

**Overall Recommendation:** 3

**Summary:**

The paper presents ALEXR, a novel stochastic algorithm for convex Finite-sum Coupled Compositional Optimization (cFCCO). It reformulates cFCCO as a convex-concave min-max problem and introduces a single-loop primal-dual block-coordinate stochastic approach. ALEXR applies mirror ascent for the dual variable and proximal gradient descent for the primal variable. The authors establish convergence rates under various smoothness conditions and derive lower complexity bounds, proving ALEXR’s near-optimality. Experiments on GDRO and pAUC maximization highlight its strong performance.

## update after rebuttal

The authors' response addressed my concern. I’m keeping my score—not because the paper lacks merit, but because there's no option for 3.5. I recommend for accept in general.

**Claims And Evidence:**

The claims made in the paper are generally well-supported by clear and convincing evidence.

**Essential References Not Discussed:**

Not that I am aware of

**Experimental Designs Or Analyses:**

The experimental designs and analyses are sound and valid. The authors compare ALEXR against several baseline algorithms on GDRO and pAUC maximization tasks, using appropriate datasets and evaluation metrics. The results show that ALEXR outperforms the baselines in most cases, demonstrating its effectiveness. However, the paper could benefit from more extensive ablation studies to understand the impact of different hyperparameters and algorithmic components on the performance of ALEXR.

**Methods And Evaluation Criteria:**

The proposed methods and evaluation criteria are appropriate for the problem at hand. The authors focus on cFCCO problems, which have applications in machine learning tasks such as GDRO and pAUC maximization. The use of benchmark datasets like Adult and CelebA for GDRO, and Covtype, Cardiomegaly, Lung-mass, and Higgs for pAUC maximization, aligns with the problem domains. The evaluation metrics, such as test accuracy and pAUC, are relevant and provide meaningful insights into the performance of the algorithms.

**Other Comments Or Suggestions:**

1. Some hyperparameter settings are not explicitly stated (e.g., specific learning rates, batch sizes for all algorithms). A table summarizing the experimental setup would improve reproducibility.

2. It would be helpful to present and discuss runtime performance across different algorithms, not just convergence speed in terms of oracle complexity.

**Other Strengths And Weaknesses:**

### Strengths
1. **Originality:** The proposed ALEXR algorithm is a novel single-loop stochastic method for convex finite-sum coupled compositional optimization (cFCCO). It improves upon prior methods by removing the nested inner loop that make previous approaches computationally complex. The algorithm incorporates a primal-dual block-coordinate approach, motivated by the literature.

2. **Significance:** The study has theoretical and practical relevance, proving lower bounds that confirm the near-optimality of ALEXR. It expands the application of cFCCO to non-smooth problems, including GDRO and partial AUC maximization. The algorithm demonstrates superior performance in real-world ML applications.

3. **Clarity:**  The paper is well-written.The comparisons to previous algorithms are well-structured in tables and thoroughly discussed.

### Weaknesses
1. While ALEXR performs well in theoretical benchmarks, the paper does not discuss practical constraints such as hyperparameter sensitivity and tuning difficulty.

 2. There is no in-depth ablation study on how different components of ALEXR (e.g., choice of $\psi$ functions, step size schedules) affect performance.

**Questions For Authors:**

1.  The paper provides strong theoretical results, but there is limited discussion on the sensitivity of ALEXR to hyperparameter choices (e.g., extrapolation parameter $\theta$, step sizes, and choice of $\psi_i$).  How sensitive is ALEXR’s performance to those choices?


2.  How does the wall-clock runtime of ALEXR compare to baselines like SOX and MSVR?

**Relation To Broader Scientific Literature:**

The key contributions of the paper are well-aligned with the broader scientific literature on stochastic optimization and machine learning. The authors build on prior work in convex stochastic compositional optimization, coupled compositional optimization, and primal-dual methods for empirical risk minimization. They extend these ideas to the cFCCO setting and propose a novel algorithm that improves upon existing methods in terms of oracle complexity and applicability to non-smooth problems.

**Theoretical Claims:**

The theoretical claims in the paper are supported by detailed proofs provided in the appendices. I didn't carefully check proofs in the appendix, but they appear to be right based on my previous understanding of existing literature.

---

> ### Author Rebuttal · Authors · 2025-03-30
>
> We thank the reviewer for taking the time to read our paper and greatly appreciate your valuable feedback.
>
> > **Q1:** The paper provides strong theoretical results, but there is limited discussion on the sensitivity of ALEXR to hyperparameter choices. How sensitive is ALEXR’s performance to those choices?
>
> **A:** ALEXR has three hyperparameters: $\eta$, $\tau$, and $\theta$. Although $\eta$ and $\tau$ need to be tuned, the sensitivity analysis below shows that ALEXR is not highly sensitive to small variations of their values. Fixing $\theta = 0.1$ seems to work well across all of our experiments and datasets. Thus, the tuning effort of ALEXR is similar to baselines SOX, MSVR, and OOA.
>
> ### i) Sensitivity to step sizes
> Note that in ALEXR, $1/\eta$ is the step size for the primal variable $x$, while $1/\tau$ is step size for the dual variable $y$. We sweep over different values of step sizes in the Group DRO experiments. The top-tier choices of step sizes are highlighted.
>
> (Adult Dataset, $\alpha=0.1$)
>
> |Primal/dual stepsizes|$\frac{1}{\eta}=$  0.02 	          | 0.05            | 0.1 | 0.2 | 0.5 | 1.0
> :-----:|:-----:|:-----:|:-----:|:-----:|:-----:|:-----:
> **$\frac{1}{\tau}=$ 0.02** |0.776 | 0.804| 0.789|0.782 | 0.770|0.773
> **0.05** | 0.739| 0.716| 0.721| 0.703| 0.719|0.737
> **0.1** | 0.694| 0.691| 0.682| 0.688| 0.704|0.704
> **0.2** | **0.681** | **0.675**| **0.678**| 0.687|0.684 |0.707
> **0.5** | **0.677**|**0.678**| **0.680**| **0.677**| 0.689|0.762
> **1.0** |**0.679**| **0.679**| **0.676**| **0.680**| 0.695|1.074
>
> (CelebA Dataset, $\alpha=0.1$, '-' refers to divergence)
> |Primal/dual stepsizes|$\frac{1}{\eta}=$  0.002 | 0.005 | 0.01 | 0.02 	          | 0.05 | 0.1
> :-----:|:-----:|:-----:|:-----:|:-----:|:-----:|:-----:
> **$\frac{1}{\tau}=$ 0.002**|0.501 | 0.501| 0.501 | 0.500| 0.506|0.670
> **0.005** | 0.497 | 0.487 | 0.490 | 0.486 | 0.493|0.721
> **0.01**| 0.486 | 0.482 | 0.477 | 0.477 | 0.481|0.785
> **0.02** | 0.479| **0.470**| **0.468**|**0.467** |**0.470** |-
> **0.05** | 0.474| **0.467** | **0.461** | **0.459**| 0.484| -
> **0.1** | 0.475 | **0.467** | **0.464** | **0.464**|0.495| -
>
> As shown in the results above, while the step sizes require tuning for each dataset, they are **not** highly sensitive to small variations.
>
> ### ii) Choice of $\psi_i$
>
> Since the GDRO problem with the CVaR divergence has a non-smooth $f_i$, we report the experimental results of ALEXR with the choice quadratic choice $\psi_i(u)=\frac{1}{2}\||u\||_2^2$, as discussed in Line 199 of our paper. Next, we empirically compare the results under the quadratic choice with those under another choice $\psi_i=f_i^*$. We separately tune the step sizes for the choice $\psi_i=f_i^*$.
>
> | Choice of $\psi_i$| Adult ($\alpha=0.1$) | Adult ($\alpha=0.15$) | CelebA ($\alpha=0.1$) | CelebA ($\alpha=0.15$)
> :-----:|:-----:|:-----:|:-----:|:-----:
> $\psi_i=f_i^*$| 0.7165| 0.6866 | 0.5139|0.4746
> **Quadratic** (chosen in our paper) | 0.6763 | 0.6646 | 0.4593| 0.4521
>
> The results show that $\psi_i(u)=\frac{1}{2}\\||u\\||_2^2$ is indeed a better choice than $\psi_i=f_i^*$ for the non-smooth GDRO problem.
>
> > **Q2:** It would be helpful to present and discuss runtime performance across different algorithms, not just convergence speed in terms of oracle complexity. How does the wall-clock runtime of ALEXR compare to baselines like SOX and MSVR?
>
> **A:** Next, we report the total wall-clock runtime for the GDRO experiment on the larger dataset CelebA.
>
> | Algorithms| OOA | SOX | MSVR| ALEXR (Ours)
> :-----:|:-----:|:-----:|:-----:|:-----:
> Total Runtime (s) | 45.31 | 44.56 | 47.88|45.89
>
> The result above shows that runtime of ALEXR is comparable to the baselines while achieving a faster convergence rate.
>
> > **Q3:** Some hyperparameter settings are not explicitly stated (e.g., specific learning rates, batch sizes for all algorithms). A table summarizing the experimental setup would improve reproducibility.
>
> Thank you for the suggestion! The tuned algorithm-specific hyperparameter settings for the GDRO experiments are:
>
> || SGD | SGD-UW | OOA | BSGD | SOX | MSVR | ALEXR
> :-----:|:-----:|:-----:|:-----:|:-----:|:-----:|:-----:|:-----:
> **Adult** | lr=0.1 | lr=0.1 | lr=1.0, lr\_dual=0.001 | lr=0.01|lr=0.01,$\gamma$=0.1 |lr=0.01,$\gamma$=0.9|$\frac{1}{\eta}$=0.1,$\frac{1}{\tau}$=1.0, $\theta$=0.1
> **CelebA** | lr=0.05 | lr=0.05|lr=0.01, lr\_dual=0.05 | lr=0.01 | lr=0.01, $\gamma$=0.5 | lr=0.01, $\gamma$=0.5| $\frac{1}{\eta}$=0.02,$\frac{1}{\tau}$=0.05, $\theta$=0.1
>
> The shared settings for all algorithms in the GDRO experiments are:
> || Batch sizes | \#Iterations | Quantile $\alpha$
> :-----:|:-----:|:-----:|:-----:
> **Adult** |64 (B=8 from each S=8)|2500|10\%, 15\%
> **CelebA**| 64 (B=8 from each S=8) |15000 |10\%, 15\%
>
> Due to character limitations, we will also include settings for the pAUC experiments in the next version of our draft.

---

### Official Review · Reviewer_Zg46 · 2025-03-23

**Overall Recommendation:** 3

**Summary:**

This paper studies a regularized convex finite-sum coupled compositional optimization (cFCCO) problems: $\min F(x),  \text{  where  } F(x) := \frac{1}{n}\sum_{i = 1}^n f_i(g_i(x)) + r(x)$. Here, all functions ($f_i, g_i, r, F$) are convex, and each inner function $g_i(x) = E_{\xi_i \sim P_i} [g_i(x; \xi_i)]$ for distinct distributions and is accessed via stochastic oracle queries. The authors also assume that $g_i, f_i$ are Lipschitz continuous (and possibly smooth), and when $g_i$ nonlinear $f_i$ is assumed monotonically non-decreasing, and the variances of the zeroth-order and first-order stochastic oracles are bounded. This kind of problems find applications in group distributionally robust optimization and learning with imbalanced data, which later their experiments study.

The authors reformulate the problem into a convex-concave min-max problem using the conjugacy of $f_i$ and propose a single-loop primal-dual algorithm, namely ALEXER, which can viewed as a randomized extrapolated coordinate algorithm on the dual side and a proximal stochastic gradient method on the primal side. Convergence analysis and nonasymptotic guarantees are provided for the proposed algorithm, under both strongly convex and convex settings, with improved dependence on the number of components $n$ and the strong convexity parameter $\mu$. The authors also include certain worst-case examples to show the tightness of their complexity results.

**Claims And Evidence:**

Yes.

**Essential References Not Discussed:**

- It would be helpful to cite the seminal works [Chambolle & Pock (2011); Chambolle et al. (2018)] when the authors introduce the min-max reformulation for the original minimization problem.
- It is helpful to discuss the prior work on cyclic coordinate methods, such as Song & Diakonikolas, 2021.
- For the min-max reformulation and the perspective of coordinates methods on the dual side, it is helpful to cite the prior work on shuffled SGD.


Chambolle, A. and Pock, T. A first-order primal-dual algorithm for convex problems with applications to
imaging. Journal of Mathematical Imaging and Vision, 40(1):120–145, 2011.

Chambolle, A., Ehrhardt, M. J., Richtárik, P., and Schonlieb, C.-B. Stochastic primal-dual hybrid gradient
algorithm with arbitrary sampling and imaging applications. SIAM Journal on Optimization, 28(4):2783–2808,
2018.

Chaobing Song and Jelena Diakonikolas. Cyclic coordinate dual averaging with extrapolation. SIAM
Journal on Optimization, 33(4):2935–2961, 2023. doi: 10.1137/22M1470104.

Xufeng Cai, Cheuk Yin Lin, and Jelena Diakonikolas. Tighter convergence bounds for shuffled SGD
via primal-dual perspective. In Proc. NeurIPS’24, 2024.

**Experimental Designs Or Analyses:**

This paper provides quite substantial experiments.

**Methods And Evaluation Criteria:**

Yes.

**Other Comments Or Suggestions:**

- What is the notation $u_t$ appearing in Section 3.1-3.2 and in the proofs in the appendix? Seems that the authors have refactored the writeup but are not careful to unify the notation.
- In Eq. (5), it seems that there should be $L(x, \bar y_{t + 1}) - L(x_{t + 1}, y)$ instead of $L(x_{t + 1}, y) - L(x, \bar y_{t + 1})$ on the right-hand side, based on the discussion and Lemma 5 in the appendix.

## Update after rebuttal

I appreciate author's responses, and they addressd my concerns.

I found this paper is interesting, from the aspects of (cf) nested stochastic optimization or bilevel optimization, though the complexity results appear to be a bit hard to parse. Possibly other dependece besides $epsilon$ can be improved.

I also suggest:

1. Make the notation more clearly defined, such as $u_t$.

2. Have the batch sizes explicitly stated in the main body and the comparison table


Overall, I recommend acceptance.

**Other Strengths And Weaknesses:**

**Strengths**
- It is interesting to see the viewpoint and the analysis of block coordinate updates on the dual side, though it is natural from the min-max reformulation and the stochastic design on the primal side.
- The convergence analysis is substantial under convex settings, including both smooth and nonsmooth cases. The authors also include lower bound results.

**Weaknesses**
- Assumption 4 looks strong to me: the authors assume the bounded variance for both zeroth- and first-order oracles.
- It seems that the analysis mainly relies on the Lipschitzness of the functions, and the smoothness only helps improve the batch size requirements.

**Questions For Authors:**

- Could the authors explain why they need the extrapolation (Line 6 of Algorithm 1) on the dual side?
- I do not quite get how the authors derive $B, S = O(1)$ from the main body. Could the authors explain their choice of $B$ and $S$?
- For the challenge the authors discussed in Eq. (5), it is usual that the analysis of the algorithm with full updates can be adapted because here one only have randomized coordinate updates to deal with (in contrast to cyclic coordinate methods). Could the authors explain the additional technical challenges here?
- For Theorem 2, what is the step size constraints for the algorithm for smooth $g_i$ and $f_i$? Do they still require $O(1/\epsilon)$ small step sizes, or they only depend on the smoothness constants?
- What are the assumption (cf Assumption 4 here) the prior work on stochastic compositional optimization made? Do they also require the bounded variance for both zeroth- and first-order oracles?
- What is the key challenge of removing the monotonicity assumption of $f_i$?

**Relation To Broader Scientific Literature:**

This work should relate to prior work on (stochastic) compositional optimization, primal dual algorithms, and coordinate algorithms.

**Theoretical Claims:**

I do not have time to check the proofs in the appendix, but the theoretical claims in the main body look reasonable to me.

---

> ### Author Rebuttal · Authors · 2025-03-30
>
> We thank the reviewer for taking the time to read our paper and greatly appreciate your valuable feedback.
>
> > **Q1:** Assumption 4 looks strong to me: the authors assume the bounded variance for both zeroth- and first-order oracles. What are the assumption (cf Assumption 4 here) the prior work on stochastic compositional optimization made? Do they also require the bounded variance for both zeroth- and first-order oracles?
>
> **A:** All prior works on stochastic compositional optimization make the same or stronger assumptions. For example, the seminal works [1, 2]  assume both of the following hold: (1) The variance of the zeroth-order oracle is bounded, which is the same as ours. (2) The first-order oracle is either almost surely bounded or has a bounded 2nd moment, which are stronger than the bounded variance assumption in our work. Recent works such as SOX and MSVR all assume the same conditions as ours.
>
> [1] Wang et al., 2017. Stochastic compositional gradient descent: algorithms for minimizing compositions of expected-value functions. Math. Program.
>
> [2] Wang et al., 2017. Accelerating stochastic composition optimization. JMLR.
>
> > **Q2:** It seems that the analysis mainly relies on the Lipschitzness of the functions, and the smoothness only helps improve the batch size requirements.
>
> **A:** That is partially true. First, the smoothness of $f_i$ also allows us to derive the optimal complexity of $O(1/\epsilon)$ for strongly convex problems, which is **better** than $O(1/\epsilon^2)$ for non-smooth $f_i$. While it is true that the smoothness of the inner functions $g_i$ only affects the batch size scaling in iteration complexity, this is expected, as standard SGD exhibits a similar distinction between smooth and non-smooth problems.
>
> > **Q3:** What is the notation $u_t$ appearing in Section 3.1-3.2 and in the proofs in the appendix? Seems that the authors have refactored the writeup but are not careful to unify the notation.
>
> **A:** Sorry for the confusion, but $u_t$ is not a notation inconsistency. Eq. (3) in our paper shows that if we set $\psi_i=f_i^*$, Lines 4–8 of our ALEXR algorithm can be rewritten in a form similar to the previous algorithms SOX and MSVR. To be specific, the updated $y_{t+1}^{(i)}$ can be expressed as the gradient $\nabla f_i$ evaluated at a moving-average estimator $u_{t+1}^{(i)} = \frac{\tau}{1+\tau}u_t^{(i)} + \frac{1}{1+\tau}\tilde{g}_t^{(i)}$. The form in (3) based on $u_t$ facilitates the discussion of relationship to SOX and MSVR in Section 3.2.
>
> >**Q4:** In Eq. (5), it seems that there should be $L(x,\bar{y}\_{t+1}) - L(x\_{t+1},y)$ instead of $L(x\_{t+1},y) - L(x,\bar{y}\_{t+1})$ on the right-hand side, based on the discussion and Lemma 5 in the appendix.
>
> **A:** Thank you for pointing out this typo! We will fix it in the next version of our draft.
>
> > **Q5:** Could the authors explain why they need the extrapolation (Line 6 of Algorithm 1) on the dual side?
>
> **A:** The extrapolation allows us to enjoy the batch size scaling in the iteration complexity by leveraging the smoothness of $g_i$. Without extrapolation (i.e., $\theta=0$), there is an additional term $\Gamma_{t+1}= \frac{1}{n} \sum_{i=1}^n (g_i(x_{t+1}) - g_i(x_t))^\top (y_*^{(i)}-y_{t+1}^{(i)})$ on the R.H.S. of (18) in Lemma 6. Directly bound this term leads to a worse rate when $g_i$ is smooth. With extrapolation, we can form a telescoping sum of $\Gamma_t$ and cancel it out. Similar idea is used in Lemma 10 for the merely convex case. We have the results in the full version of our paper and will include the discussions in the revision.
>
> > **Q6:** I do not quite get how the authors derive $B,S=O(1)$ from the main body. Could the authors explain their choice of $B$ and $S$?
>
> **A:** $B,S=O(1)$ in Table 1 means that our algorithm only requires a constant mini-batch size to converge (c.f. impractically large batch size of $O(\epsilon^{-1})$ or $O(\epsilon^{-1})$ in previous work BSGD) and enjoys a parallel speedup using mini-batches.
>
> > **Q7:** For the challenge the authors discussed in Eq. (5), it is usual that the analysis of the algorithm with full updates can be adapted because here one only have randomized coordinate updates to deal with (in contrast to cyclic coordinate methods). Could the authors explain the additional technical challenges here?
>
> **A:** There are two additional technical challenges here:
> - First, our algorithm includes the randomized coordinate stochastic updates for the dual variable, which cause the **dependence issue** discussed above Theorem 2, making it difficult to directly apply standard convergence analyses used in full-update methods.
> - The second challenge comes from the **stochastic** mirror ascent update for the sampled $y_i$, which make it more involved to derive a batch size scaled convergence rate when the inner functions are smooth.
>
> > **Q8:** Suggestions on citing prior works.
>
> **A:** Thank you! We will add these references in the next version of our draft.

---

### Official Review · Reviewer_7wWG · 2025-03-25

**Overall Recommendation:** 3

**Summary:**

This work studied convex FCCO problems, by reformulating into a convex-concave min-max problem, this work proposed ALEXR algorithm by incorporating coordinate descent and SAPD, the convergence guarantees are provided, also the lower bound analysis show that the complexity of proposed algorithm is near-optimal.

**Claims And Evidence:**

The claims come with convincing evidence in general. Here are some questions or comments:

1. For the lower bound part, there seems to be no clear definition of the oracle, regarding the layer-wise structure, you need to access $\nabla f_i$ and $\nabla g_i$ separately, also you further need to compute the proximal operator, I expect you can explicitly state the definition of the oracle, with the corresponding input and query output, similar to Zhang & Lan (2020) as you cited.
2. The lower bound result $\Omega(\max(S/\mu,n)\epsilon^{-1})$ looks weird to me, I think lower bound should be problem-intrinsic and depends on the function behavior, so the dependence on $n$ and $\mu$ (intrinsic problem parameters) looks fine to me; but $S$ should be a user-specified parameter, if $\mu$ is small enough, then you can choose $S$ to have a varing lower bound, which looks weird to me.
3. The algorithm for nonsmooth case works for simple $f$ only to compute the proximal mapping. Even though authors provided some examples for illustration, but the restriction may still hinder the applicability of the algorithm in general nonsmooth functions, compared to existing works.
4. What is the potential obstacles if you extend your cFCCO problem from finite-sum to stochastic problem ($\frac{1}{n}\sum_{i=1}^nf_i$ to $\mathbb{E}_if_i$), or you just intentionally choose the finite-sum problem in the study?

With that I think the work provided some interesting results for cFCCO, while some novel contribution should be further clarified to enhance the significance. I am open for further discussion.

**Essential References Not Discussed:**

No

**Experimental Designs Or Analyses:**

See above

**Methods And Evaluation Criteria:**

Make sense in general.

**Other Comments Or Suggestions:**

/

**Other Strengths And Weaknesses:**

/

**Questions For Authors:**

/

**Relation To Broader Scientific Literature:**

This work will advance the theoretical understanding of the algorithms for coupled compositional optimization problem.

**Theoretical Claims:**

See above

---

> ### Author Rebuttal · Authors · 2025-03-30
>
> We thank the reviewer for taking the time to read our paper and greatly appreciate your valuable feedback.
>
> > **Q1:** For the lower bound part, there seems to be no clear definition of the oracle, regarding the layer-wise structure, you need to access $\nabla f_i$ and $\nabla g_i$ separately, also you further need to compute the proximal operator, I expect you can explicitly state the definition of the oracle, with the corresponding input and query output, similar to Zhang & Lan (2020) as you cited.
>
> **A:** The oracle complexity in lower bound corresponds to the total number of calls of $g_i(x;\zeta^{(i)})$ and $\nabla g_i(x;\tilde{\zeta}^{(i)})$ required by any algorithm within the abstract scheme to achieve an $\epsilon$-accurate solution. In addition, we assume the proximal mapping of $f_i^*$ can be computed when $f_i$ is non-smooth similar to Zhang & Lan (2020). Nevertheless, the number of proximal mappings of $f_i^*$ is bounded by the number of calls of $g_i(x;\zeta^{(i)})$.  We will include this clarification in the next version of our draft.
>
> > **Q2:** The lower bound result $\Omega(\max(S/\mu,n)\epsilon^{-1})$ looks weird to me, I think lower bound should be problem-intrinsic and depends on the function behavior, so the dependence on $n$ and $\mu$ (intrinsic problem parameters) looks fine to me; but $S$ should be a user-specified parameter, if $\mu$ is small enough, then you can choose $S$ to have a varing lower bound, which looks weird to me.
>
> **A:** Thanks for your insightful question. While the lower bound naturally depends on important problem parameters such as $n$ and $\mu$, it is also tied to an abstract update scheme that describes how the space of the iterate evolves across the iterations.  Please note that $S$ is a parameter of our abstract scheme presented in pg.6, which covers the typical setting $S=1$ for lower bound analysis of randomized coordinate algorithms (e.g., Ch. 5.1.4 in [1]). We intended to be general so that the abstract scheme covers our ALEXR and some baselines listed in Table 1.  Our lower bound $\Omega(\max(S/\mu,n)\epsilon^{-1})$ characterizes the different behaviors of the abstract scheme in two different configurations of the problem-dependent parameters $n$, $\mu$:
> - Case I ($\frac{n}{S}>\frac{1}{\mu}$): In this case, the problem's hardness primarily arises from the high dimension of the dual variable $y$. Here, the block-coordinate update of $y$ makes the term $\frac{n}{S\epsilon}$ dominate the iteration complexity, resulting in an oracle complexity bound of $\Omega(n\epsilon^{-1})$. Increasing $S$ makes more coordinates of $y$ are updated per iteration, thereby reducing the number of required iterations to find an $\epsilon$-accurate solution of the whole problem.
> - Case II ($\frac{n}{S}\leq \frac{1}{\mu}$): In this case, the main hardness of the problem is from the ill-conditioning of the stochastic proximal update of primal variable $x$. Thus, increasing $S$ (i.e., updating more coordinates of the dual variable $y$) does not improve the leading term $\frac{1}{\mu\epsilon}$ of the iteration complexity, leading to a higher oracle complexity.
>
> > **Q3:** The algorithm for nonsmooth case works for simple $f$ only to compute the proximal mapping. Even though authors provided some examples for illustration, but the restriction may still hinder the applicability of the algorithm in general nonsmooth functions, compared to existing works.
>
> **A:** We agree with the reviewer. However, existing works on cFCCO that only require computing the gradient of $f_i$, either focus on smooth problems (e.g., SOX and MSVR) or require a gigantic batch size to converge (e.g., BSGD). All of them have suboptimal complexities. Our work is the first to study the more challenging non-smooth cFCCO problem and achieve the optimal rates. In cases where the the proximal mapping of $f_i^*$ is not easily computed, our algorithm can be combined with an approach that solves the proximal sub-problems inexactly (e.g., [2]). Then, the convergence guarantee can be still established.
>
> > **Q4:** What is the potential obstacles if you extend your cFCCO problem from finite-sum to stochastic problem ($\frac{1}{n}\sum_{i=1}^n f_i$ to $\mathbb{E}_i f_i$), or you just intentionally choose the finite-sum problem in the study?
>
> **A:** If there are an infinite number of $f_i$, our reformulation and algorithm cannot be applied directly, because the dual variable $y$ becomes infinite-dimensional. Hu et al. (2020) considered a more general setting that covers the stochastic problem, but their algorithm requires a large batch size to converge (in both finite-sum and stochastic settings). Given the broad applicability of the finite-sum structure in machine learning, we believe this is an important class to study.
>
> **Refs:**
>
> [1] Lan, 2020. First-order and Stochastic Optimization Methods for Machine Learning.
>
> [2] Wang et al., 2017. Inexact proximal stochastic gradient method for convex composite optimization.

---

> > ### Comment · Reviewer_7wWG · 2025-04-07
> >
> > Thank you for the reply, I will keep the score.

---

### Decision · Program_Chairs · 2025-05-01

**Decision:**

Accept (poster)

**Comment:**

This submission proposes a new algorithm for solving finite sum compositional convex optimization problem. The authors also provide a lower bound to certify near-optimality of their algorithm.

The reviewers are in agreement in their acceptance recommendation. In my reading, the work can improve its positioning with respect to the literature and provide clearer comparisons with the related work.